# On the Complexity of Verifying Quantized GNNs with Readout

## Abstract

In this paper, we introduce a logical language for reasoning about quantized graph neural networks (GNNs) with Global Readout. We then prove that verifying quantized GNNs with Global Readout is NEXPTIME-complete. We also experimentally show the relevance of quantization in the context of ACR-GNNs.

## 1 Introduction

Graph neural networks (GNNs) are models used for classification and regression tasks on graphs or graph-node pairs, aka pointed graphs. GNNs are applied for recommendation in social network [30], knowledge graphs [40], chemistry [29], drug discovery [39], etc.

Quantization designates the fact that numbers are represented by a small amount of bits, opposed to e.g., integers or real numbers whose number of bits can be arbitrary long. Standard IEEE 754 64-bit floats, INT8, or FP8 [22] enter in our setting. Essentially, our setting reflects GNNs as they are practically implemented (e.g., in PyTorch), rather than idealized GNNs that assume integer or perfect mathematical real number weights, as studied in previous research comparing GNNs and logic [4], [24] or [8].

GNNs, as several other machine learning models are difficult to interpret, understand and verify. This is a major issue for their adoption, morally and legally, with the enforcement of regulatory policies like the EU AI Act [13]. In the literature, verifying quantized GNNs has already been addressed [32]. The methodology is to design a logical language to represent both the properties to check and the computation of a GNN. However, global readout has not been considered whereas it is an essential element of GNNs, especially for graph classification.

In this paper, we focus on verifying Aggregate-Combine Graph Neural Networks with global Readout (ACR-GNNs) and we design a logical framework called $q\mathcal{L}$.

**Example 1.** *Assume a class of knowledge graphs (KGs) representing communities of people and animals, where each node corresponds to an individual. Each individual can be Animal, Human, Leg, Fur, White, Black, etc. These concepts can be encoded with features $x_0, x_1, \ldots, x_5, \ldots$ respectively, taking values $0$ or $1$. Edges in a KG represent a generic 'has' relationship: a human can have an animal (pet); an animal can have a human (owner), a leg, a fur; a fur can have a color; etc. Suppose that $\mathcal{A}$ is a GNN processing those KGs and is trained to supposedly recognize dogs. We can verify that the nodes recognized by $\mathcal{A}$ are animals—arguably a critical property of the domain—by checking the validity (i.e., the non-satisfiability of the negation) of $\varphi_{\mathcal{A}} \to x_0 = 1$ where $\varphi_{\mathcal{A}}$ is a $q\mathcal{L}$-formula corresponding to $\mathcal{A}$'s computation, true in exactly the pointed graphs accepted by $\mathcal{A}$. Ideally, $\mathcal{A}$ should not overfit the concept of dog as a perfect prototypical animal. For instance, three-legged dogs do exist. We can verify that $\mathcal{A}$ lets it be a possibility by checking the satisfiability of the formula $\varphi_{\mathcal{A}} \wedge \Diamond^{\leq 3}(x_2 = 1)$.*

Submitted to 39th Conference on Neural Information Processing Systems (NeurIPS 2025). Do not distribute.

*More complex $q\mathcal{L}$ formulas can be written to express graph properties to be evaluated against an ACR-GNN, that will be formalized later in Example 2: 1. Has a human owner, whose pets are all two-legged. 2. A human in a community that has more than twice as many animals as humans, and more than five animals without an owner[1]. 3. An animal in a community where some animals have white and black fur.*

**Contribution.** In Section 3, we define logic $q\mathcal{L}$ extending the one from [32] for capturing global readout. It is expressive enough to capture quantized ACR-GNNs with arbitrary activation functions. Moreover, $q\mathcal{L}$ can serve as a flexible graph property specification language reminiscent of modal logics [9], for expressing e.g. properties 1-3 in Example 1.

Section 4 shows that the satisfiability problem of $q\mathcal{L}$ is in NEXPTIME, i.e. it can be decided by a non-deterministic algorithm in exponential time. To do that, we reuse the concept of mathematical logic called Hintikka sets [] which are complete sets of subformulas that can be true at a given vertex of a graph. We then introduce a quantized variant of Quantifier-Free Boolean algebra Presburger Arithmetic (QFBAPA) logic, denoted by $QFBAPA_{\mathbb{K}}$, and prove that it is in NP as the original QFBAPA on integers. We then reduce the satisfiability problem of $q\mathcal{L}$ to the one of $QFBAPA_{\mathbb{K}}$.

In Section 5, we then prove that $q\mathcal{L}$ is NEXPTIME-complete, while it is PSPACE-complete without global readout [32]. In a similar way, we also add global counting to the logic $K^\sharp$ previously introduced by [24]. We show that it corresponds to AC-GNNs over $\mathbb{Z}$ with global readout and trReLU activation functions. We prove that the satisfiability problem is NEXPTIME-complete, partially addressing a problem left open in the literature—that is, for the case of integer values and trReLU activation functions [7, 8]. Details are in the appendix for keep the main text concise.

As NEXPTIME is highly intractable, in Section 6, we relax the satisfiability problem of $q\mathcal{L}$ and ACR-GNNs, searching graph counterexamples whose number of vertices is bounded. This problem is NP-complete. We provide an implementation in this line.

We experimentally show in Section 7 that quantization of GNNs provide minimal accuracy degradation. Our results confirm that the quantized models retain strong predictive performance while achieving substantial reductions in model size and inference cost. These findings demonstrate the practical viability of quantized ACR-GNNs for deployment in resource-constrained environments.

**Related work.** [4] showed that ACR-GNNs are capable of capturing the expressive power of $FOC_2$, that is, two-variable first-order logic with counting. Recent work has explored the logical expressiveness of GNN variants in more detail. Notably, [24] and [7] introduced logics to exactly characterize the capabilities of different forms of GNNs. Similarly, [11] analyzed Max-Sum-GNNs through the lens of Datalog. [32] considered the expressivity of GNN with quantized parameters but without global readout.

On the verification side, [17] studied the complexity of verification of quantized feedforward neural networks (FNNs), while [31, 34] investigated reachability and reasoning problems for general FNNs and GNNs. Approaches to verification are proposed via integer linear programming (ILP) by [18] and [41], and via model checking by [33].

From a logical perspective, reasoning over structures involving arithmetic constraints is closely tied to several well-studied logics. Relevant work includes Kuncak and Rinard's decision procedures for QFBAPA ([20]), as well as developments by [12], [2], [6], and [14]. These logics form the basis for the characterizations established in [24, 7].

Quantization techniques have studied in neural networks, with surveys such as [15, 23] providing comprehensive overviews focused on maintaining model accuracy. Although most practical advancements target convolutional neural networks (CNNs), many of the underlying principles extend to GNNs as well ([42]). NVIDIA has demonstrated hardware-ready quantization strategies ([38]), and frameworks like PyTorch ([1]) support both post-training quantization and quantization-aware training (QAT), the latter simulating quantization effects during training to improve low-precision performance. QAT has been particularly effective in closing the gap between quantized and full-precision models, especially for highly compressed or edge-deployed systems ([19]). In the context of GNNs, [35] proposed Degree-Quant, incorporating node degree information to mitigate quantization-related issues. Based

---

[1]Interestingly, $q\mathcal{L}$ goes beyond graded modal logic and even first-order logic. The property of item 2 in Example 1 cannot be expressed in FOL.

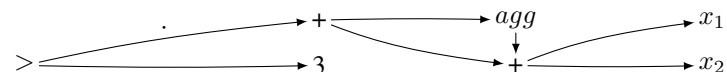

Figure 1: DAG data structure for the formula $agg(x_1 + x_2) + (x_1 + x_2) \geq 3$.

86  on this, [43] introduced $A^2Q$, a mixed-precision framework that adapts bitwidths on graph topology
87  to achieve high compression with minimal performance loss.

## 2   Background

89  Let $\mathbb{K}$ be a set of quantized numbers, and let $n$ denote the *bitwidth* of $\mathbb{K}$, that is, the number of bits
90  required to represent a number in $\mathbb{K}$. The bitwidth $n$ is written in unary; this is motivated by the fact
91  that $n$ is small and that we would in any case need to allocate $n$-bit consecutive memory for storing a
92  number. Formally, we consider a sequence $\mathbb{K}_1, \mathbb{K}_2, \ldots$ corresponding to bitwidths 1, 2, etc., but we
93  retain the notation $\mathbb{K}$ for simplicity. We suppose that $\mathbb{K}$ saturates: e.g., if $x \geq 0$, $y \geq 0$, $x + y \geq 0$
94  (i.e., no modulo behavior like in `int` in C for instance). We suppose that $1 \in \mathbb{K}$.

95  We consider Aggregate-Combine Graph Neural Networks with global Readout (ACR-GNNs), a
96  standard class of message-passing GNNs [4, 16]. An ACR-GNN layer is defined by a triple
97  $(comb, agg, agg_g)$, where $comb : \mathbb{K}^{3m} \to \mathbb{K}^n$ is a combination function, and $agg$, $agg_g$ are lo-
98  cal and global aggregation functions that map multisets of vectors in $\mathbb{K}^m$ to a single vector in
99  $\mathbb{K}^m$.

100  An ACR-GNN is composed of a sequence of such layers $(\mathcal{L}^{(1)}, \ldots, \mathcal{L}^{(L)})$ followed by a final
101  classification function $cls : \mathbb{K}^m \to \{0, 1\}$. Given a graph $G = (V, E)$ and an initial node labelling
102  $x_0 : V \to \{0, 1\}^k$, the state of a node $u$ in layer $i$ is recursively defined as:

$$x_i(u) = comb(x_{i-1}(u), \ agg(\{\{x_{i-1}(v) \mid uv \in E\}\}), \ agg_g(\{\{x_{i-1}(v) \mid v \in V\}\}))$$

103  The final output of the GNN for a pointed graph $(G, u)$ is $\mathcal{A}(G, u) = cls(x_L(u))$. A more detailed
104  definition is provided in Appendix C.2.

105  Our study focuses on a specific subclass where both $agg$ and $agg_g$ perform summation over vectors,
106  and where $comb(x, y, z) = \vec{\sigma}(xC + yA_1 + zA_2 + b)$, using matrices $C, A_1, A_2$ with entries from
107  $\mathbb{K}$, and a bias $b \in \mathbb{K}$. The classification function is a linear threshold: $cls(x) = \sum_i a_i x_i \geq 1$ with
108  weights $a_i \in \mathbb{K}$. Moreover, we assume that all arithmetic operations are executed according to the
109  arithmetic related to $\mathbb{K}$. It is assumed that the context makes clear the $\mathbb{K}$ and arithmetic being used.
110  We note $[[\mathcal{A}]]$ the set of pointed graphs $(G, u)$ such that $\mathcal{A}(G, u) = 1$. An ACR-GNN $\mathcal{A}$ is satisfiable
111  if $[[\mathcal{A}]]$ is non-empty. The *satisfiability problem* for ACR-GNNs is: Given a ACR-GNN $\mathcal{A}$, decide
112  whether $\mathcal{A}$ is satisfiable.

## 3   Logic $q\mathcal{L}$ for Representing GNN Computations and Properties on Graphs

114  We set up a logical framework called $q\mathcal{L}$ extending the logic in [32] with global aggregation: it is a
115  *lingua franca* to represent GNN computation and properties on graphs.

**Syntax.**   Let $F$ be a finite set of features and $\mathbb{K}$ be some finite-width arithmetic. We consider a set
of *expressions* defined by the following grammar in Backus-Naur form:

$$\vartheta ::= c \mid x_i \mid \alpha(\vartheta) \mid agg(\vartheta) \mid agg_\forall(\vartheta) \mid \vartheta + \vartheta \mid c \times \vartheta$$

116  where $c$ is a number in $\mathbb{K}$, $x_i$ is a feature in $F$, $\alpha$ is a symbol for denoting the activation function, and
117  $agg$ and $agg_\forall$ denote the aggregation function for local and global readout respectively. A *formula* is
118  a construction of the formula $\vartheta \geq k$ where $\vartheta$ is an expression and $k$ is an element of $\mathbb{K}$. If $-1 \in \mathbb{K}$,
119  and $-\vartheta$ is not, we can write $-\vartheta$ instead of $(-1) \times \vartheta$. Other standard abbreviations can be used.

120  Formulas are represented as direct acyclic graphs, aka circuits, meaning that we do not repeat the same
121  expressions several times. For instance, the formula $agg(x_1 + x_2) + (x_1 + x_2) \geq 3$ can be represented
122  as the DAG given in Figure 1. Formulas can also be represented by a sequence of assignments via
123  new fresh intermediate variables. For instance: $y := x_1 + x_2, z := agg(y) + y, res := z \geq 3$.

124 **Semantics.** Consider a graph $G = (V, E)$, where vertices in $V$ are labeled via a labeling function
125 $\ell : V \to \mathbb{K}^n$ with feature values. The value of an expression $\vartheta$ in a vertex $u \in V$ is denoted by
126 $[[\vartheta]]_{G,u}$ and is defined by induction on $\vartheta$:

127
$$[[c]]_{G,u} = c,$$
$$[[x_i]]_{G,u} = \ell(u)_i,$$
$$[[\vartheta + \vartheta']]_{G,u} = [[\vartheta]]_{G,u} +_{\mathbb{K}} [[\vartheta']]_{G,u},$$

$$[[c \times \vartheta]]_{G,u} = c \times_{\mathbb{K}} [[\vartheta]]_{G,u},$$
$$[[\alpha(\vartheta)]]_{G,u} = [[\alpha]]([[\vartheta]]_{G,u}),$$
$$[[agg(\vartheta)]]_{G,u} = \Sigma_{v|uEv}[[\vartheta]]_{G,v},$$
$$[[agg_{\forall}(\vartheta)]]_{G,u} = \Sigma_{v \in V}[[\vartheta]]_{G,v},$$

128 We define $[[\vartheta \geq k]] = \{G, u \mid [[\vartheta]]_{G,u} \geq_{\mathbb{K}} [[k]]_{G,u}\}$ (we write $\geq$ for the symbol in the syntax and
129 $\geq_{\mathbb{K}}$ for the comparison in $\mathbb{K}$). A formula $\varphi$ is satisfiable if $[[\varphi]]$ is non-empty. The *satisfiability*
130 *problem* for $q\mathcal{L}$ is: Given a $q\mathcal{L}$-formula $\varphi$, decide whether $\varphi$ is satisfiable.

131 **ACR-GNN verification tasks.** We are interested in the following decision problems. Given a GNN
132 $\mathcal{A}$, and a $q\mathcal{L}$ formula $\varphi$: (VT1, sufficiency) Do we have $[[\varphi]] \subseteq [[\mathcal{A}]]$? (VT2, necessity) Do we have
133 $[[\mathcal{A}]] \subseteq [[\varphi]]$? (VT3, consistency) Do we have $[[\varphi]] \cap [[\mathcal{A}]] \neq \emptyset$?

134 **Representing a GNN computation.** To reason formally about ACR-GNNs, we represent their
135 computations using $q\mathcal{L}$. Logic $q\mathcal{L}$ facilitates the modeling of the acceptance condition of ACR-GNNs.

136 We explain this via example. Consider a two-layer ACR-GNN $\mathcal{A}$ with input and output dimension 2,
137 using summation for aggregation, activation via $\alpha(x) := \max(0, \min(1, x))$—the truncated ReLU—
138 and a classification function $2x_1 - x_2 \geq 1$. The combination functions are:

$$comb_1((x_1, x_2), (y_1, y_2), (z_1, z_2)) := \begin{pmatrix} \sigma(2x_1 + x_2 + 5y_1 - 3y_2 + 1) \\ \sigma(-x_1 + 4x_2 + 2y_1 + 6y_2 - 2) \end{pmatrix},$$

$$comb_2((x_1, x_2), (y_1, y_2), (z_1, z_2)) := \begin{pmatrix} \sigma(3x_1 - y_1 + 2z_2) \\ \sigma(-2x_1 + 5y_2 + 4z_1) \end{pmatrix}.$$

139 Note that this assumes that $\mathcal{A}$ operates over $\mathbb{K}$ with at least three bits. Then, the corresponding
140 $q\mathcal{L}$ formula $\varphi_{\mathcal{A}}$ is given by: $\psi_1 = \alpha(2x_1 + x_2 + 5agg(x_1) - 3agg(x_1) + 1)$, $\psi_2 := \alpha(-x_1 +$
141 $4x_2 + 2agg(x_1) + 6agg(x_2) - 2)$, $\chi_1 := \alpha(3\psi_1 - agg(\psi_1) + 2(agg_{\forall}(psi_2)))$, $\chi_2 := \alpha(-2\psi_1 +$
142 $5(agg(\psi_2)) + 4agg_{\forall}(psi1))$, $\varphi_A := 2(\chi_1) - \chi_2 \geq 1$. To sum up, given a GNN $\mathcal{A}$, we compute
143 $q\mathcal{L}$-formula in poly-time in the size of $\mathcal{A}$ with $[[\mathcal{A}]] = [[\varphi_{\mathcal{A}}]]$ (as done in [32]).

144 **Simulating a modal logic in the logic $q\mathcal{L}$.** In this section, we show that extending $q\mathcal{L}$ with
145 modal operators [9] does not increase the expressivity. We can even compute an equivalent $q\mathcal{L}$
146 without Boolean connectives and without modal operators in poly-time. It means that formulas like
147 $\varphi_{\mathcal{A}_1} \to x_0 = 1$ or $\varphi_{\mathcal{A}_1} \wedge \Diamond^{\leq 3}(x_2 = 1)$ have equivalent formulas in $q\mathcal{L}$.

148 Assume that $\alpha$ is ReLU. Let $Atm_0$ be the set of atomic formulas of $q\mathcal{L}$ of the form $\vartheta \geq 0$. We
149 suppose that $\vartheta$ takes integer values. In general, $\vartheta \geq k$ is an atomic formula equivalent to $\vartheta - k \geq 0$.
150 Without loss of generality, we thus assume that formulas of $q\mathcal{L}$ are over $Atm_0$. Let modal $q\mathcal{L}$ be the
151 propositional logic on $Atm_0$ extended with modalities and a restricted variant of graded modalities
152 where number $k$ in $\mathbb{K}$.

$$[[\Box\varphi]] = \{G, u \mid G, v \in [[\varphi]] \text{ for every } v \text{ s.t. } uEv\}$$
$$[[\Box_g\varphi]] = \{G, u \mid G, v \in [[\varphi]] \text{ for every } v \text{ in } V\}$$

153
$$[[\Diamond^{\geq k}\varphi]] = \{G, u \mid |\{G, v \mid uEv \text{ and } G, v \in [[\varphi]]\}| \geq_{\mathbb{K}} k\} \quad [[\Diamond_g^{\geq k}\varphi]] = \{G, u \mid |[[\varphi]]| \geq_{\mathbb{K}} k\}$$

154 and modalities $\Diamond^{\leq k}\varphi$ and $\Diamond_g^{\leq k}\varphi$ defined the same way but with $\leq_{\mathbb{K}}$. We can turn back to the graph
155 properties mentioned in Example 1.

156 **Example 2.** *We first define a few simple formulas to characterize the concepts of the domain. Let*
157 $\varphi_A := x_0 = 1$ *(Animal),* $\varphi_H := x_1 = 1$ *(Human),* $\varphi_L := x_2 = 1$ *(Leg),* $\varphi_F := x_3 = 1$ *(Fur),*
158 $\varphi_W := x_4 = 1$ *(White), and* $\varphi_B := x_5 = 1$ *(Black).*

159     *1. Has a human owner, whose all pets are two-legged:* $\Diamond(\varphi_H \wedge \Box(\varphi_A \to \Diamond^{=2}\varphi_L))$.
160     *2. A human in a community that has more that twice as many animals as humans, and more than*
161        *five animals without an owner:* $\varphi_H \wedge (agg_{\forall}(x_0) - 2 \times agg_{\forall}(x_1) \geq 0) \wedge \Diamond_g^{\geq 5}((\varphi_A \wedge \Box(\neg\varphi_H))$.

162 *3. An animal in a community where some animals have white and black fur:*
163 $$\varphi_A \wedge \Diamond_g(\Diamond(\varphi_F \wedge \Diamond\varphi_W) \wedge \Diamond(\varphi_F \wedge \Diamond\varphi_B)).$$

164 We can see the boolean operator $\neg$, and the various modalities as functions from $Atm_0$ into $Atm_0$,
165 and the boolean operator $\vee$ as a function from $Atm_0 \times Atm_0$ to $Atm_0$.

$$f_\neg(\vartheta \geq 0) := -\vartheta - 1 \geq 0 \qquad f_\vee(\vartheta_1 \geq 0, \vartheta_2 \geq 0) := \vartheta_1 + ReLU(\vartheta_2 - \vartheta_1) \geq 0$$
$$f_\Box(\vartheta \geq 0) := agg(-ReLU(-\vartheta)) \geq 0$$
$$f_{\Diamond^{\geq k}}(\vartheta \geq 0) := agg(ReLU(\vartheta + 1) - ReLU(\vartheta)) - k \geq 0$$
$$f_{\Diamond^{\leq k}}(\vartheta \geq 0) := k - agg(ReLU(\vartheta + 1) - ReLU(\vartheta)) \geq 0$$

166 For the corresponding global modalities ($f_{\Box_g}(\vartheta \geq 0)$, $f_{\Diamond^{\geq k}}(\vartheta \geq 0)$, and $f_{\Diamond^{\leq k}}(\vartheta \geq 0)$), it suffices to
167 use $agg_\forall$ in place of $agg$. The previous transformations can be generalized to arbitrary formulas of
168 modal $q\mathcal{L}$ as follows.

$$mod2expr(\vartheta \geq 0) := \vartheta \geq 0 \qquad mod2expr(\neg\varphi) := f_\neg(mod2expr(\varphi))$$
$$mod2expr(\varphi_1 \vee \varphi_2) := f_\vee(mod2expr(\varphi_1), mod2expr(\varphi_2))$$
$$mod2expr(\boxplus\varphi) := f_\boxplus(mod2expr(\varphi)), \qquad \boxplus \in \{\Box, \Box_g, \Diamond^{\geq k}, \Diamond_g^{\geq k}, \Diamond^{\leq k}, \Diamond_g^{\leq k}\}$$

169 We can show that formulas of modal $q\mathcal{L}$ can be captured by a unique expression $\vartheta \geq 0$. This is a
170 consequence of the following lemma [2].

171 **Lemma 3.** *Let $\varphi$ be a formula of modal $q\mathcal{L}$. The formulas $\varphi$ and $mod2expr(\varphi)$ are equivalent.*

172 Now, ACR-GNN verification tasks can be solved by reduction to the satisfiability problem of $q\mathcal{L}$.
173 VT1 by checking that $\varphi \wedge \neg\varphi_\mathcal{A}$ is not satisfiable; VT2 by checking that $\neg\varphi \wedge \varphi_\mathcal{A}$ is not satisfiable;
174 VT3 by checking that $\varphi \wedge \varphi_\mathcal{A}$ is satisfiable.

## 175 4 NEXPTIME Membership of the Satisfiability Problem

176 In this section, we prove the NEXPTIME membership of reasoning in modal quantized logic, and
177 also of solving of ACR-GNN verification tasks (by reduction to the former). Remember that the
178 activation function $\alpha$ can be arbitrary in our setting. Our result holds with the loose restriction that
179 $[[\alpha]]$ is computable in exponential-time in the bit-width $n$ of $\mathbb{K}$.

180 **Theorem 4.** *The satisfiability problem of $q\mathcal{L}$ is decidable and in NEXPTIME, and so is VT3. VT1*
181 *and VT2 are in coNEXPTIME.*

182 In order to prove Theorem 4, we adapt the NEXPTIME membership of the description logic
183 $\mathcal{ALCSCC}^{++}$ from [2] to logic $q\mathcal{L}$. The difference resides in the definition of Hintikka sets and
184 the treatment of quantization. The idea is to encode the constraints of a $q\mathcal{L}$-formula $\varphi$ in a formula of
185 exponential length of a quantized version of QFBAPA, that we prove to be in NP.

### 186 4.1 Hintikka Sets

187 Consider $q\mathcal{L}$-formula $\varphi$. Let $E(\varphi)$ be the set of subexpressions in $\varphi$. For instance, if $\varphi$ is
188 $3 \times agg(\alpha(x_2 + agg_\forall(x_1))) \geq 5$ then $E(\varphi) := \{agg(\alpha(x_2 + agg_\forall(x_1))), \alpha(x_2 + agg_\forall(x_1)), x_2,$
189 $agg_\forall(x_1), x_1\}$. From now on, we consider equality subformulas that are of the form $\vartheta = k$ where $\vartheta$ is
190 a subexpression of $\varphi$ and $k \in \mathbb{K}$.

191 **Definition 5.** *A Hintikka set $H$ for $\varphi$ is a subset of subformulas of $\varphi$ such that:*

192 *1. For all $\vartheta \in E(\varphi)$, there is a unique value $k \in \mathbb{K}$ such that $\vartheta = k \in H$*
193 *2. $\vartheta_1 = k_1, \vartheta_2 = k_2 \in H$ then $\vartheta_1 + \vartheta_2 = k_1 + k_2 \in H$*
194 *3. If $\vartheta \geq k \in H$ then $c \times \vartheta = k' \in H$ where $k' = c \times_\mathbb{K} k$*
195 *4. $\vartheta = k \in H$ and $\alpha(\vartheta) = k'$ implies $k' = [[\alpha]](k)$*

---

[2]For simplicity, we do not present how to handle $\vartheta \geq 0$ when $\vartheta$ is not an integer. We could introduce several
activation functions $\alpha$ in $q\mathcal{L}$, one of them could be interpreted as the Heavyside step function. In the sequel
Definition 5, Point 4 is just repeated for each $\alpha$.

Informally, a *Hintikka set* is a set of equality subformulas obtained from a choice of a value for each subexpression of $\varphi$ (point 1), provided that the set is consistent *at the current vertex* (point 2-4). Note that the notion of Hintikka set does not take any constraints about $agg$ and $agg_\forall$ into consideration since checking consistency of aggregation would require information about the neighbor or the whole graph.

**Example 6.** *If $\varphi$ is $3 \times agg(\alpha(x_2 + agg_\forall(x_1))) \geq 5$ then the following set is an example of Hintikka set:* $\{agg(\alpha(x_2 + agg_\forall(x_1)) = 8, \alpha(x_2 + agg_\forall(x_1)) = 9, x_2 + agg_\forall(x_1) = 9, x_2 = 7, agg_\forall(x_1) = 2, x_1 = 5\}$.

**Proposition 7.** *The number of Hintikka sets is bounded by $2^{n|\varphi|}$ where $|\varphi|$ is the size of $\varphi$, and $n$ is the bitwidth of $\mathbb{K}$.*

## 4.2 Quantized Version of QFBABA (Quantifier-free Boolean Algebra and Presburger Arithmetics)

A QFBAPA formula is propositional formula where each atom is either an inclusion of sets or equality of sets or linear constraints [20]. Sets are denoted by Boolean algebra expression e.g., $(S \cup S') \setminus S''$, or $\mathcal{U}$ where $\mathcal{U}$ denotes the set of all points in some domain. Here $S$, $S'$, etc. are set variables. Linear constraints are over $|S|$ denoting the cardinality of the set denoted by the set expression $S$. For instance, the QFBAPA-formula $(pianist \subseteq happy) \wedge (|happy| + |\mathcal{U} \setminus pianist| \geq 6) \wedge (|happy| < 2)$ is read as 'all pianists are happy and the number of happy persons + the number of persons that are not pianists is greater than 6 and the number of happy persons is smaller than 2'.

We now introduce a *quantized* version QFBAPA$_\mathbb{K}$ of QFBAPA. It has the same syntax as QFBAPA except that hard-coded numbers in expressions are in $\mathbb{K}$. Concerning the semantics, every numerical expression is interpreted in $\mathbb{K}$. For each set expression $S$, the interpretation of $|S|$ is not the cardinality $c$ of the interpretation of $S$, but the result of the computation $1 + 1 + \ldots + 1$ in $\mathbb{K}$ with $c$ occurrences of 1 in the sum.

We consider that $\mathbb{K}$ that saturates, meaning that if $x + y$ exceed the upper bound limit of $\mathbb{K}$, there is a special value denoted by $+\infty$ such that $x + y = +\infty$.

**Proposition 8.** *If bitwidth $n$ is in unary, and if $\mathbb{K}$ saturates, then satisfiability in QFBAPA$_\mathbb{K}$ is in NP.*

## 4.3 Reduction to QFBAPA$_\mathbb{K}$

Let $\varphi$ be a formula of $q\mathcal{L}$. For each Hintikka set $H$, we introduce the set variable $X_H$ that intuitively represents the $H$-vertices, i.e., the vertices in which subformulas of $H$ hold. The following QFBAPA$_\mathbb{K}$-formulas say that the interpretation of $X_H$ form a partition of the universe. For each subformula $\vartheta' = k$, we introduce the set variable $X_{\vartheta'=k}$ that intuitively represents the vertices in which $\vartheta' = k$ holds. Formula (1) expresses that $\{X_H\}_H$ form a partition of the universe. Formula (2) makes the bridge between variables $X_{\vartheta'=k}$ and $X_H$.

$$( \bigwedge_{H \neq H'} X_H \cap X_{H'} = \emptyset) \wedge (\bigcup_H X_H = \mathcal{U}) \quad (1) \qquad \bigwedge_{\vartheta' \in E(\varphi)} \bigwedge_{k \in \mathbb{K}} (X_{\vartheta'=k} = \bigcup_{H | \vartheta'=k \in H} X_H) \quad (2)$$

We introduce also a variable $S_H$ that denotes the set of all successors of some $H$-vertex. If there is no $H$-vertex then the variable $S_H$ is just irrelevant.

The following QFBAPA$_\mathbb{K}$-formula encodes the semantics of $agg(\vartheta)$. More precisely, it says that for all subexpressions $agg(\vartheta)$, for all values $k$, for all Hintikka sets $H$ containing subformula $agg(\vartheta)=k$, for all $H$ containing $agg(\vartheta)=k$, it says that, if there is some $H$-vertex (i.e., vertices in $S_H$), then the aggregation obtained by summing over the successors of some $H$-vertex is $k$.

$$\bigwedge_{agg(\vartheta) \in E(\varphi)} \bigwedge_{k \in \mathbb{K}} \bigwedge_{\substack{\text{Hintikka set } H \\ | \, agg(\vartheta)=k \, \in \, H}} [(X_H \neq \emptyset) \to \sum_{k' \in \mathbb{K}} |S_H \cap X_{\vartheta=k'}| \times k' = k] \quad (3)$$

In the previous sum, we partition $S_H$ into subsets $S_H \cap X_{\vartheta=k'}$ for all possible values $k'$. Each contribution for a successor in $S_H \cap X_{\vartheta=k'}$ is $k'$. We rely here on the fact[3] that $(1+1+\ldots+1) \times k' =$

---

[3]This is true for some fixed-point arithmetics but not for floating-point arthmetics. See Appendix B.

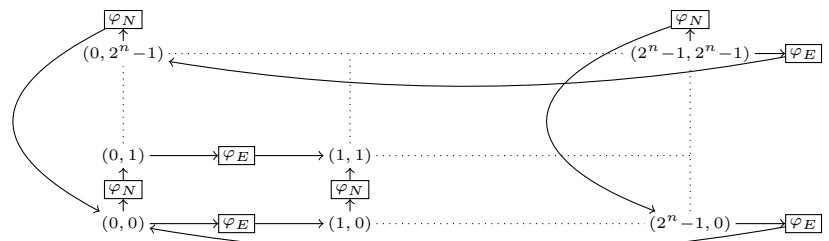

Figure 2: Encoding a torus of exponential size with (modal) $q\mathcal{L}$ formulas. $(x, y)$ are the vertices of the graph that correspond to locations in the torus while $\varphi_N$ and $\varphi_E$ denote intermediate vertices indicating the direction (resp., north and east).

$k' + k' + \ldots + k'$. We also fix a specific order over values $k'$ in the summation (it means that $agg(\vartheta)$ is computed as follows: first order the successors according to the taken values of $\vartheta$ in that specific order, then perform the summation). Finally, the semantics of $agg_\forall$ is captured by the formula:

$$\bigwedge_{agg_\forall(\vartheta) \in E(\varphi)} \bigwedge_{k \in \mathbb{K}} X_{agg_\forall(\vartheta)=k} \neq \emptyset \rightarrow \sum_{k' \in \mathbb{K}} |X_{\vartheta=k'}| \times k' = k \qquad (4)$$

Note that intuitively Formula (4) implies that for $X_{agg_\forall(\vartheta)=k}$ is interpreted as the universe, for the value $k$ which equals the semantics of $\sum_{k' \in \mathbb{K}} |X_{\vartheta=k'}| \times k'$.

Given $\varphi = \vartheta \geq k$, we define $tr(\varphi) := \psi \wedge \bigvee_{k' \geq k} X_{\vartheta=k'} \neq \emptyset$ where $\psi$ the conjunction of Formulas 1–4. The function $tr$ requires to compute all the Hintikka sets. So we need in particular to check Point 4 of Definition 5 and we get the following when $[[\alpha]]$ is computable in exponential time in $n$.

**Proposition 9.** $tr(\varphi)$ is computable in exponential-time in $|\varphi|$ and $n$.

**Proposition 10.** Let $\varphi$ be a formula of $q\mathcal{L}$. $\varphi$ is satisfiable iff $tr(\varphi)$ is QFBAPA$_\mathbb{K}$ satisfiable.

Finally, in order to check whether a $q\mathcal{L}$-formula $\varphi$ is satisfiable, we construct a QFBAPA$_\mathbb{K}$-formula $tr(\varphi)$ in exponential time. As the satisfiability problem of QFBAPA$_\mathbb{K}$ is in NP, we obtain that the satisfiability problem of $q\mathcal{L}$ is in NEXPTIME. We proved Theorem 4,

**Remark 11.** *Our methodology can be generalized to reason in subclasses of graphs. For instance, we may tackle the problem of satisfiability in a graph where vertices are of bounded degree bounded by $d$. To do so, we add the constraint $\bigwedge_H |S_H| \leq d$.*

## 5    Complexity Lower Bound

The NEXPTIME upper-bound is tight. Having defined modalities in $q\mathcal{L}$ and stated Lemma 3, Theorem 12 is proven by adapting the proof of NEXPTIME-hardness of deciding the consistency of $\mathcal{ALCQ}$-$T_C$Boxes presented in [36]. So we already have the hardness result for ReLU.

NEXPTIME-hardness is proven via a reduction from the tiling problem by Wang tiles of a torus of size $2^n \times 2^n$. A Wang tile is a square with colors, e.g., ◪, ◩, etc. That problem takes as input a number $n$ in unary, and Wang tile types, and an initial condition – let say the bottom row is already given. The objective is to decide whether the torus of $2^n \times 2^n$ can be tiled while colors of adjacent Wang tiles match. A slight difficulty resides in adequately capturing a two-dimensional grid structure—as in Figure 2—with only a single relation. To do that, we introduce special formulas $\varphi_E$ and $\varphi_N$ to indicate the direction (east or north). In the formula computed by the reduction, we also need to bound the number of vertices corresponding to tile locations by $2^n \times 2^n$. Thus $\mathbb{K}$ needs to encode $2^n \times 2^n$. We need a bit-width of at least $2n$.

**Theorem 12.** *The satisfiability problem in $q\mathcal{L}$ is NEXPTIME-hard, and so is* VT3. VT1 *and* VT2 *are coNEXPTIME-hard.*

**Remark 13.** *It turns out that the verification task only needs the fragment of $q\mathcal{L}$ where $agg$ is applied directly on an expression $\alpha(..)$. Indeed, this is the case when we represent a GNN in $q\mathcal{L}$ or when we translate logical formulas in $q\mathcal{L}$ (Lemma 3). Reasoning about $q\mathcal{L}$ when $\mathbb{K} = \mathbb{Z}$ and the activation function is truncated ReLU is also NEXPTIME-complete (see Appendix E).*

## 6  Bounding the Number of Vertices

The satisfiability problem is NEXPTIME-complete, thus far from tractable. The complexity comes essentially because counterexamples can be arbitrary large graphs. However, usually we are search for small counterexamples. Let $\mathcal{G}^{\leq N}$ be the set of pointed graphs with at most $N$ vertices. We consider the $q\mathcal{L}$ and ACR-GNN *satisfiability problems with a bound on the number of vertices*: given a number $N$ given in unary, 1. given a $q\mathcal{L}$-formula $\varphi$, is it the case that $[[\varphi]] \cap \mathcal{G}^{\leq N} \neq \emptyset$, 2. given an ACR-GNN $\mathcal{A}$, is it the case that $[[\mathcal{A}]] \cap \mathcal{G}^{\leq N} \neq \emptyset$.

**Theorem 14.** *The satisfiability problems with bounded number of vertices are NP-complete.*

We then can extend the methodology of [33] but for verifying GNNs. Our implementation proposal is a Python program that takes a learnt quantized GNN $\mathcal{A}$ as an input, a precondition, a postcondition and a bound $N$. It then produces a C program that mimics the execution of $\mathcal{A}$ on an arbitrary graph with at most $N$ vertices, and embeds the pre/postcondition. We then apply ESBMC (efficient SMT-based context-bounded model checker) [21] on the C program.

## 7  Quantization Effects on Accuracy, Performance and Model Size

To confirm that the GNN models considered in this paper are promising, we now investigate the application of Dynamic Post-Training Quantization (PTQ) to Aggregate-Combined Readout Graph Neural Networks (ACR-GNNs). Our experimental design builds on the framework introduced in [4], using their publicly available implementation [5] as the baseline. ACR-GNNs with specific structural configurations are used as the primary model class for evaluation. Dynamic PTQ, implemented in PyTorch [1, 26], converts a pre-trained floating-point model into a quantized version without retraining. This approach quantizes weights to INT8 statically, while activations remain in floating point until dynamically quantized at compute time. This enables efficient INT8-based computation, reducing memory usage and improving inference speed. PyTorch's implementation employs per-tensor quantization for weights and stores activations in floating-point format between operations. The evaluation focuses on accuracy, model size, and latency. Experiments are conducted on both synthetic and real-world datasets, with the synthetic benchmark—based on dense Erdös–Rényi graph structures and logical labeling schemes—serving as the primary focus.

The synthetic graphs were generated using the dense Erdös–Rényi model, a classical approach for constructing random graphs. Each graph includes five initial node colours, encoded as one-hot feature vectors. Following [4], labels were assigned using formulas from the logic fragment $FOC_2$. Specifically, a hierarchy of classifiers $\alpha_i(x)$ was defined as:

$$\alpha_0(x) := \mathrm{Blue}(x), \quad \alpha_{i+1}(x) := \exists^{[N,M]} y \left( \alpha_i(y) \wedge \neg E(x,y) \right)$$

where $\exists^{[N,M]}$ denotes the quantifier "there exist between $N$ and $M$ nodes" satisfying a given condition. Each classifier $\alpha_i(x)$ can be expressed within $FOC_2$, as the bounded quantifier can be rewritten using $\exists^{\geq N}$ and $\neg \exists^{\geq M+1}$. Each property $p_i$ corresponds to a classifier $\alpha_i$ with $i \in 1,2,3$. Summary statistics for the dataset are provided in Appendix G, Table 3.

Table 1: Accuracy difference (%) and model size (MB) of the ACR-GNN model before and after dynamic post-training quantization (PTQ) across FO-properties $p_1$, $p_2$, and $p_3$. Values are reported for three model depths (1, 2, and 3 layers) and three dataset splits (Train, Test 1, Test 2). Accuracy values represent the change after quantization (QINT8 – FP32). $p_1, p_2, p_3$ are FO-properties described in Appendix G.

| # | $p_1$ | | | $p_2$ | | | $p_3$ | | | Size (MB) |
|---|-------|--------|--------|-------|--------|--------|-------|--------|--------|-----------|
| | Train | Test 1 | Test 2 | Train | Test 1 | Test 2 | Train | Test 1 | Test 2 | |
| 1 | –0.452% | –0.760% | +0.522% | –0.127% | –0.183% | +8.891% | –0.299% | –0.648% | –0.693% | 0.034 |
| 2 | –0.001% | 0.000% | –0.043% | +0.083% | –0.125% | +0.144% | –0.178% | –0.226% | +0.018% | 0.068 |
| 3 | –0.036% | +0.062% | –0.494% | –0.161% | –0.143% | –0.342% | –0.015% | +0.280% | –0.346% | 0.103 |

Table 1 presents the difference in accuracy and model size between the quantized (QINT8[4]) and original (FP32) versions of the ACR-GNN model across three configurations (1, 2, and 3 layers). The

---

[4]The difference between INT8 and QINT8 lies in their implementation and is detailed in Appendix G

evaluation is conducted on three FO-properties ($p_1$, $p_2$, $p_3$) over three data splits: Train, Test1, and Test2. The table highlights how quantization affects accuracy at various depths. In most cases, the impact of quantization on accuracy is minor and bounded, with some configurations even showing positive differences. For instance, in the 2-layer configuration—the overall best performer—the accuracy loss remains within ±0.1 across all properties and splits, while yielding a model size reduction of 0.068 MB. The 1-layer model shows greater fluctuation: while $p_2$ on Test2 experiences a significant positive spike (+8.891), $p_3$ on Test2 drops by –0.693. This suggests sensitivity to quantization in shallow models, likely due to limited representational capacity. The results confirm that dynamic post-training quantization (PTQ) enables significant compression—up to 60% reduction in size—while maintaining acceptable levels of accuracy. Additional breakdowns, including baseline results and extended configurations, are provided in Appendix G.

Table 2: PPI benchmark. Accuracy (%) and size (MB) of the ACR-GNN with ReLU activation function before and after dynamic PTQ across different layer configurations.

| # | Original (FP32) | | | | Quantized (QINT8) | | | | Difference | | | |
|---|-------|-------|-------|-----------|-------|-------|-------|-----------|--------|-------|--------|-----------|
|   | Train | Val   | Test  | Size (MB) | Train | Val   | Test  | Size (MB) | Train  | Val   | Test   | Size (MB) |
| 1 | 54.7% | 43.1% | 39.5% | 0.922     | 55.0% | 50.8% | 50.2% | 0.242     | +0.3%  | +7.7% | +10.7% | 0.680     |
| 2 | 52.5% | 44.6% | 45.7% | 1.718     | 52.3% | 47.8% | 47.2% | 0.451     | -0.2%  | +3.2% | +1.5%  | 1.267     |
| 3 | 52.3% | 42.6% | 44.0% | 2.515     | 51.9% | 45.7% | 42.8% | 0.660     | -0.4%  | +3.1% | -1.2%  | 1.855     |

Table 2 shows the results of evaluating the ACR-GNN model on the Protein-Protein Interaction (PPI) benchmark before and after applying dynamic post-training quantization (PTQ). The evaluation covers three model configurations (1 to 3 layers) and reports performance in terms of accuracy (Train, Validation, and Test) and model size (in MB). Quantization results in substantial compression across all configurations. The model size decreases from 0.922 MB to 0.242 MB (a 73% reduction) for the 1-layer network, while the 2- and 3-layer models achieve reductions of 1.267 MB and 1.855 MB, respectively. Accuracy-wise, quantization leads to improvements in the Validation and Test sets for shallower networks. The 1-layer model gains +0.077 on validation and +0.107 on test accuracy, indicating potential for enhanced generalization. The 2-layer model shows minor improvements across all splits, with negligible loss in training accuracy. However, the 3-layer configuration reveals a slight drop in test accuracy (–0.012), suggesting increased sensitivity to quantization at greater depth. See Appendix G, Tables 16,17, and 18 for additional quantitative breakdowns.

## 8 Conclusion and Future Work

The central result is the NEXPTIME-complete of the logic $q\mathcal{L}$ in which both the computations of GNNs and modal properties can be expressed. It helps to understand the inherent complexity of verifying quantized GNNs. We also provide a prototype for verifying GNNs over a set of graphs with a bounded number of vertices. Finally some experiments confirmed that the quantization of ACR-GNNs is promising.

There are many directions to go. First, characterizing the modal flavor of $q\mathcal{L}$ for other activation functions than ReLU. New extensions of $q\mathcal{L}$ could be proposed to tackle other classes GNNs. Verification of neural networks is challenging and is currently tackled by the verification community [10]. So it will be for GNNs as well. Our verification tool with a bound on the number of vertices is still preliminary. One obvious path would be to improve the tool, to compare different approaches (bounded model checking vs. linear programming as in [18]) and apply it to real GNN verification scenarios. Designing a practical verification procedure in the general case (without any bound on the number of vertices) and overcoming the high computational complexity is an exciting challenge for future research towards the verification of GNNs.

**Limitations.** Section 4 and 5 reflect theoretical results. Some practical implementations of GNNs may not fully align with them. In particular, the order in the (non-associative) summation over values in $\mathbb{K}$ is fixed in formulas (3) and (4). It means that we suppose that the aggregation $agg(\vartheta)$ is computed in that order too (we sort the successors of a vertex according the values of $\vartheta$ and then perform the summation). The verification tool discussed in Section 6 remains a prototype, thus its application warrants careful consideration.

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

 # A Proofs of statements in the main text

**Lemma 3.** *Let $\varphi$ be a formula of modal $q\mathcal{L}$. The formulas $\varphi$ and $mod2expr(\varphi)$ are equivalent.*

*Proof.* We have to prove that for all $G, u$, we have $G, u \models \varphi$ iff $G, u \models mod2expr(\varphi)$. We proceed by induction on $\varphi$.

- The base case is obvious: $G, u \models \varphi$ iff $G, u \models mod2expr(\varphi)$ is $G, u \models \varphi$ iff $G, u \models mod2expr(\varphi)$.

- $G, u \models \neg\varphi$ iff $G, u \not\models \varphi$

  iff (by induction) $G, u \not\models mod2expr(\varphi)$

  iff (by writing $mod2expr(\varphi) = \vartheta \geq 0$) $G, u \models \vartheta \geq 0$

  iff $G, u \models \vartheta < 0$

  iff $G, u \models \vartheta \leq -1$ (because we suppose that $\vartheta$ takes its value in the integers

  iff $G, u \models \vartheta + 1 \leq 0$

  iff $G, u \models -\vartheta - 1 \geq 0$.

- $G, u \models (\varphi_1 \vee \varphi_2)$

  iff $G, u \models \varphi_1$ or $G, u \models \varphi_2$

  iff $G, u \models (\vartheta_1 \geq 0)$ or $G, u \models (\vartheta_2 \geq 0)$

  iff $G, u \models \vartheta_1 + ReLU(\vartheta_2 - \vartheta_1) \geq 0$

  Indeed, ($\Rightarrow$) if $G, u \models (\vartheta_1 \geq 0)$ then $G, u \models \vartheta_1 + ReLU(\vartheta_2 - \vartheta_1) \geq \vartheta_1 \geq 0$.

  If $G, u \models (\vartheta_2 \geq 0)$ and $G, u \models (\vartheta_1 < 0)$ then $G, u \models \vartheta_1 + ReLU(\vartheta_2 - \vartheta_1) = \vartheta_1 + \vartheta_2 - \vartheta_1 = \vartheta_2 \geq 0$.

  ($\Leftarrow$) Conversely, by contrapositive, if $G, u \models (\vartheta_2 < 0)$ and $G, u \models (\vartheta_1 < 0)$, then $G, u \models \vartheta_1 + ReLU(\vartheta_2 - \vartheta_1) = \vartheta_1 + \vartheta_2 - \vartheta_1 = \vartheta_2 < 0$ or $G, u \models \vartheta_1 + ReLU(\vartheta_2 - \vartheta_1) = \vartheta_1 + 0 = \vartheta_1 < 0$. In the two cases, $G, u \models \vartheta_1 + ReLU(\vartheta_2 - \vartheta_1) < 0$.

- $G, u \models \Diamond^{\geq k}\varphi$ iff the number of vertices $v$ that are successors of $u$ and with $G, v \models \varphi$ is greater than $k$

  iff the number of vertices $v$ that are successors of $u$ and with $G, v \models mod2expr(\varphi)$ is greater than $k$

  iff (written $\vartheta \geq 0$) iff the number of vertices $v$ that are successors of $u$ and with $G, v \models \vartheta \geq 0$ is greater than $k$

  iff the number of vertices $v$ that are successors of $u$ and with $G, v \models ReLU(\vartheta + 1) - ReLU(\vartheta) = 1$ is greater than $k$ (since we know by defining of modal $q\mathcal{L}$ that $\vartheta$ takes its value in integers)

  iff $G, u \models agg(ReLU(\vartheta + 1) - ReLU(\vartheta) \geq k$

  iff $G, u \models mod2expr(\Diamond^{\geq k}\varphi)$

- Other cases are similar.

$\square$

**Proposition 7.** *The number of Hintikka sets is bounded by $2^{n|\varphi|}$ where $|\varphi|$ is the size of $\varphi$, and $n$ is the bitwidth of $\mathbb{K}$.*

*Proof.* For each expression $\vartheta$, we choose a number in $\mathbb{K}$. There is $2^n$ different numbers. There are $|\varphi|$ number of expressions. So we get $(2^n)^{|\varphi|} = 2^{n|\varphi|}$ possible choices for a Hintikka set. $\square$

**Proposition 8.** *If bitwidth $n$ is in unary, and if $\mathbb{K}$ saturates, then satisfiability in $QFBAPA_{\mathbb{K}}$ is in NP.*

548 *Proof.* Here is a non-deterministic algorithm for the satisfiability problem in QFBAPA$_\mathbb{K}$.

549     1. Let $\chi$ be a QFBAPA$_\mathbb{K}$ formula.

550     2. For each set expression $B$ appearing in some $|B|$, guess a non-negative integer number $k_B$
551        in $\mathbb{K}$.

552     3. Let $\chi'$ be a (grounded) formula in which we replaced $|B|$ by $k_B$.

553     4. Check that $\chi'$ is true (can be done in poly-time since $\chi'$ is a grounded formula, it is a
554        Boolean formula on variable-free equations and inequations in $\mathbb{K}$).

555     5. If not we reject.

    6. We now build a standard QFBAPA formula $\delta = \bigwedge_B constraint(B)$ where:

$$constraint(B) = \begin{cases} |B| = k_B \text{ if } k_B < \infty_\mathbb{K} \\ |B| \geq limit \text{ if } k_B = +\infty_\mathbb{K} \end{cases}$$

556        where $limit$ is the maximum number that is considered as infinity in $\mathbb{K}$.

557     7. Run a non-deterministic poly-time algorithm for the QFBAPA satisfiability on $\delta$. Accepts if
558        it accepts. Otherwise reject.

559 The algorithm runs in poly-time. Guessing a number $n_B$ is in poly-time since it consists in guessing
560 $n$ bits ($n$ in unary). Step 4 is just doing the computations in $\mathbb{K}$. In Step 6, $\delta$ can be computed in
561 poly-time.

562 If $\chi$ is QFBAPA$_\mathbb{K}$ satisfiable, then there is a solution $\sigma$ such that $\sigma \models \chi$. At step 2, we guess
563 $n_B = |\sigma(B)|_\mathbb{K}$. The algorithm accepts the input.

564 Conversely, if the algorithm accepts its input, $\chi'$ is true for the chosen values $n_B$. $\delta$ is satisfiable. So
565 there is a solution $\sigma$ such that $\sigma \models \delta$. By the definition of $constraint$, $\sigma \models \chi$. $\qquad\square$

566 **Remark 15.** *If the number $n$ of bits to represent $\mathbb{K}$ is given in unary and if $\mathbb{K}$ is "modulo",*
567 *then the satisfiability problem in QFBAPA$_\mathbb{K}$ is also in NP. The proof is similar except than now*
568 $constraint(B) = (|B| = k_B + Ld_B)$ *where $d_B$ is a new variable.*

569 **Proposition 9.** $tr(\varphi)$ *is computable in exponential-time in $|\varphi|$ and $n$.*

570 *Proof.* In order to create $tr(\varphi)$, we write an algorithm where each big conjunction, big disjunction,
571 big union and big sum is replaced by a loop. For instance, $\bigwedge_{H \neq H'}$ is replaced by two inner loops
572 over Hintikka sets. Note that we create check whether a candidate $H$ is a Hintikka set in exponential
573 time in $n$ since Point 4 can be checked in exponential time in $n$ (thanks to our loose assumption on
574 the computability of $[[\alpha]]$ in exponential time in $n$. There are $2^{n|\varphi|}$ many of them. In the same way,
575 $\bigwedge_{k \in \mathbb{K}}$ is a loop over $2^n$ values. There is a constant number of nested loops, each of them iterating
576 over an exponential number (in $n$ and $|\varphi|$ of elements. QED. $\qquad\square$

577 **Proposition 10.** *Let $\varphi$ be a formula of $q\mathcal{L}$. $\varphi$ is satisfiable iff $tr(\varphi)$ is QFBAPA$_\mathbb{K}$ satisfiable.*

578 *Proof.* $\boxed{\Rightarrow}$ Let $G, u$ such that $G, u \models \varphi$. We set $\sigma(X_{\vartheta'=k}) := \{v \mid [[\vartheta']]_{G,v} = k\}$ and $\sigma(X_H) =$
579 $\{v \mid G, v \models H\}$ where $G, u \models H$ means that for all $\vartheta' = k \in H$, we have $[[\vartheta']]_{G,v} = k$. For all
580 Hintikka sets $H$ such that there is $v$ such that $G, v \models H$, we set: $\sigma(S_H) := \{w \mid vEw\}$.

581 We check that $\sigma \models tr(\varphi)$. First, $\sigma$ satisfies Formulas 1 and 2 by definition of $\sigma$. Now, $\sigma$ also satisfies
582 Formula 3. Indeed, if $agg(\vartheta') = k \in H$, then if there is no $H$-vertex in $G$ then the implication is
583 true. Otherwise, consider the $H$-vertex $v$. But, then by definition of $X_{agg(\vartheta')=k}$, $[[agg(\vartheta')]]_{G,v} = k$.
584 But then the semantics of $agg$ exactly corresponds to $\sum_{k' \in \mathbb{K}} |S_H \cap X_{\vartheta=k'}| \times k' = k$. Indeed, each
585 $S_H \cap X_{\vartheta=k'}$-successor contributes with $k'$. Thus, the contribution of successors where $\vartheta$ is $k'$ is
586 $|S_H \cap X_{\vartheta=k'}| \times k'$.

587 Formula 4 is also satisfied by $\sigma$. Actually, let $k$ such that $\sigma \models X_{agg_\forall(\vartheta)=k} = \mathcal{U}$. This means that
588 the value of $agg_\forall(\vartheta)$ (which does not depend on a specific vertex $u$ but only on $G$) is $k$. The sum
589 $\sum_{k' \in \mathbb{K}} |X_{\vartheta=k'}| \times k' = k$ is the semantics of $agg_\forall(\vartheta) = k$.

590 Finally, as $G, u \models \varphi$, and $\varphi$ is of the form $\vartheta \geq k$, there is $k' \geq k$ such that $[[\vartheta]]_{G,u} = k'$. So
591 $X_{\vartheta = k'} \neq \emptyset$.

592 $\boxed{\Leftarrow}$ Conversely, consider a solution $\sigma$ of $tr(\varphi)$. We construct a graph $G = (V, E)$ as follows.

$$V := \sigma(\mathcal{U})$$
$$E := \{(u, v) \mid \text{for some } H, u \in \sigma(X_H) \text{ and } v \in \sigma(S_H)\}$$
$$\ell(v)_i := k \text{ where } v \in X_{x_i = k}$$

593 i.e. the set of vertices is the universe, and we add an edge between any $H$-vertex $u$ and a vertex
594 $v \in \sigma(S_H)$, and the labeling for features is directly given $X_{x_i = k}$. Note that the labeling is well-
595 defined because of formulas 1 and 2.

596 As $\sigma \models |X_\varphi| \geq 1$, there exists $u \in \sigma(X_\varphi)$. Let us prove that $G, u \models \varphi$. By induction on $\vartheta'$, we
597 prove that $u \in X_{\vartheta' = k}$ implies $[[\vartheta']]_{G,u} = k$. The base case is obtained via the definition of $\ell$. Cases
598 for $+$, $\times$ and $\alpha$ are obtained because each vertices is in some $\sigma(X_H)$ for some $H$. As the definition of
599 Hintikka set takes care of the semantics of $+$, $\times$ and $\alpha$, we have $[[\vartheta_1 + \vartheta_2]]_{G,u} = [[\vartheta_1]]_{G,u} + [[\vartheta_2]]_{G,u}$,
600 etc.

601 $[[agg(\vartheta)]]_{G,u} = \Sigma_{v|uEv}[[\vartheta]]_{G,v}$ and $[[agg_\forall(\vartheta)]]_{G,u} = \Sigma_{v \in V}[[\vartheta]]_{G,v}$ hold because of $\sigma$ satisfies
602 respectively formula 3 and 4. $\qquad\square$

603 **Theorem 12.** *The satisfiability problem in $q\mathcal{L}$ is NEXPTIME-hard, and so is* VT3. VT1 *and* VT2 *are*
604 *coNEXPTIME-hard.*

605 *Proof.* We reduce the NEXPTIME-hard problem of deciding whether a domino system $\mathcal{D} =$
606 $(D, V, H)$, given an initial condition $w_0 \ldots w_{n-1} \in D^n$, can tile an exponential torus [36]. In
607 the domino system, $D$ is the set of tile types, and $V$ and $H$ respectively are the respectively vertical
608 and horizontal color compatibility relations. We are going to write a set of modal $q\mathcal{L}$ formulas that
609 characterize the torus $\mathbb{Z}^{2n+1} \times \mathbb{Z}^{2n+1}$ and the domino system. We use $2n + 2$ features. We use
610 $x_0, \ldots x_{n-1}$, and $x'_0, \ldots, x'_{n-1}$, to hold the (binary-encoded) coordinates of vertices in the torus. We
611 use the feature $x_N$ to denote a vertex 'on the way north' (when $x_N = 1$) and $x_E$ to denote a vertex
612 'on the way east' (when $x_E = 1$), with abbreviations $\varphi_N := x_N = 1$, and $\varphi_E := x_E = 1$. See
613 Figure 2.

614 For every $n \in \mathbb{N}$, we define the following set of formulas. $T_n =$

$$\begin{aligned}
\{ \quad &\Box_g(x_N = 1 \vee x_N = 0) &,& \quad \Box_g(x_E = 1 \vee x_E = 0), \\
&\Box_g(\bigwedge_{k=0}^{n-1}(x_i = 1 \vee x_i = 0)) &,& \quad \Box_g(\bigwedge_{k=0}^{n-1}(x'_i = 1 \vee x'_i = 0)), \\
&\Box_g(\neg(x_N = 1 \wedge x_E = 1)) &,& \quad \Box_g(\neg(\varphi_N \vee \varphi_E) \to agg(1) = 2), \\
&\Box_g(\neg(\varphi_N \vee \varphi_E) \to (agg(x_N) = 1)) &,& \quad \Box_g(\neg(\varphi_N \vee \varphi_E) \to (agg(x_E) = 1)), \\
&\Box_g(\varphi_N \to agg(1) = 1) &,& \quad \Box_g(\varphi_E = 1 \to agg(1) = 1), \\
&\Diamond_g^{=1}\varphi_{(0,0)} &,& \quad \Diamond_g^{=1}\varphi_{(2^n-1, 2^n-1)}, \\
&\Box_g(\neg(\varphi_N \vee \varphi_E) \to \varphi_{east}) &,& \quad \Box_g(\neg(\varphi_N \vee \varphi_E) \to \varphi_{north}), \\
&\Diamond_g^{\leq 2^n \times 2^n} \neg(\varphi_N \vee \varphi_E), &\quad \Diamond_g^{\leq 2^n \times 2^n}\varphi_N,& \quad \Diamond_g^{\leq 2^n \times 2^n}\varphi_E \quad \}
\end{aligned}$$

615 where $\varphi_{(0,0)} := \bigwedge_{k=0}^{n-1} x_i = 0 \wedge \bigwedge_{k=0}^{n-1} x'_i = 0$, and $\varphi_{(2^n-1, 2^n-1)} := \bigwedge_{k=0}^{n-1} x_i = 1 \wedge \bigwedge_{k=0}^{n-1} x'_i = 1$
616 represent two nodes, namely those at coordinates $(0, 0)$ and $(2^n - 1, 2^n - 1)$. The formulas $\varphi_{north}$ and
617 $\varphi_{east}$ enforce constraints on the coordinates of states, such that going north increases the coordinate
618 encoding using the $x_i$ features by one, leaving the $x'_i$ features unchanged, and going east increases
619 coordinate encoding using the $x'_i$ features by one, leaving the $x_i$ features unchanged. For every

formula $\varphi$, $\forall east.\varphi$ stands for $\square(\varphi_E \to \square\varphi)$ and $\forall north.\varphi$ stands for $\square(\varphi_N \to \square\varphi)$.

$$\varphi_{north} := \bigwedge_{k=0}^{n-1} (\bigwedge_{j=0}^{k-1} (x_j = 1)) \to (((x_k = 1) \to \forall north.(x_k = 0)) \wedge ((x_k = 0) \to \forall north.(x_k = 1)))\wedge$$

$$\bigwedge_{k=0}^{n-1} (\bigvee_{j=0}^{k-1} (x_j = 0)) \to (((x_k = 1) \to \forall north.(x_k = 1)) \wedge ((x_k = 0) \to \forall north.(x_k = 0)))\wedge$$

$$\bigwedge_{k=0}^{n-1} (((x_k' = 1) \to \forall north.(x_k' = 1)) \wedge ((x_k' = 0) \to \forall north.(x_k' = 0)))$$

$$\varphi_{east} := \bigwedge_{k=0}^{n-1} (\bigwedge_{j=0}^{k-1} (x_j' = 1)) \to (((x_k' = 1) \to \forall east.(x_k' = 0)) \wedge ((x_k' = 0) \to \forall east.(x_k' = 1)))\wedge$$

$$\bigwedge_{k=0}^{n-1} (\bigvee_{j=0}^{k-1} (x_j' = 0)) \to (((x_k' = 1) \to \forall east.(x_k' = 1)) \wedge ((x_k' = 0) \to \forall east.(x_k' = 0)))\wedge$$

$$\bigwedge_{k=0}^{n-1} (((x_k = 1) \to \forall east.(x_k = 1)) \wedge ((x_k = 0) \to \forall east.(x_k = 0)))$$

The problem of deciding whether a domino system $\mathcal{D} = (D, V, H)$, given an initial condition $w_0 \ldots w_{n-1} \in D^n$, can tile a torus of exponential size can be reduced to the problem satisfiability in $q\mathcal{L}$, checking the satisfiability of the set of formulas $T(n, \mathcal{D}, w) = T_n \cup T_\mathcal{D} \cup T_w$, where $T_n$ is as above, $T_\mathcal{D}$ encodes the domino system, and $T_w$ encodes the initial condition as follows. We define

$$T_\mathcal{D} = \{ \quad \square_g(\bigwedge_{d\in D}(x_d = 1 \vee x_d = 0)),$$
$$\square_g(\neg(\varphi_N \vee \varphi_E) \to (\bigvee_{d\in D} \varphi_d)),$$
$$\square_g(\neg(\varphi_N \vee \varphi_E) \to (\bigwedge_{d\in D} \bigwedge_{d'\in D\setminus\{d\}} \neg(\varphi_d \wedge \varphi_{d'}))),$$
$$\square_g(\bigwedge_{d\in D}(\varphi_d \to (\forall east.\bigvee_{(d,d')\in H} \varphi_{d'}))),$$
$$\square_g(\bigwedge_{d\in D}(\varphi_d \to (\forall north.\bigvee_{(d,d')\in V} \varphi_{d'}))) \quad \}$$

where for every $d \in D$, there is a feature $x_d$ and $\varphi_d := x_d = 1$. Finally, we define

$$T_w = \{ \quad \square_g(\varphi_{(0,0)} \to \varphi_{w_0}), \ldots, \square_g(\varphi_{(n-1,0)} \to \varphi_{w_{n-1}}) \quad \}$$

The size of $T(n, \mathcal{D}, w)$ is polynomial in the size of the tiling problem instance, that is in $|D| + |H| + |V| + n$. The rest of the proof is analogous to the proof of [36, Corollary 3.9]. The NEXPTIME-hardness of $q\mathcal{L}$ follows from Lemma 3 and [36, Corollary 3.3] stating the NEXPTIME-hardness of deciding whether a domino system with initial condition can tile a torus of exponential size.

For the complexity of ACR-GNN verification tasks, we observe the following.

1. We reduce the satisfiability problem in (modal) $q\mathcal{L}$ (restricted to graded modal logic + graded universal modality, because it is sufficient to encode the tiling problem) to VT3 in poly-time as follows. Let $\varphi$ be a $q\mathcal{L}$. We build in poly-time an ACR-GNN $\mathcal{A}$ that recognizes all pointed graphs. We have $\varphi$ is satisfiable iff $[[\varphi]] \cap [[\mathcal{A}]] \neq \emptyset$ So VT3 is NEXPTIME-hard.

2. The validity problem of $q\mathcal{L}$ (dual problem of the satisfiability problem, i.e., given a formula $\varphi$, is $\varphi$ true in all pointed graphs $G, u$?) is coNEXPTIME-hard. We reduce the validity problem of $q\mathcal{L}$ to VT2. Let $\varphi$ be a $q\mathcal{L}$ formula. We construct an ACR-GNN $\mathcal{A}$ that accepts all pointed graphs. We have $\varphi$ is valid iff $[[\mathcal{A}]] \subseteq [[\varphi]]$. So VT2 is coNEXPTIME-hard.

3. We reduce the validity problem of $q\mathcal{L}$ to VT1. Let $\psi$ be a $q\mathcal{L}$ formula. (again in graded modal logic + graded global modalities). So by [4], We construct in poly-time an ACR-GNN $\mathcal{A}$ that is equivalent to $\psi$ (by [4]). We have $\psi$ is valid iff $[[\top]] \subseteq [[\mathcal{A}]]$. So VT1 is coNEXPTIME-hard.

$\square$

**Theorem 14.** *The satisfiability problems with bounded number of vertices are NP-complete.*

 *Proof.* NP upper bound is obtained by guessing a graph with at most $N$ vertices and then check that $\varphi$
holds. The obtained algorithm is non-deterministic, runs in poly-time and decides the satisfiability
problem with bounded number of vertices. NP-hardness already holds for $agg$-free formulas by
reduction from SAT for propositional logic (the reduction is $mod2expr$, see Lemma 3). $\qquad\square$

# B   Checking distributivity

We provide C source code for checking distributivity. The reader may run the model checker ESBMC
on it to see whether distributivity holds or not.

# C   Extension of logic $K^\sharp$ and ACR-GNNs over $\mathbb{Z}$

A *(labeled directed) graph* $G$ is a tuple $(V, E, \ell)$ such that $V$ is a finite set of vertices, $E \subseteq V \times V$ a
set of directed edges and $\ell$ is a mapping from $V$ to a valuation over a set of atomic propositions. We
write $\ell(u)(p) = 1$ when atomic proposition $p$ is true in $u$, and $\ell(u)(p) = 0$ otherwise. Given a graph
$G$ and vertex $u \in V$, we call $(G, u)$ a *pointed graph*.

## C.1   Logic

Consider a countable set $Ap$ of propositions. We define the language of logic $K^{\sharp,\sharp_g}$ as the set of
formulas generated by the following BNF:

$$\varphi ::= p \mid \neg\varphi \mid \varphi \vee \varphi \mid \xi \geq 0$$
$$\xi ::= c \mid \mathbb{1}\varphi \mid \sharp\varphi \mid \sharp_g\varphi \mid \xi + \xi \mid c \times \xi$$

where $p$ ranges over $Ap$, and $c$ ranges over $\mathbb{Z}$. We assume that all formulas $\varphi$ are represented as
directed acyclic graph (DAG) and refer by *the size of $\varphi$* to the size of its DAG representation.

Atomic formulas are propositions $p$, inequalities and equalities of linear expressions. We consider
linear expressions over $\mathbb{1}\varphi$ and $\sharp\varphi$ and $\sharp_g\varphi$. The number $\mathbb{1}\varphi$ is equal to 1 if $\varphi$ holds in the current
world and equal 0 otherwise. The number $\sharp\varphi$ is the number of successors in which $\varphi$ hold. The
number $\sharp_g\varphi$ is the number of worlds in the model in which $\varphi$ hold. The language seems strict but we
write $\xi_1 \leq \xi_2$ for $\xi_2 - \xi_1 \geq 0$, $\xi = 0$ for $(\xi \geq 0) \wedge (-\xi \geq 0)$, etc.

As in modal logic, a formula $\varphi$ is evaluated in a pointed graph $(G, u)$ (also known as pointed Kripke
model). We define the truth conditions $(G, u) \models \varphi$ ($\varphi$ is true in $u$) by

$$
\begin{aligned}
(G, u) &\models p && \text{if} && \ell(u)(p) = 1, \\
(G, u) &\models \neg\varphi && \text{if} && \text{it is not the case that } (G, u) \models \varphi, \\
(G, u) &\models \varphi \wedge \psi && \text{if} && (G, u) \models \varphi \text{ and } (G, u) \models \psi, \\
(G, u) &\models \xi \geq 0 && \text{if} && [[\xi]]_{G,u} \geq 0,
\end{aligned}
$$

and the semantics $[[\xi]]_{G,u}$ (the value of $\xi$ in $u$) of an expression $\xi$ by mutual induction on $\varphi$ and $\xi$ as
follows.

$$
\begin{aligned}
[[c]]_{G,u} &= c, \\
[[\xi_1 + \xi_2]]_{G,u} &= [[\xi_1]]_{G,u} + [[\xi_2]]_{G,u}, \\
[[c \times \xi]]_{G,u} &= c \times [[\xi]]_{G,u}, \\
[[\mathbb{1}\varphi]]_{G,u} &= \begin{cases} 1 & \text{if } (G, u) \models \varphi \\ 0 & \text{otherwise,} \end{cases} \\
[[\sharp\varphi]]_{G,u} &= |\{v \in V \mid (u, v) \in E \text{ and } (G, v) \models \varphi\}| \\
[[\sharp_g\varphi]]_{G,u} &= |\{v \in V \mid (G, v) \models \varphi\}|.
\end{aligned}
$$

A local modality $\Box\varphi$ can be defined as $\Box\varphi := (-1) \times \sharp(\neg\varphi) \geq 0$. That is, to say that $\varphi$ holds
in all successors, we say that the number of successors in which $\neg\varphi$ holds is zero. Similarly, a
global/universal modality can be defined as $\Box_g\varphi := (-1) \times \sharp_g(\neg\varphi) \geq 0$.

## C.2   Aggregate-Combine Graph Neural Networks

In this section, we consider a detailed definition of quantized (global) Aggregate-Combine GNNs
(ACR-GNN) [4], also called message passing neural networks [16]. We stick to the former term.

A *(global) ACR-GNN layer* $\mathcal{L} = (comb, agg, agg_g)$ is a tuple where $comb : \mathbb{R}^{2m} \to \mathbb{R}^n$ is a so-called *combination function*, $agg$ is a so-called *local aggregation function*, mapping multisets of vectors from $\mathbb{R}^m$ to a single vector from $\mathbb{R}^n$, $agg_g$ is a so-called *global aggregation function*, also mapping multisets of vectors from $\mathbb{R}^m$ to a single vector from $\mathbb{R}^n$. We call $m$ the *input dimension* of layer $\mathcal{L}$ and $n$ the *output dimension* of layer $\mathcal{L}$. Then, a *(global) ACR-GNN* is a tuple $(\mathcal{L}^{(1)}, \ldots, \mathcal{L}^{(L)}, cls)$ where $\mathcal{L}^{(1)}, \ldots, \mathcal{L}^{(L)}$ are $L$ ACR-GNN layers and $cls : \mathbb{R}^m \to \{0, 1\}$ is a *classification function*. We assume that all GNNs are well-formed in the sense that output dimension of layer $\mathcal{L}^{(i)}$ matches input dimension of layer $\mathcal{L}^{(i+1)}$ as well as output dimension of $\mathcal{L}^{(L)}$ matches input dimension of $cls$.

Let $G = (V, E)$ be a graph with atomic propositions $p_1, \ldots, p_k$ and $\mathcal{A} = (\mathcal{L}^{(1)}, \ldots, \mathcal{L}^{(L)}, cls)$ an ACR-GNN. We define $x_0 : V \to \{0, 1\}^k$, called the *initial state of* $G$, as $x_0(u) := (\ell(u)(p_1), \ldots, \ell(u)(p_k))$ for all $u \in V$. Then, the $i$-th layer of $\mathcal{A}$ computes an updated state of $G$ by

$$x_i(u) := comb(x_{i-1}(u), agg(\{\{x_{i-1}(v) \mid uv \in E\}\}), agg_g(\{\{x_{i-1}(v) \mid v \in V\}\}))$$

where $agg$, $agg_g$, and $comb$ are respectively the local aggregation, global aggregation and combination function of the $i$-th layer. Let $(G, u)$ be a pointed graph. We write $\mathcal{A}(G, u)$ to denote the application of $\mathcal{A}$ to $(G, u)$, which is formally defined as $\mathcal{A}(G, u) = cls(x_L(u))$ where $x_L$ is the state of $G$ computed by $\mathcal{A}$ after layer $L$. Informally, this corresponds to a binary classification of node $u$.

In this work, we exclusively consider the following form of ACR-GNN $\mathcal{A}$: all local and global aggregation functions are given by the sum of all vectors in the input multiset, all combination functions are given by $comb(x, y, z) = \vec{\sigma}(xC + yA_1 + zA_2 + b)$ where $\vec{\sigma}(x)$ is the componentwise application of the *truncated ReLU* $\sigma(x) = max(0, min(1, x))$, with matrices $C$, $A_1$ and $A_2$ and vector $b$ of $\mathbb{K}$ parameters, and where the classification function is $cls(x) = \sum_i a_i x_i \geq 1$, where $a_i$ are from $\mathbb{K}$ as well.

We note $[[\mathcal{A}]]$ the set of pointed graphs $(G, u)$ such that $\mathcal{A}(G, u) = 1$. An ACR-GNN $\mathcal{A}$ is satisfiable if $[[\mathcal{A}]]$ is non-empty. The *satisfiability problem* for ACR-GNNs is: Given a ACR-GNN $\mathcal{A}$, decide whether $\mathcal{A}$ is satisfiable.

# D    Capturing GNNs with $K^{\sharp, \sharp_g}$

In this section, we demonstrate that the expressive power of (global) ACR-GNNs, as defined in Section C.2 and $K^{\sharp, \sharp_g}$, is equivalent. Informally, this means that for every formula $\varphi$ of $K^{\sharp, \sharp_g}$, there exists an ACR-GNNs $\mathcal{A}$ that expresses the same query, and vice-versa. To achieve this, we define a translation of one into the other and substantiate that this translation is efficient. This enables ways to employ $K^{\sharp, \sharp_g}$ for reasoning about ACR-GNN.

We begin by showing that global ACR-GNNs are at least as expressive as $K^{\sharp, \sharp_g}$. We remark that the arguments are similar to the proof of Theorem 1 in [24].

**Theorem 16.** *Let $\varphi \in K^{\sharp, \sharp_g}$ be a formula. There is $\mathcal{A}_\varphi$ such that for all pointed graphs $(G, u)$ we have $(G, u) \models \varphi$ if and only if $\mathcal{A}_\varphi(G, u) = 1$. Furthermore, $\mathcal{A}_\varphi$ can be built in polynomial time regarding the size of $\varphi$.*

*Proof sketch.* We construct a GNN $\mathcal{A}_\varphi$ that evaluates the semantics of a given $K^{\sharp, \sharp_g}$ formula $\varphi$ for some given pointed graph $(G, v)$. The network consists of $n$ layers, one for each of the $n$ subformulas $\varphi_i$ of $\varphi$, ordered so that the subformulas are evaluated based on subformula inclusion. The first layer evaluates atomic propositions, and each subsequent messages passing layer $l_i$ uses a fixed combination and fixed aggregation function to evaluate the semantics of $\varphi_i$.

The correctness follows by induction on the layers: the $i$-th layer correctly evaluates $\varphi_i$ at each vertex of $G$, assuming all its subformulas are correctly evaluated in previous layers. Finally, the classifying function $cls$ checks whether the $n$-th dimension of the vector after layer $l_n$, corresponding to the semantics of $\varphi_n$ for the respective vertex $v$, indicates that $\varphi_n = \varphi$ is satisfied by $(G, v)$. The network size is polynomial in the size of $\varphi$ due to the fact that the total number of layers and their width is polynomially bounded by the number of subformulas of $\varphi$. A full formal proof is given in Appendix F. $\qquad\square$

**Theorem 17.** *Let $\mathcal{A}$ be a GNN. We can compute in polynomial time wrt. $|\mathcal{A}|$ a $K^{\sharp,\sharp_g}$-formula $\varphi_{\mathcal{A}}$, represented as a DAG, such that $[[\mathcal{A}]] = [[\varphi_{\mathcal{A}}]]$.*

*Proof sketch.* We construct a $K^{\sharp,\sharp_g}$-formula $\varphi_{\mathcal{A}}$ that simulates the computation of a given GNN $\mathcal{A}$. For each layer $l_i$ of the GNN, we define a set of formulas $\varphi_{i,j}$, one per output dimension, that encode the corresponding node features using linear threshold expressions over the formulas from the previous layer. At the base, the input features are the atomic propositions $p_1, \ldots, p_{m_1}$.

Each formula $\varphi_{i,j}$ mirrors the computation of the GNN layer, including combination, local aggregation, and global aggregation. The final classification formula $\varphi_{\mathcal{A}}$ encodes the output of the linear classifier on the top layer features. Correctness follows from the fact that all intermediate node features remain Boolean under message passing layers with integer parameters and truncated ReLU activations. This allows expressing each output as a Boolean formula over the input propositions. The construction is efficient: by reusing shared subformulas via a DAG representation, the total size remains polynomial in the size of $\mathcal{A}$. □

# E Complexity of the satisfiability of $K^{\sharp,\sharp_g}$ and its implications for ACR-GNN verification

In this section, we establish the complexity of reasoning with $K^{\sharp,\sharp_g}$.

Instrumentally, we first show that every $K^{\sharp,\sharp_g}$ formula can be translated into a $K^{\sharp,\sharp_g}$ formula that is equi-satisfiable, and has a tree representation of size at most polynomial in the size of the original formula. An analogous result was obtained in [24] for $K^{\sharp}$. It can be shown using a technique reminiscent of [37] and consisting of factorizing subformulas that are reused in the DAG by introducing a fresh proposition that is made equivalent. Instead of reusing a 'possibly large' subformula, a formula then reuses the equivalent 'small' atomic proposition.

**Lemma 18.** *The satisfiability problem of $K^{\sharp,\sharp_g}$ reduces to the satisfiability of $K^{\sharp,\sharp_g}$ with tree formulas in polynomial time.*

*Proof.* Let $\varphi$ be a $K^{\sharp,\sharp_g}$ formula represented as a DAG. For every subformula $\psi$ (i.e., for every node in the DAG representation of $\varphi$), we introduce a fresh atomic proposition $p_\psi$. We can capture the meaning of these new atomic propositions with the formula $\Phi := \bigwedge_{\psi \text{ node in the DAG}} sem(\psi)$ where:

$$sem(\psi \vee \chi) := p_{\psi \vee \chi} \leftrightarrow (p_\psi \vee p_\chi)$$
$$sem(\neg \psi) := p_{\neg \psi} \leftrightarrow \neg p_\psi$$
$$sem(\xi \geq 0) := p_{\xi \geq 0} \leftrightarrow \xi' \geq 0$$

$$(c)' := c \qquad (\xi_1 + \xi_2)' := \xi_1' + \xi_2' \qquad (c \times \xi)' := c \times \xi'$$
$$(\mathbb{1}\psi)' := \mathbb{1}p_\psi \qquad (\sharp\psi)' := \sharp p_\psi \qquad (\sharp_g\psi)' := \sharp_g p_\psi$$

Now, define $\varphi_t := p_\varphi \wedge \Box_g \Phi$, where $\Box_g \Phi := (-1) \times \sharp_g(\neg\Phi) \geq 0$, enforcing the truth of $\Phi$ in every vertex. The size of its tree representation is polynomial in the size of $\varphi$. Moreover, $\varphi_t$ is satisfiable iff $\varphi$ is satisfiable.

□

**Theorem 19.** *$K^{\sharp,\sharp_g}_{tree}$-satisfiability problem is NEXPTIME-complete.*

*Proof.* For membership, we translate the problem into the NEXPTIME-complete problem of concept description satisfiability in the Description Logics with Global and Local Cardinality Constraints [2], noted $\mathcal{ALCSCC}^{++}$. The Description Logic $\mathcal{ALCSCC}^{++}$ uses the Boolean Algebra with Presburger Arithmetic [20], noted QFBAPA, to formalize cardinality constraints. See Section H for a presentation of $\mathcal{ALCSCC}^{++}$ and QFBAPA.

Let $\varphi_0$ be a $K^{\sharp,\sharp_g}$ formula.

For every proposition $p$ occurring in $\varphi_0$, let $A_p$ be an $\mathcal{ALCSCC}^{++}$ concept name. Let $R$ be an $\mathcal{ALCSCC}^{++}$ role name. For every occurrence of $\mathbb{1}\varphi$ in $\varphi_0$, let $ZOO_\varphi$ be an $\mathcal{ALCSCC}^{++}$ role name. $ZOO$-roles stand for 'zero or one'. The rationale for introducing $ZOO$-roles is to be able to capture

765 the value of $\mathbb{1}\varphi$ in $\mathcal{ALCSCC}^{++}$ making it equal to the number of successors of the role $ZOO_\varphi$ which
766 can then be used in QFBAPA constraints. A similar trick was used, in another context, in [14]. Here,
767 we enforce this with the QFBAPA constraint

$$\chi_0 = \bigwedge_{\mathbb{1}\varphi \in \varphi_0} \left( (|ZOO_\varphi| = 0 \vee |ZOO_\varphi| = 1) \wedge \overline{\tau}(\varphi) = \mathsf{sat}(|ZOO_\varphi| = 1) \right)$$

768 which states that $ZOO_\varphi$ has zero or one successor, and has one successor exactly when (the translation
769 of) $\varphi$ is true. The concept descriptions $\overline{\tau}(\varphi)$ and arithmetic expressions $\overline{\tau}(\xi)$ are defined inductively
770 as follows:

$$
\begin{array}{rcl}
\overline{\tau}(p) & = & A_p \\
\overline{\tau}(\neg\varphi) & = & \neg\overline{\tau}(\varphi) \\
\overline{\tau}(\varphi \vee \psi) & = & \overline{\tau}(\varphi) \sqcup \overline{\tau}(\psi) \\
\overline{\tau}(\xi \geq 0) & = & \mathsf{sat}(-1 < \overline{\tau}(\xi)) \\
\overline{\tau}(c) & = & c \\
\overline{\tau}(\xi_1 + \xi_2) & = & \overline{\tau}(\xi_1) + \overline{\tau}(\xi_2) \\
\overline{\tau}(c \times \xi) & = & \overline{\tau}(c \cdot \xi) \\
\overline{\tau}(\sharp\varphi) & = & |R \cap \overline{\tau}(\varphi)| \\
\overline{\tau}(\mathbb{1}\varphi) & = & |ZOO_\varphi| \\
\overline{\tau}(\sharp_g\varphi) & = & |\overline{\tau}(\varphi)|
\end{array}
$$

771 Finally, we define the $\mathcal{ALCSCC}^{++}$ concept description $C_{\varphi_0} = \overline{\tau}(\varphi_0) \sqcap \mathsf{sat}(\chi_0)$.

772 **Claim 20.** *The concept description $C_{\varphi_0}$ is $\mathcal{ALCSCC}^{++}$-satisfiable iff the formula $\varphi_0$ is $K^{\sharp,\sharp_g}$-*
773 *satisfiable. Moreover, the concept description $C_{\varphi_0}$ has size polynomial in the size of $\varphi_0$.*

774 *Proof.* From right to left, suppose that $\varphi_0$ is $K^{\sharp,\sharp_g}$-satisfiable. It means that there is a pointed
775 graph $(G, u)$ where $G = (V, E)$ and $u \in V$, such that $(G, u) \models \varphi_0$. Let $I_0 = (\Delta^{I_0}, \cdot^{I_0})$ be
776 the $\mathcal{ALCSCC}^{++}$ interpretation over $N_C$ and $N_R$, such that $N_C = \{A_p \mid p \text{ a proposition in } \varphi_0\}$,
777 $N_R = \{R\} \cup \{ZOO_\varphi \mid \mathbb{1}\varphi \in \varphi_0\}$, $\Delta^{I_0} = V$, $A_p^{I_0} = \{v \mid v \in V, (G, v) \models p\}$ for every $p$ in $\varphi_0$,
778 $R^{I_0} = E$, $ZOO_\varphi^{I_0} = \{(v, v) \mid v \in V, (G, v) \models \varphi\}$ for every $\mathbb{1}\varphi$ in $\varphi_0$. We can show that $u \in C_{\varphi_0}^{I_0}$.
779 Basically $I^0$ is like $G$ with the addition of adequately looping $ZOO$-roles. An individual in $\Delta^{I_0}$ has
780 exactly one $ZOO_\varphi$-successor (itself), exactly when $\varphi$ is true, and no successor otherwise; $A_p$ is true
781 exactly where $p$ is true, and the role $R$ corresponds exactly to $E$.

782 From left to right, suppose that $C_{\varphi_0}$ is $\mathcal{ALCSCC}^{++}$-satisfiable. It means that there is an $\mathcal{ALCSCC}^{++}$
783 finite interpretation $I_0 = (\Delta^{I_0}, \cdot^{I_0})$ and an individual $d \in \Delta^{I_0}$ such that $d \in C_{\varphi_0}^{I_0}$. Let $G = (V, E)$ be
784 a graph such that $V = \Delta^{I_0}$, $E = R^{I_0}$, and $\ell(d)(p) = 1$ iff $d \in A_p^{I_0}$. We can show that $(G, d) \models \varphi_0$.

785 Since there are at most $|\varphi_0|$ subformulas in $\varphi_0$, the representation of $ZOO_\varphi$ for every subformula $\varphi$
786 of $\varphi_0$ can be done in size $\log_2(|\varphi_0|)$. For every formula $\varphi$, the size of the concept description $\overline{\tau}(\varphi)$ is
787 polynomial (at most $O(n \log(n))$). The overall size of $\overline{\tau}(\varphi_0)$ is polynomial in the size of $\varphi_0$, and so
788 is the size of $\mathsf{sat}(\xi_0)$ (at most $O(n^2(\log(n))^2)$). $\qquad\square$

789 The NEXPTIME-membership follows from Claim 20 and the fact that the concept satisfiability
790 problem in $\mathcal{ALCSCC}^{++}$ is in NEXPTIME (Theorem 25).

791 For the hardness, we reduce the problem of consistency of $\mathcal{ALCQ}$-$T_C$Boxes which is NEXPTIME-
792 hard [36, Corollary 3.9]. See Section I and Theorem 27 that slightly adapts Tobies' proof to show
793 that the problem is hard even with only one role.

794 We define the translation $\underline{\tau}$ from the set of $\mathcal{ALCQ}$ concept expressions and $\mathcal{ALCQ}$ cardinality
795 constraints, with only one role $R$.

$$
\begin{array}{rcl}
\underline{\tau}(A) & = & p_A \\
\underline{\tau}(\neg C) & = & \neg\underline{\tau}(C) \\
\underline{\tau}(C_1 \sqcup C_2) & = & \underline{\tau}(C_1) \vee \underline{\tau}(C_2) \\
\underline{\tau}(\geq n\, R.C) & = & \sharp\underline{\tau}(C) + (-1) \times n \geq 0 \\
\underline{\tau}(\geq n\, C) & = & \sharp_g\underline{\tau}(C) + (-1) \times n \geq 0 \\
\underline{\tau}(\leq n\, C) & = & (-1) \times \sharp_g\underline{\tau}(C) + n \geq 0
\end{array}
$$

It is routine to check the following claim.

**Claim 21.** *Let $TC$ be an $\mathcal{ALCQ}$-$T_C$Box. $TC$ is consistent iff $\bigwedge_{\chi \in TC} \underline{\tau}(\chi)$ is $K^{\sharp,\sharp_g}$-satisfiable.*

Moreover, the reduction is linear. Hardness thus follows from the NEXPTIME-hardness of consistency of $\mathcal{ALCQ}$-$T_C$Boxes. $\qquad\square$

Lemma 18 and Theorem 19 yield the following corollary.

**Corollary 22.** $K^{\sharp,\sharp_g}$-*satisfiability problem is NEXPTIME-complete.*

Furthermore, from Theorem 16 and Corollary 22, we obtain the complexity of reasoning with ACR-GNNs with truncated ReLU and integer weights.

**Corollary 23.** *Satisfiability of ACR-GNN with global readout and truncated ReLU is NEXPTIME-complete.*

The decidability of the problem is left open in [7] and in the recent long version [8] when the weights are rational numbers. The theorem answers it positively in the case of integer weights and pinpoints the computational complexity.

# F  Formal proofs

*Proof of Theorem 16.* Let $\varphi$ be a $K^{\sharp,\sharp_g}$ formula over the set of atomic propositions $p_1, \ldots, p_m$. Let $\varphi_1, \ldots, \varphi_n$ denote an enumeration of the subformulas of $\varphi$ such that $\varphi_i = p_i$ for $i \leq m$, $\varphi_n = \varphi$, and whenever $\varphi_i$ is a subformula of $\varphi_j$, it holds that $i \leq j$. Without loss of generality, we assume that all subformulas of the form $\xi \geq 0$ are written as

$$\sum_{j \in J} k_j \cdot \mathbb{1} \varphi_j + \sum_{j' \in J'} k_{j'} \cdot \sharp \varphi_{j'} + \sum_{j'' \in J''} k_{j''} \cdot \sharp_g \varphi_{j''} - c \geq 0,$$

for some index sets $J, J', J'' \subseteq \{1, \ldots, n\}$.

We construct the GNN $\mathcal{A}_\varphi$ in a layered manner. Note that $\mathcal{A}_\varphi$ is fully specified by defining the combination function $comb_i$, including its local and global aggregation, for each layer $l_i$ with $i \in \{1, \ldots, n\}$ and the final classification function $cls$. Each $comb_i$ produces output vectors of dimension $n$. The first layer $comb_1$ has input dimension $2m$ and is defined by $comb_1(x, y, z) = (x, 0, \ldots, 0)$, ensuring that the first $m$ dimensions correspond to the truth values of the atomic propositions $p_1, \ldots, p_m$, while the remaining entries are initialized to zero. Note that $comb_1$ is easily realized by an FNN with ReLU activations. For $i > 1$, the combination function $comb_i$ is defined as

$$comb_i(x, y, z) = \vec{\sigma}(xC + yA_1 + zA_2 + b),$$

where $C, A_1, A_2$ are $n \times n$ matrices corresponding to self, local (neighbor), and global aggregation respectively, and $b \in \mathbb{R}^n$ is a bias vector. The parameters are defined sparsely as follows:

- $C_{ii} = 1$ for all $i \leq m$ (preserving the atomic propositions),

- If $\varphi_i = \neg\varphi_j$, then $C_{ji} = -1$ and $b_i = 1$,

- If $\varphi_i = \varphi_j \vee \varphi_l$, then $C_{ji} = C_{li} = 1$, and

- If $\varphi_i = \sum_{j \in J} k_j \cdot 1_{\varphi_j} + \sum_{j' \in J'} k_{j'} \cdot \sharp\varphi_{j'} + \sum_{j'' \in J''} k_{j''} \cdot \sharp_g\varphi_{j''} - c \geq 0$, then

$$C_{ji} = k_j, \quad A_{1,j'i} = k_{j'}, \quad A_{2,j'i} = k_{j''}, \quad b_i = -c + 1.$$

Note that each $comb_i$ has the same functional form, differing only in the non-zero entries of its parameters. The classification function is defined by $cls(x) = x_n \geq 1$.

Let $l_i$ denote the $i$th layer of $\mathcal{A}_\varphi$, and fix a vertex $v$ in some input graph. We show, by induction on $i$, that the following invariant holds: for all $j \leq i$, $(x_i(v))_j = 1$ if and only if $v \models \varphi_j$, and $(x_i(v))_j = 0$ otherwise. Assume that $i = 1$. By construction, $x_1(v)$ contains the truth values of the atomic propositions $p_1, \ldots, p_m$ in its first $m$ coordinates. Thus, the statement holds at layer 1.

834 Next, assume the statement holds for layer $x_{i-1}$. Let $j < i$. By assumption, the semantics of $\varphi_j$ are
835 already correctly encoded in $x_{j-1}$ and preserved by $comb_i$ due to the fixed structure of $C$, $A_1$, $A_2$,
836 and $b$. Now consider $j = i$. The semantics of all subformulas of $\varphi_i$ are captured in $x_{i-1}$, either at
837 the current vertex or its neighbors. By the design of $comb_i$, which depends only on the values of
838 relevant subformulas, we conclude that $\varphi_i$ is correctly evaluated. This holds regardless of whether $\varphi_i$
839 is a negation, disjunction, or numeric threshold formula. Thus, the statement holds for all $i$, and in
840 particular for $x_n(v)$ and $\varphi_n = \varphi$. Finally, the classifier $cls$ evaluates whether $x_n(v)_n \geq 1$, which is
841 equivalent to $G, v \models \varphi$. The size claim is obvious given that $n$ depends polynomial on the size of $\varphi$.
842 We note that this assumes that the enumeration of subformulas of $\varphi$ does not contain duplicates. $\square$

843 *Proof of Theorem 17.* Let $\mathcal{A}$ be a GNN composed of layers $l_1, \ldots, l_k$, where each $comb_i$ has input
844 dimension $2m_i$, output dimension $n_i$, and parameters $C_i$, $A_{i,1}$, $A_{i,2}$, and $b_i$. The final classification
845 is defined via a linear threshold function $cls(x) = a_1 x_1 + \cdots + a_{n_k} x_{n_k} \geq 1$. We assume that the
846 dimensionalities match across layers, i.e. $m_i = n_{i-1}$ for all $i \geq 2$, so that the GNN is well-formed.

847 We construct a formula $\varphi_{\mathcal{A}}$ over the input propositions $p_1, \ldots, p_{m_1}$ inductively, mirroring the structure
848 of the GNN computation.

849 We begin with the first layer $l_1$. For each $j \in \{1, \ldots, n_1\}$, we define:

$$\varphi_{1,j} = \sum_{k=1}^{m_1} (C_1)_{kj} \cdot \mathbb{1} p_k + (A_{1,1})_{kj} \cdot \sharp p_k + (A_{1,2})_{kj} \cdot \sharp_g p_k + (b_1)_j \geq 1.$$

850 Now suppose that we have already constructed formulas $\varphi_{i-1,1}, \ldots, \varphi_{i-1,n_{i-1}}$ for some layer $i \geq 2$.
851 Then, for each output index $j \in \{1, \ldots, n_i\}$, we define:

$$\varphi_{i,j} = \sum_{k=1}^{m_i} (C_i)_{kj} \cdot \mathbb{1} \varphi_{i-1,k} + (A_{i,1})_{kj} \cdot \sharp \varphi_{i-1,k} + (A_{i,2})_{kj} \cdot \sharp_g \varphi_{i-1,k} + (b_i)_j \geq 1.$$

852 Once all layers have been encoded in this way, we define the final classification formula as

$$\varphi_{\mathcal{A}} = a_1 \mathbb{1} \varphi_{k,1} + \cdots + a_{n_k} \mathbb{1} \varphi_{k,n_k} \geq 1.$$

853 Let $G, v$ be a pointed graph. The correctness of our translation follows directly from the following
854 observations: all weights and biases in $\mathcal{A}$ are integers, and the input vectors $x_0(u)$ assigned to
855 nodes $u$ in $G$ are Boolean. Moreover, each layer applies a linear transformation followed by a
856 pointwise truncated ReLU, which preserves the Boolean nature of the node features. It follows that
857 the intermediate representations $x_i(v)$ remain in $\{0, 1\}^{n_i}$ for all $i$. Consequently, each such feature
858 vector can be expressed via a set of Boolean $K^{\sharp, \sharp_g}$-formulas as constructed above. Taken together,
859 this ensures that the overall formula $\varphi_{\mathcal{A}}$ faithfully simulates the GNN's computation.

860 It remains to argue that this construction can be carried out efficiently. Throughout, we represent
861 the (sub)formulas using a shared DAG structure, avoiding duplication of equivalent subterms. This
862 ensures that subformulas $\varphi_{i-1,k}$ can be reused without recomputation. For each layer, constructing all
863 $\varphi_{i,j}$ requires at most $n_i \cdot m_i$ steps, plus the same order of additional operations to account for global
864 aggregation terms. Since the number of layers, dimensions, and parameters are bounded by $|\mathcal{A}|$, and
865 each operation can be performed in constant or linear time, the total construction is polynomial in the
866 size of $\mathcal{A}$. $\square$

# G  Experimental data and further analyses

868 This study investigates the application of dynamic Post-Training Quantization (PTQ) to Aggregate-
869 Combined Readout Graph Neural Networks (ACR-GNNs). Implemented in PyTorch [1, 26], dynamic
870 PTQ transforms a pre-trained floating-point model into a quantized version without requiring retrain-
871 ing. In this approach, model weights are statically quantized to INT8, while activations remain in
872 floating-point format until they are dynamically quantized at compute time. This hybrid representation
873 enables efficient low-precision computation using INT8-based matrix operations, thereby reducing
874 memory footprint and improving inference speed. PyTorch's implementation applies per-tensor
875 quantization to weights and stores activations as floating-point values between operations to balance
876 precision and performance.

We adopt INT8 and QINT8 representations as the primary quantization format. According to theory, INT8 refers to 8-bit signed integers that can encode values in the range $[-128, 127]$. In contrast, QINT8, as defined in the PyTorch documentation [1, 27, 28], is a quantized tensor format that wraps INT8 values together with quantization metadata: a scale (defining the float value represented by one integer step) and a zero-point (the INT8 value corresponding to a floating-point zero). This additional information allows QINT8 tensors to approximate floating-point representations efficiently while enabling high-throughput inference.

To evaluate the practical impact of quantization, we conducted experiments on both synthetic and real datasets. The synthetic data setup was based on the benchmark introduced by [4]. Graphs were generated using the dense Erdös–Rényi model, a classical method for constructing random graphs, and each graph was initialized with five node colours encoded as one-hot feature vectors. The dataset is structured as follows, as shown in Table 3. The training set consists of 5000 graphs, each with 40 to 50 nodes and between 560 and 700 edges. The test set is divided into two subsets. The first subset comprises 500 graphs with the same structure as the training set, featuring 40 to 50 nodes and 560 to 700 edges. The second subset contains 500 larger graphs, with 51 to 69 nodes and between 714 and 960 edges. This design allows us to evaluate the model's generalization capability to unseen graph sizes.

Table 3: Dataset statistics summary.

| Classifier | Dataset | Node | | | Edge | | |
|---|---|---|---|---|---|---|---|
| | | Min | Max | Avg | Min | Max | Avg |
| $p_1$ | Train | 40 | 50 | 45 | 560 | 700 | 630 |
| | Test1 | 40 | 50 | 45 | 560 | 700 | 633 |
| | Test2 | 51 | 60 | 55 | 714 | 960 | 832 |
| $p_2$ | Train | 40 | 50 | 45 | 560 | 700 | 630 |
| | Test1 | 40 | 50 | 44 | 560 | 700 | 628 |
| | Test2 | 51 | 60 | 55 | 714 | 960 | 832 |
| $p_2$ | Train | 40 | 50 | 44 | 560 | 700 | 629 |
| | Test1 | 40 | 50 | 45 | 560 | 700 | 630 |
| | Test2 | 51 | 60 | 55 | 714 | 960 | 831 |

For this experiment, we used simple ACR-GNN models with the following specifications. We applied the *sum* function for both the aggregation and readout operations. The combination function was defined as: $comb(x, y, z) = \vec{\sigma}(xC + yA + zR + b)$, where $\vec{\sigma}$ denotes the activation function. Following the original work, we set the hidden dimension to 64, used a batch size of 128, and trained the model for 20 epochs using the Adam optimizer with default PyTorch parameters. We used two activation functions for the experimental part, ReLU and truncated ReLU. For implementation, we used PyTorch [1]: `nn.ReLU` and `nn.Hardtanh(0, 1)` in accordance.

We trained ACR-GNN on complex formulas $FOC_2$ for labeling. They are presented as a classifier $\alpha_i(x)$ that constructed as:

$$\alpha_0(x) := \text{Blue}(x), \alpha_{i+1}(x) := \exists^{[N,M]} y \, (\alpha_i(y) \wedge \neg E(x, y))$$

where $\exists^{[N,M]}$ stands for "there exist between $N$ and $M$ nodes". satisfying a given property.

Observe that each $\alpha_i(x)$ is in $FOC_2$, as $\exists^{[N,M]}$ can be expressed by combining $\exists^{\geq N}$ and $\neg \exists^{\geq M+1}$.

The data set has the following specifications: Erdös–Rényigraphs and is labeled according to $\alpha_1(x)$, $\alpha_2(x)$, and $\alpha_3(x)$:

- $\alpha_0(x) := \text{Blue}(x)$

- $p_1 : \alpha_1(x) := \exists^{[8,10]} y \, (\alpha_0(y) \wedge \neg E(x, y))$

- $p_2 : \alpha_2(x) := \exists^{[10,30]} y \, (\alpha_1(y) \wedge \neg E(x, y))$

- $p_3 : \alpha_3(x) := \exists^{[10,30]} y \, (\alpha_2(y) \wedge \neg E(x, y))$

In this section, we present experiments for two activation functions: ReLU and truncated ReLU (implemented via nn.Hardtanh(0,1)) to study the influence of the activation function on the model.

Experiments for the ACR-GNN were conducted with different numbers of hidden layers, ranging from 1 to 10. To measure the precision of the results, we use the strategy as [4]: accuracy is calculated as the total number of correctly classified nodes among all nodes in all graphs in the dataset.

Table 4: Accuracy of the ACR-GNN with ReLU according to the number of layers.

| Layer | $p_1$ | | | $p_2$ | | | $p_3$ | | |
|---|---|---|---|---|---|---|---|---|---|
| | Train | Test 1 | Test 2 | Train | Test 1 | Test 2 | Train | Test 1 | Test 2 |
| 1 | 96.9% | 96.4% | 74.8% | 69.8% | 71.0% | 56.7% | 69.1% | 68.8% | 75.4% |
| 2 | 100.0% | 100.0% | 99.5% | 83.7% | 84.5% | 75.3% | 76.6% | 76.8% | 77.0% |
| 3 | 97.6% | 97.3% | 87.2% | 83.6% | 84.2% | 75.1% | 76.7% | 76.4% | 66.9% |
| 4 | 68.6% | 68.4% | 67.3% | 83.5% | 84.0% | 76.1% | 77.7% | 76.3% | 46.6% |
| 5 | 68.5% | 68.3% | 67.0% | 83.5% | 83.9% | 77.6% | 78.2% | 76.8% | 34.1% |
| 6 | 68.5% | 68.4% | 66.1% | 83.6% | 84.1% | 79.6% | 77.6% | 75.8% | 34.8% |
| 7 | 68.5% | 68.5% | 67.3% | 83.5% | 83.8% | 80.5% | 77.1% | 77.7% | 49.4% |
| 8 | 68.5% | 68.4% | 65.8% | 83.4% | 83.8% | 73.2% | 76.7% | 75.7% | 75.1% |
| 9 | 68.5% | 68.3% | 66.7% | 83.0% | 83.4% | 79.1% | 77.3% | 76.9% | 48.0% |
| 10 | 68.6% | 68.3% | 65.5% | 83.1% | 83.7% | 77.3% | 76.4% | 75.6% | 37.4% |

Table 4 presents the accuracy of the ACR-GNN model with ReLU activation across three FO-properties ($p_1$, $p_2$, and $p_3$), evaluated on Train, Test1, and Test2 splits. For $p_1$, the model achieves high accuracy in the first three layers, peaking at 99.5% on Test2 at layer 2. From layer 4 and beyond, the accuracy on Test2 declines and stabilizes around 66–67%, suggesting a decreased performance in deeper models for this property. For $p_2$, initial accuracy is modest (e.g., 69.8% on Train and 56.7% on Test2 at layer 1), but improves rapidly with depth, surpassing 83% from layer 2 onward on Train and Test1. In particular, the accuracy of Test2 continues to improve with depth, reaching a peak at 80.5% in layer 7, indicating that $p_2$ benefits from deeper architectures. In contrast, $p_3$ exhibits less consistent behavior. Accuracy improves early, reaching 77.0% on Test2 at layer 2, but then drops sharply: Test2 accuracy drops to 46.6% at layer 4 and reaches a minimum of 34.1% at layer 5. Some recovery is observed at layers 7 and 8, yet performance remains unstable, with Test2 accuracy at 37.4% by layer 10. Overall, the results demonstrate that model depth significantly affects performance depending on the target property. While $p_2$ benefits from deeper configurations, both $p_1$ and $p_3$ achieve higher generalization performance in shallower networks, with deeper layers leading to overfitting or reduced representation quality on unseen data.

Table 5: Accuracy of the ACR-GNN with ReLU after dynamic PTQ according to the number of layers.

| Layer | $p_1$ | | | $p_2$ | | | $p_3$ | | |
|---|---|---|---|---|---|---|---|---|---|
| | Train | Test 1 | Test 2 | Train | Test 1 | Test 2 | Train | Test 1 | Test 2 |
| 1 | 96.5% | 95.7% | 75.3% | 69.7% | 70.8% | 65.6% | 68.8% | 68.2% | 74.7% |
| 2 | 100.0% | 100.0% | 99.4% | 83.8% | 84.4% | 75.5% | 76.4% | 76.6% | 77.0% |
| 3 | 97.6% | 97.4% | 86.7% | 83.5% | 84.1% | 74.7% | 76.7% | 76.7% | 66.5% |
| 4 | 68.6% | 68.5% | 66.9% | 83.3% | 84.2% | 76.2% | 77.6% | 76.1% | 44.6% |
| 5 | 68.5% | 68.2% | 67.2% | 83.4% | 84.0% | 77.8% | 78.3% | 76.6% | 33.4% |
| 6 | 68.6% | 68.4% | 66.2% | 83.5% | 83.9% | 80.3% | 77.4% | 75.6% | 35.8% |
| 7 | 68.5% | 68.4% | 67.1% | 83.3% | 83.6% | 80.6% | 77.1% | 77.6% | 48.7% |
| 8 | 68.5% | 68.3% | 65.8% | 83.3% | 83.7% | 73.2% | 76.7% | 75.5% | 74.6% |
| 9 | 68.5% | 68.3% | 66.6% | 83.0% | 83.6% | 78.9% | 77.1% | 76.2% | 44.3% |
| 10 | 68.5% | 68.2% | 58.1% | 83.0% | 83.7% | 77.5% | 76.3% | 75.4% | 36.6% |

Table 5 presents the node-level accuracy of the ACR-GNN model with ReLU activation after applying dynamic post-training quantization (PTQ). Results are reported for three FO-properties ($p_1$, $p_2$,

and $p_3$), evaluated across the Train, Test1, and Test2 splits. For $p_1$, the quantized model achieves near-perfect accuracy at layer 2 (Train: 100.0%, Test1: 100.0%, Test2: 99.4%), indicating optimal performance at this depth. Beyond layer 3, accuracy gradually degrades, with Test2 accuracy falling to 58.1% by layer 10. This suggests that deeper networks may amplify quantization-related degradation, especially in generalization. For $p_2$, the quantized model demonstrates stable and robust accuracy across most depths. Starting from moderate performance in layer 1 (Train: 69.7%, Test2: 65.6%), accuracy increases quickly and exceeds 83.0% from layer 2 onward in Train and Test1 splits. In particular, the accuracy of Test2 continues to improve up to layer 7 (80.6%), showing resilience to quantization effects even in deeper architectures. In contrast, $p_3$ exhibits more irregular behavior. Accuracy improves slightly in the early layers (Test2 peaks at 77.0% at layer 2), but then drops substantially, reaching a low of 33.4% at layer 5. Despite stable Train and Test1 accuracy ( 76–78%), the significant reduction in Test2 suggests overfitting and reduced generalization performance in deeper networks due to quantization. Dynamic PTQ preserves performance well for $p_2$ in depths, but negatively impacts $p_1$ and especially $p_3$ in deeper configurations. This underscores the need for depth-sensitive or property-sensitive quantization strategies when deploying GNNs under resource constraints.

Table 6: Difference in the percentages of the accuracy of ACR-GNN with ReLU before and after dynamic PTQ, rounded to two decimal places.

| Layer | $p_1$ | | | $p_2$ | | | $p_3$ | | |
|---|---|---|---|---|---|---|---|---|---|
| | Train | Test 1 | Test 2 | Train | Test 1 | Test 2 | Train | Test 1 | Test 2 |
| 1 | -0.45% | -0.76% | 0.52% | -0.13% | -0.18% | 8.89% | -0.30% | -0.65% | -0.69% |
| 2 | 0.00% | 0.00% | -0.04% | 0.08% | -0.13% | 0.14% | -0.18% | -0.23% | 0.02% |
| 3 | -0.04% | 0.06% | -0.49% | -0.16% | -0.14% | -0.34% | -0.02% | 0.28% | -0.35% |
| 4 | 0.01% | 0.02% | -0.40% | -0.19% | 0.19% | 0.06% | -0.05% | -0.20% | -1.99% |
| 5 | -0.06% | -0.13% | 0.19% | -0.11% | 0.06% | 0.26% | 0.03% | -0.22% | -0.73% |
| 6 | 0.02% | 0.01% | 0.06% | -0.03% | -0.18% | 0.70% | -0.23% | -0.25% | 0.95% |
| 7 | 0.00% | -0.11% | -0.16% | -0.19% | -0.26% | 0.12% | -0.00% | -0.17% | -0.75% |
| 8 | -0.03% | -0.09% | -0.01% | -0.12% | -0.12% | -0.02% | -0.05% | -0.28% | -0.49% |
| 9 | -0.03% | -0.01% | -0.04% | 0.01% | 0.21% | -0.13% | -0.26% | -0.72% | -3.74% |
| 10 | -0.00% | -0.10% | -7.38% | -0.14% | 0.05% | 0.20% | -0.08% | -0.14% | -0.78% |

Table 6 reports the accuracy differences in percentage points between the original ACR-GNN model with ReLU activation and its dynamically quantized counterpart, using Post-Training quantization (PTQ). The results cover three FO properties ($p_1$, $p_2$, $p_3$), three dataset splits (Train, Test1, Test2). Positive values indicate better accuracy after quantization, while negative values indicate degradation. For $p_1$, quantization generally causes negligible or negative changes in accuracy. For example, at layer 2, the differences are minimal (Train: 0.00%, Test1: 0.00%, Test2: -0.04%), showing near-identical behavior between the models. However, deeper networks experience more substantial performance drops, especially at layer 10 in Test2 (-7.38%), indicating increased instability due to depth quantization. These patterns highlight a general sensitivity to depth, particularly when generalizing to larger test graphs. In contrast, $p_2$ exhibits greater resilience to quantization, with occasional performance gains. A notable improvement appears in layer 1 on Test2 (+8.89%), along with smaller gains in layers 5 (+0.26%), 6 (+0.70%) and 10 (+0.20%). However, inconsistencies are still present, for example, a Test2 drop at layer 3 (-0.34%) – which implies that while $p_2$ benefits more than $p_1$, gains are not uniform across the board. $p_3$, on the other hand, exhibits the most erratic behavior and is generally more susceptible to quantization. Although a modest gain appears in layer 6 in Test2 (+0.95%), severe degradation is observed in layer 4 (-1.99%) and layer 9 (-3.74%). Across layers and divisions, accuracy losses dominate, suggesting that $p_3$ is particularly sensitive to quantization, especially in deeper models. In summary, dynamic PTQ results in non-uniform effects across properties, dataset splits, and depths. Although $p_2$ shows the most consistent tolerance and even improvement in certain cases, $p_1$ and $p_3$ are more susceptible to degradation, especially in the Test2 split in deeper configurations. These results emphasize the importance of property-specific and depth-aware quantization strategies to maintain performance in FO-property learning with GNN.

Table 7 presents the accuracy of the ACR-GNN model with truncated ReLU activation on three FO properties ($p_1$, $p_2$, and $p_3$), evaluated on the Train, Test1, and Test2 datasets as the number

Table 7: Accuracy of the ACR-GNN with truncated ReLU according to the number of layers.

| Layer | $p_1$ | | | $p_2$ | | | $p_3$ | | |
|---|---|---|---|---|---|---|---|---|---|
| | Train | Test 1 | Test 2 | Train | Test 1 | Test 2 | Train | Test 1 | Test 2 |
| 1 | 98.7% | 98.4% | 87.0% | 77.2% | 78.3% | 51.1% | 69.9% | 69.8% | 71.5% |
| 2 | 100.0% | 100.0% | 98.3% | 69.8% | 70.0% | 63.7% | 75.2% | 76.5% | 75.3% |
| 3 | 63.1% | 61.7% | 57.9% | 67.8% | 67.6% | 62.9% | 66.3% | 65.7% | 70.6% |
| 4 | 58.4% | 58.0% | 48.6% | 66.4% | 66.3% | 61.3% | 61.2% | 59.2% | 50.3% |
| 5 | 55.7% | 54.3% | 50.4% | 63.0% | 64.3% | 39.6% | 64.4% | 65.1% | 66.5% |
| 6 | 55.5% | 54.6% | 50.1% | 63.0% | 64.3% | 39.5% | 58.2% | 57.3% | 34.6% |
| 7 | 53.8% | 54.2% | 51.4% | 63.4% | 64.9% | 41.7% | 57.1% | 56.0% | 23.3% |
| 8 | 52.7% | 53.6% | 50.8% | 63.1% | 64.0% | 40.0% | 61.4% | 61.5% | 55.3% |
| 9 | 52.5% | 52.5% | 51.1% | 65.0% | 65.0% | 49.2% | 57.2% | 56.0% | 24.7% |
| 10 | 54.7% | 54.8% | 51.1% | 63.0% | 64.3% | 39.6% | 57.2% | 55.6% | 23.4% |

of GNN layers increases from 1 to 10. For $p_1$, the model exhibits strong performance in shallow configurations, peaking at layer 2 with 100.0% (Train), 100.0% (Test1), and 98.3% (Test2) accuracy. However, performance deteriorates significantly beyond this point: by layer 3, Test2 accuracy drops to 57.9%, and continues to decline in deeper layers, stabilizing around 51.1% by layer 10. This trend suggests overfitting, as training accuracy remains high while generalization performance on Test2 degrades with depth. The accuracy profile of $p_2$ is more stable. While initial performance is moderate (Test2: 51.1% at layer 1), the model maintains consistent accuracy from layer 3 onward, with minor fluctuations. The narrower gap between training and testing accuracy indicates that $p_2$ is less sensitive to overfitting and more robust to increasing depth. For $p_3$, the model initially performs well, reaching 75.3% on Test2 at layer 2. However, deeper architectures result in a steep decline in generalization performance: Test2 accuracy falls to 50.3% at layer 4, 34.6% at layer 6, and just 23.3% by layer 7. Despite relatively stable scores on Train and Test1, the Test2 drop—evidenced by a gap of over 38 percentage points at layer 7—reflects significant overfitting. In summary, ACR-GNN model with truncated ReLU benefits most from shallow architectures for $p_1$ and $p_3$, whereas $p_2$ exhibits more resilient behavior across network depths. These results highlight the need for depth-aware design when targeting different FO properties under quantization constraints.

Table 8: Accuracy of the ACR-GNN with truncated ReLU after dynamic PTQ according to the number of layers.

| Layer | $p_1$ | | | $p_2$ | | | $p_3$ | | |
|---|---|---|---|---|---|---|---|---|---|
| | Train | Test 1 | Test 2 | Train | Test 1 | Test 2 | Train | Test 1 | Test 2 |
| 1 | 98.8% | 98.8% | 86.4% | 76.2% | 77.8% | 59.5% | 69.4% | 69.3% | 74.8% |
| 2 | 100.0% | 100.0% | 94.4% | 69.6% | 69.7% | 42.4% | 74.8% | 76.3% | 59.6% |
| 3 | 61.5% | 59.1% | 54.9% | 67.8% | 68.0% | 63.6% | 66.1% | 65.3% | 70.7% |
| 4 | 58.3% | 57.7% | 47.9% | 66.2% | 66.7% | 43.1% | 61.0% | 57.5% | 46.0% |
| 5 | 55.4% | 54.0% | 50.5% | 63.0% | 64.3% | 39.6% | 63.9% | 57.4% | 65.5% |
| 6 | 55.5% | 55.8% | 50.0% | 63.0% | 64.3% | 39.8% | 57.5% | 56.8% | 32.5% |
| 7 | 53.4% | 53.1% | 50.9% | 62.4% | 62.5% | 44.8% | 56.8% | 56.2% | 24.5% |
| 8 | 52.5% | 53.6% | 51.0% | 61.4% | 63.0% | 40.0% | 61.4% | 62.7% | 50.0% |
| 9 | 52.6% | 52.4% | 51.2% | 65.0% | 65.7% | 53.7% | 57.2% | 55.6% | 23.7% |
| 10 | 54.8% | 53.9% | 51.3% | 63.1% | 64.3% | 39.6% | 56.9% | 55.1% | 23.6% |

Table 8 reports the accuracy of the ACR-GNN model after applying dynamic PTQ across three logical query patterns ($p_1$, $p_2$, $p_3$) and a range of GNN layers ($l$ from 1 to 10). A general observation is that dynamic PTQ causes more pronounced performance degradation as the number of layers increases, particularly for $p_1$ and $p_3$. While accuracy remains high for shallow configurations, especially at $l = 1$ and $l = 2$ (e.g., $p_1$ reaches 98.8% on Test1 at $l = 1$ and 100.0% on Train and Test1 at $l = 2$)—a sharp decline follows beyond $l = 2$. For instance, $p_1$ training accuracy drops from 100.0% at $l = 2$

to 61.5% at $l = 3$, with continued degradation in deeper layers.In contrast, $p_2$ starts with slightly lower accuracy but exhibits relatively stable behavior across layers. Its accuracy remains in the 60–78% range across all datasets, showing less sensitivity to depth. However, a gradual decline in the precision of Test2 is noticeable, ranging from 59.5% at $l = 1$ to 39.6% at $l = 10$, suggesting that generalization to more complex test graphs is still affected by quantization. The pattern $p_3$ is the most affected. Although some recovery is observed at intermediate layers (e.g., 70.7% Test2 accuracy at $l = 3$), performance deteriorates with increasing depth, reaching only 23.6% on Test2 at $l = 10$. In summary, dynamic PTQ enables significant model compression for ACR-GNNs, but at the cost of accuracy, particularly in deeper architectures and complex FO-query patterns such as $p_1$ and $p_3$. Shallow configurations (e.g., $l \leq 2$) maintain good performance after quantization, indicating that careful depth-aware quantization strategies are essential for preserving generalization.

Table 9: Difference in the percentages of the accuracy of ACR-GNN with truncated ReLU before and after dynamic PTQ.

| Layer | $p_1$ | | | $p_2$ | | | $p_3$ | | |
|---|---|---|---|---|---|---|---|---|---|
| | Train | Test 1 | Test 2 | Train | Test 1 | Test 2 | Train | Test 1 | Test 2 |
| 1 | 0.1% | 0.3% | -0.6% | -1.0% | -0.5% | 8.4% | -0.5% | -0.5% | 3.4% |
| 2 | 0.0% | 0.0% | -3.9% | -0.2% | -0.3% | -21.3% | -0.5% | -0.2% | -15.7% |
| 3 | -1.6% | -2.7% | -3.0% | 0.0% | 0.4% | 0.7% | -0.2% | -0.4% | 0.1% |
| 4 | -0.2% | -0.3% | -0.8% | -0.2% | 0.5% | -18.2% | -0.2% | -1.7% | -4.3% |
| 5 | -0.3% | -0.3% | 0.2% | 0.0% | 0.0% | 0.0% | -0.6% | -7.7% | -1.0% |
| 6 | -0.0% | 1.2% | -0.1% | -0.0% | 0.0% | 0.3% | -0.6% | -0.5% | -2.2% |
| 7 | -0.4% | -1.2% | -0.5% | -1.0% | -2.3% | 3.1% | -0.4% | 0.2% | 1.2% |
| 8 | -0.2% | 0.0% | 0.2% | -1.7% | -1.0% | -0.0% | 0.0% | 1.3% | -5.3% |
| 9 | 0.2% | -0.1% | 0.1% | 0.0% | 0.7% | 4.5% | 0.1% | -0.5% | -1.0% |
| 10 | 0.1% | -0.9% | 0.3% | 0.0% | 0.0% | 0.0% | -0.3% | -0.5% | 0.2% |

Table 9 presents the percentage changes in accuracy of the ACR-GNN model with truncated ReLU after applying Dynamic Post-Training quantization (PTQ), across three query patterns ($p_1, p_2, p_3$) and for different numbers of GNN layers ($l = 1$ to $l = 10$). The difference is calculated as the quantized accuracy minus the original, scaled to a percentage. In the case of this table, we can see changes layer by layer. Here, where $l = 1$, we observe small improvements in accuracy. If we examine this more precisely, for $p_1$, the precision improves across all datasets, with the highest gain in Test2 (+11.1%). $p_2$ shows a mixed pattern with small increases in Train / Test1, but a decrease in Test2 (-6.1%). $p_3$ remains stable, showing minimal change ($\leq 1.2\%$). When $l = 2$, the results show early degradation, as $p_2$ suffers significant drops, especially on Test2 (-33.0%), while $p_3$ sees a drop in Test2 of -17.4%, $p_1$ remains unchanged on Train / Test1 and slightly lower (-5.0%) on Test2. A major drop occurs when $l = 3$ for $p_1$, with -36.1% on Train and -38.3% on Test1. $p_2$ also shows a negative trend, but Test2 is impacted less than in Layer 2. Interestingly, $p_3$ has a positive change in Test2 (+4.2%), indicating some robustness in this setting. The continuous trend for layers from 4 to 9. For $l = 10$, $p_1$ appears to recover slightly in Test2 (-6.8%, compared to - 15% previously). However, $p_2$ and $p_3$ still show substantial losses (-37.9% and -13.1% respectively), suggesting that deeper architectures struggle consistently after dynamic quantization. In summary, Table 9 highlights the accuracy losses due to dynamic PTQ. This correlates with the literature [15], where the authors noted some loss in accuracy, but the quantized model should provide better results in comparing the size. Although some early layers benefit slightly, deeper layers consistently show reduced accuracy, especially in Test2, the data set with larger, more complex graphs. The pattern confirms that dynamic PTQ, though efficient, can harm generalization, particularly in deeper and more expressive GNN configurations.

After presenting the accuracy results before and after applying dynamic Post-Training Quantization (PTQ), we proceed to analyze the influence of the activation function on the performance of the model. This comparison is provided both graphically and in tabular form. For the graphical representation, we utilized box plots, a statistical tool designed to visualize the distribution of a variable in terms of its quartiles. In these plots, the box itself spans from the first quartile (Q1) to the third quartile (Q3), with the median value (Q2) marked by a line within the box. The whiskers of the box plot extend to the minimum and maximum values that do not qualify as outliers, providing insight into the spread

and concentration of the data. In addition to these visualizations, a detailed table complements the analysis by presenting summary statistics. The table includes the mean, standard deviation, minimum, and maximum values for each configuration. It also presents the three quartiles: Q1, which represents the 25th percentile, Q2, or the median, which is the 50th percentile, and Q3, the 75th percentile. These quartiles divide the data into four equal parts, helping to identify the central tendency and variability. Furthermore, we calculate the interquartile range (IQR), defined as the difference between the third quartile (Q3) and the first quartile (Q1), which serves as a measure of statistical dispersion. Based on the IQR, we also determine the lower and upper bounds using the standard rule, which involves subtracting 1.5 times the IQR from Q1 and adding it to Q3, respectively. These bounds enable the identification of potential outliers and provide a more comprehensive understanding of how the activation function and quantization impact the distribution of model accuracy. All metrics were applied to all datasets: Train, Test1, and Test2. For the visualization part, we used the Python library Plotly.

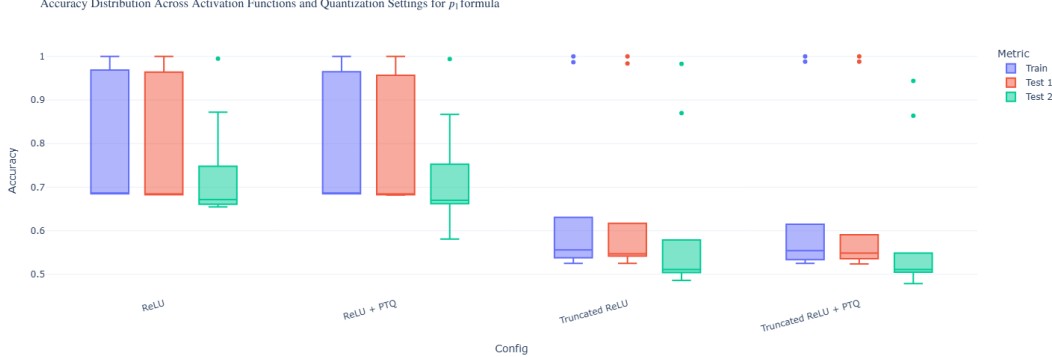

Figure 3: Detailed summary statistics across configurations for $p_1$ formula.

Table 10: Detailed summary statistics across configurations for $p_1$ formula.

| Statistic | ReLU | ReLU + PTQ | Truncated ReLU | Truncated ReLU + PTQ |
|---|---|---|---|---|
| Mean | 0.758 | 0.755 | 0.628 | 0.623 |
| Std | 0.132 | 0.134 | 0.178 | 0.177 |
| Min | 0.655 | 0.581 | 0.486 | 0.479 |
| 25% (Q1) | 0.683 | 0.682 | 0.525 | 0.524 |
| 50% (Median) | 0.685 | 0.685 | 0.547 | 0.544 |
| 75% (Q3) | 0.841 | 0.839 | 0.609 | 0.589 |
| Max | 1.000 | 1.000 | 1.000 | 1.000 |
| IQR | 0.158 | 0.157 | 0.084 | 0.065 |
| Lower Bound | 0.446 | 0.447 | 0.399 | 0.427 |
| Upper Bound | 1.078 | 1.073 | 0.734 | 0.686 |

Table 10 and Figure 3 present summary statistics for the accuracy results obtained from four configurations of the ACR-GNN model: ReLU, ReLU with dynamic Post-Training Quantization (PTQ), Truncated ReLU, and Truncated ReLU with PTQ. The results show that the highest mean accuracy is achieved with the ReLU configuration (0.758), closely followed by ReLU + PTQ (0.755). This indicates that applying dynamic quantization to the ReLU model does not significantly reduce the average accuracy. In contrast, both Truncated ReLU (0.628) and Truncated ReLU + PTQ (0.623) result in noticeably lower mean values, suggesting that this activation function may degrade performance on the $p_1$ query pattern. The median values align with the mean, further confirming this trend. In terms of variability, the standard deviation is lower for the ReLU-based models ( 0.13), whereas the truncated ReLU configurations show higher variability ( 0.18). This pattern is also reflected in the interquartile range (IQR): ReLU configurations exhibit wider IQRs (0.158 and 0.157), while truncated versions have narrower ranges (0.084 and 0.065). Despite the narrower spread, the performance is consistently lower with truncated ReLU. All configurations include samples that achieve a maximum

accuracy of 1.0, indicating that optimal predictions are possible in all cases. However, minimum accuracy drops more sharply in truncated ReLU models (0.486 and 0.479) compared to ReLU (0.655 and 0.581), indicating a higher risk of underperformance. The lower and upper bounds provide insight into potential outliers. The lower bounds are lower in the truncated models, while the upper bounds are higher in ReLU configurations (exceeding 1.0 due to statistical calculation), indicating a wider spread and potentially higher ceiling for performance.

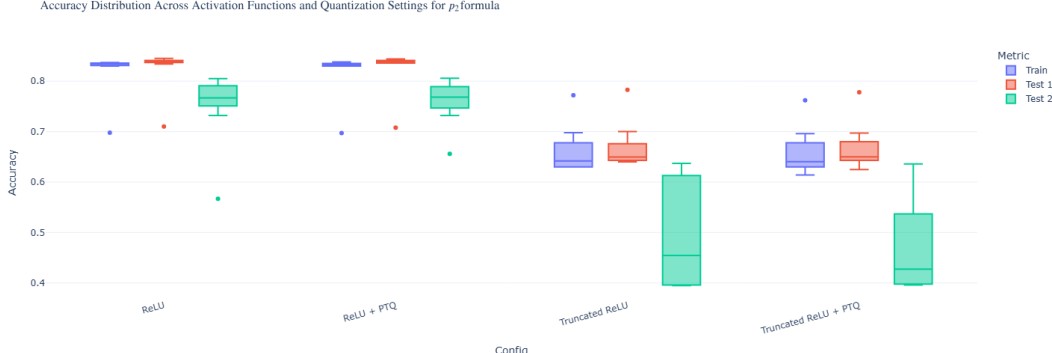

Figure 4: Detailed summary statistics across configurations for $p_2$ formula.

Table 11: Detailed summary statistics across configurations for $p_2$ formula.

| Statistic | ReLU | ReLU + PTQ | Truncated ReLU | Truncated ReLU + PTQ |
|---|---|---|---|---|
| Mean | 0.7992 | 0.8020 | 0.6064 | 0.5967 |
| Std | 0.0615 | 0.0511 | 0.1085 | 0.1122 |
| Min | 0.5670 | 0.6560 | 0.3950 | 0.3960 |
| 25% (Q1) | 0.7738 | 0.7758 | 0.6170 | 0.5515 |
| 50% (Median) | 0.8340 | 0.8330 | 0.6385 | 0.6305 |
| 75% (Q3) | 0.8370 | 0.8368 | 0.6598 | 0.6608 |
| Max | 0.8450 | 0.8440 | 0.7830 | 0.7780 |
| IQR | 0.0632 | 0.0610 | 0.0428 | 0.1093 |
| Lower Bound | 0.6789 | 0.6843 | 0.5529 | 0.3876 |
| Upper Bound | 0.9319 | 0.9282 | 0.7239 | 0.8246 |

Table 11 and Figure 4 present a comprehensive overview of the accuracy results in four model configurations: ReLU, ReLU with dynamic post-training quantization (PTQ), Truncated ReLU, and Truncated ReLU with PTQ - for the query formula $p_2$. From the mean accuracy values, ReLU and ReLU + PTQ clearly outperform the other configurations, achieving 0.7992 and 0.8020, respectively. This indicates that both setups yield strong overall performance, with dynamic quantization having a slightly positive effect on average accuracy in this case. In contrast, Truncated ReLU (0.6064) and Truncated ReLU + PTQ (0.5967) show substantially lower mean values, highlighting a notable drop in predictive performance when using truncated activation. Looking at the variability, the standard deviation is lower for the ReLU configurations (0.0615 and 0.0511), suggesting a more consistent accuracy. The truncated versions, especially the quantized one (0.1122), are more dispersed, reflecting greater instability. This is further emphasized by the IQR values: 0.0632 and 0.0610 for ReLU and ReLU + PTQ versus 0.0428 for Truncated ReLU and a larger 0.1093 for Truncated ReLU + PTQ. The larger IQR for Truncated ReLU + PTQ implies a larger fluctuation in the middle 50% of the data, despite its lower central values. The median values confirm this trend: both ReLU configurations cluster around 0.833–0.834, while truncated versions fall between 0.6305 and 0.6385. The lower bounds, derived from Q1 − 1.5 × IQR, are also lower in the Truncated ReLU + PTQ case (0.3876), indicating a greater potential for underperformance and a higher risk of poor accuracy. The maximum and minimum values highlight the performance extremes. ReLU configurations reach up to 0.845 and 0.844, significantly higher than the 0.783 and 0.778 of truncated variants. The lower minimum accuracy (0.395–0.396) in truncated settings further reinforces concerns about their reliability.

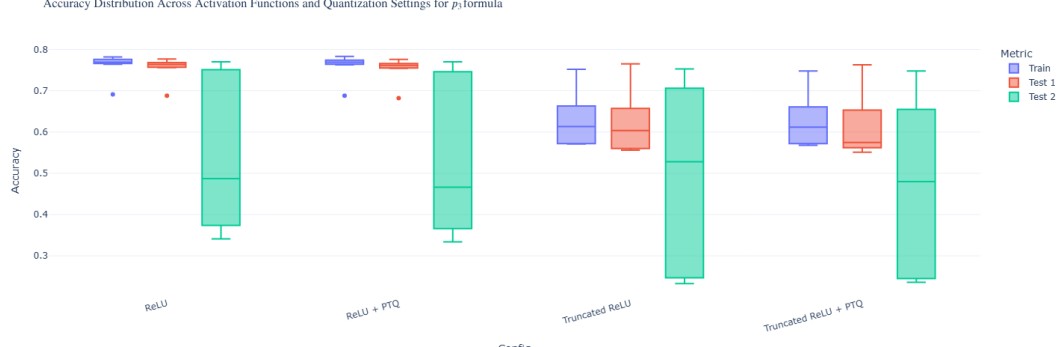

Figure 5: Detailed summary statistics across configurations for $p_3$ formula.

Table 12: Detailed summary statistics across configurations for $p_3$ formula.

| Statistic | ReLU | ReLU + PTQ | Truncated ReLU | Truncated ReLU + PTQ |
|---|---|---|---|---|
| Mean | 0.6883 | 0.6844 | 0.5821 | 0.5694 |
| Std | 0.1434 | 0.1466 | 0.1441 | 0.1427 |
| Min | 0.3410 | 0.3340 | 0.2330 | 0.2360 |
| 25% (Q1) | 0.6888 | 0.6835 | 0.5600 | 0.5575 |
| 50% (Median) | 0.7635 | 0.7615 | 0.6020 | 0.5750 |
| 75% (Q3) | 0.7688 | 0.7670 | 0.6645 | 0.6545 |
| Max | 0.7820 | 0.7830 | 0.7650 | 0.7630 |
| IQR | 0.0800 | 0.0835 | 0.1045 | 0.0970 |
| Lower Bound | 0.5687 | 0.5582 | 0.4032 | 0.4120 |
| Upper Bound | 0.8888 | 0.8922 | 0.8213 | 0.8000 |

Table 12 and Figure 5 provide descriptive statistics for the accuracy of the ACR-GNN model under four configurations—ReLU, ReLU with dynamic Post-Training Quantization (PTQ), Truncated ReLU, and Truncated ReLU with PTQ—for the $p_3$ query formula. Starting with the mean accuracy, ReLU (0.6883) and ReLU + PTQ (0.6844) again outperform the Truncated ReLU configurations, which register noticeably lower means of 0.5821 and 0.5694, respectively. This indicates that models that use ReLU activations are generally more effective for $p_3$. The standard deviation values are relatively similar across all configurations (approximately 0.14), suggesting that while the truncated configurations perform worse on average, they do not fluctuate more widely than the ReLU-based ones. The minimum values further emphasize the performance gap: ReLU models maintain minimum accuracies above 0.33, while truncated variants drop to as low as 0.233. This shows that truncated configurations are more prone to poor performance in the worst-case scenarios. In terms of quartiles, ReLU and ReLU + PTQ have Q1 and Q3 clustered around 0.68–0.77, indicating that the middle 50% of their results are concentrated within a tight and relatively high accuracy range. Truncated ReLU variants have their Q1 around 0.56 and Q3 near 0.65, which not only shows lower performance but also a wider IQR (0.1045 for Truncated ReLU and 0.0970 for Truncated ReLU + PTQ). This reflects more variability across the central portion of the data in the truncated setups. The median accuracy is again higher in ReLU configurations (around 0.76), compared to 0.60 and 0.575 for truncated ones, reinforcing the conclusion that ReLU configurations are more reliable. Examining the bounds, the ReLU models show a lower bound above 0.55 and upper bounds above 0.88, suggesting strong and consistent performance. Truncated models exhibit lower bounds near 0.40 and upper bounds around 0.80, indicating both a lower floor and a lower ceiling in performance.

Across all query patterns ($p_1$, $p_2$, and $p_3$), ReLU and ReLU + PTQ consistently demonstrate higher average accuracy and more stable performance, making them the most reliable configurations. In contrast, Truncated ReLU and its quantized variant result in lower accuracy and greater variability, especially in worst-case scenarios. Dynamic PTQ tends to maintain or slightly enhance performance in ReLU models, but its effect on truncated activations is less favorable, often introducing further

inconsistency. Overall, ReLU-based configurations—quantized or not—are better suited for the ACR-GNN model across the evaluated formulas.

Other parameters of interest to us are the time and size of the models. In the event of changes in size, it is easy to compare the data using the bar plots presented in Figure 6. The size changes in percentages we calculated according to the formula:

$$\text{Difference in percentages} = \frac{\text{Value}_{dPTQ} - \text{Value}_{original}}{\text{Value}_{original}} * 100\%$$

In other words, this formula shows how much the dynamic PTQ value deviates from the original value as a percentage of the original value.

In this section, we compare parameters for different activation functions. We observe that the results of size changes in the following models remain unchanged when we modify the training dataset. We present the results not only graphically but also in a tabular format. In the plots, it is possible to see the trends and, in the tabular format, the numerical changes.

Table 13: Detailed information about the size of the model. The size values are in megabytes and refer to the file sizes of the GNNs.

| Layer | Original Size (MB) | Quantized Size (MB) | Difference (MB) | Reduction (%) |
|---|---|---|---|---|
| 1 | 0.057 | 0.023 | 0.034 | 59.604% |
| 2 | 0.112 | 0.044 | 0.068 | 60.993% |
| 3 | 0.167 | 0.064 | 0.103 | 61.559% |
| 4 | 0.221 | 0.085 | 0.137 | 61.804% |
| 5 | 0.276 | 0.105 | 0.171 | 61.975% |
| 6 | 0.331 | 0.126 | 0.206 | 62.068% |
| 7 | 0.386 | 0.146 | 0.240 | 62.148% |
| 8 | 0.441 | 0.167 | 0.274 | 62.194% |
| 9 | 0.496 | 0.187 | 0.309 | 62.230% |
| 10 | 0.551 | 0.208 | 0.343 | 62.251% |

Table 13 provides a detailed comparison of the model sizes before and after applying dynamic post-training quantization (PTQ). As the number of layers increases, both the original and quantized model sizes grow; however, the percentage reduction remains remarkably consistent, ranging from approximately 60.993% at 2 layers to 62.251% at 10 layers. This stable percentage reduction, approximately 60–62%—indicates that PTQ effectively compresses the model regardless of its depth, significantly reducing the memory footprint without altering the underlying architecture of the GNN. Such a reduction is particularly crucial for deployments in resource-constrained environments.

Furthermore, after presenting the tabular data, our graphs (Figure 6) reveal a clear trend: While the absolute sizes of the original and quantized models increase with the number of layers, the relative reduction achieved through dynamic PTQ remains consistent. The size of the original model increases approximately linearly from 0.057 MB for $l = 1$ to 0.551 MB at $l = 10$, while the quantized model grows from 0.023 MB to 0.208 MB, preserving the growth structure, but on a reduced scale. The absolute size difference increases from 0.034 MB in $l = 1$ to 0.343 MB in $l = 10$, demonstrating that quantization becomes more beneficial for deeper models. Overall, the consistent percentage reduction across all tested configurations confirms that PTQ scales effectively, delivering stable compression rates and making it an attractive option for deeper GNN deployments in real-world edge or mobile environments.

Moreover, we observed that the query property had no noticeable impact on the model size. This can be clearly seen in the bar plots in Figure 6a, Figure 6c, and Figure 6e.

We also measured the change over time. Specifically, we considered three distinct time metrics: `Elapsed time` (the time taken during training), `Time Original` (the time required for inference on the test datasets using the original trained model), and `Time quantized` (the inference time on the test datasets using the quantized model). These results are presented in Figure 7.

The data in Figure 7 reflect the impact of dynamic PTQ on the ACR-GNN model in three query patterns ($p_1$, $p_2$, and $p_3$) and for GNN depths ranging from 1 to 10 layers. Across all patterns,

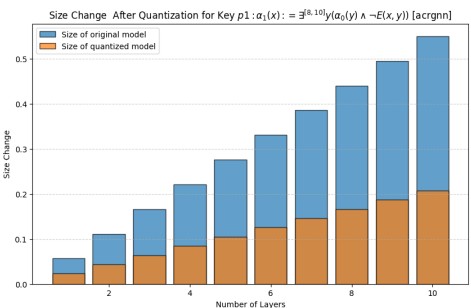

(a) Size changes in MB for the first formula

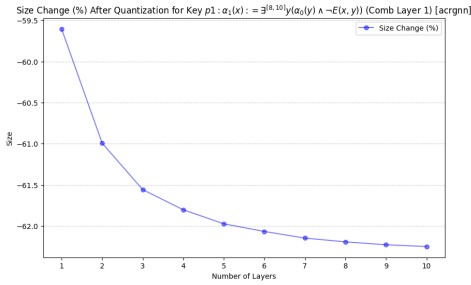

(b) Size changes in MB for the first formula. Difference present in percentage.

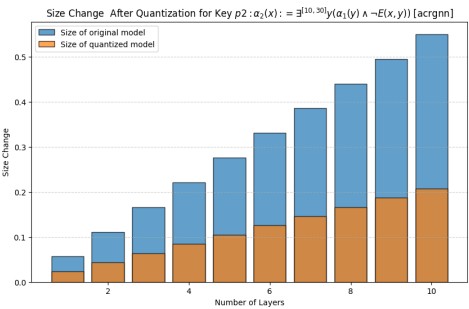

(c) Size changes in MB for the second formula

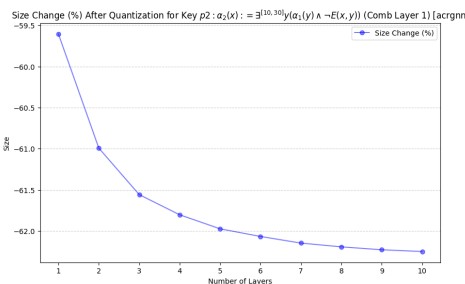

(d) Size changes in MB for the second formula. Difference present in percentage.

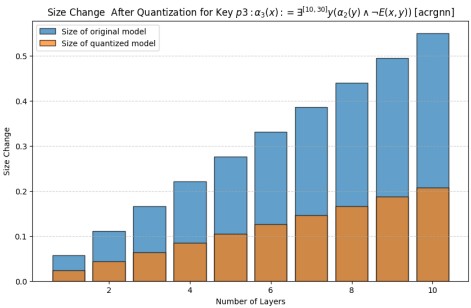

(e) Size changes in MB for the third formula

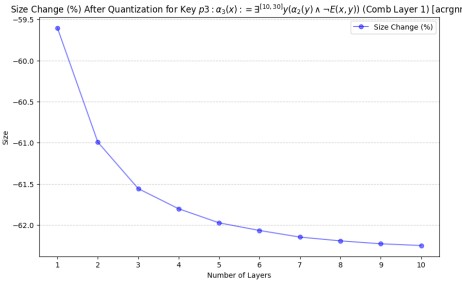

(f) Size changes in MB for the third formula Difference present in percentage.

Figure 6: Impact of dynamic Post-Training quantization on model size (MB). Changes of size in percentages

quantized models consistently require more inference time than their original counterparts. This increased time is expected as a result of the real-time quantization of weights and activations during inference. Additionally, both the original and quantized models exhibit a consistent, near-linear increase in inference time with model depth, suggesting that computational complexity grows linearly as layers are added.

Despite this overhead, which ranges between 0.1 and 0.9 s depending on the number of layers, the significant reduction in model size (as demonstrated in Table 13 and the corresponding graphs) makes quantized models especially attractive for resource-constrained environments where minimizing the memory footprint is more critical than achieving the lowest possible latency.

To test the technique not only on synthetic data, we chose the Protein-Protein Interactions (PPI) benchmark. The PPI dataset consists of graph-level mini-batches, with separate splits for Training, Validation, and Testing.

In Table 14, we present a summary of the PPI dataset, which consists of 20 training graphs, 2 validation graphs, and 2 test graphs. Each graph contains nodes with 50-dimensional features and supports multi-label classification with 121 possible labels. On average, each node is associated with

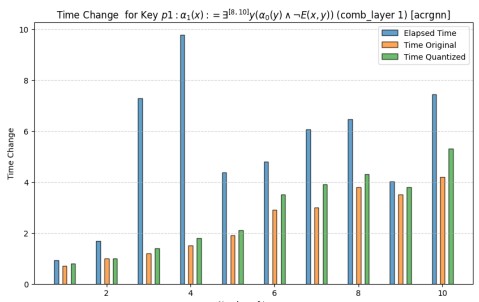

(a) Time changes in seconds for the first formula

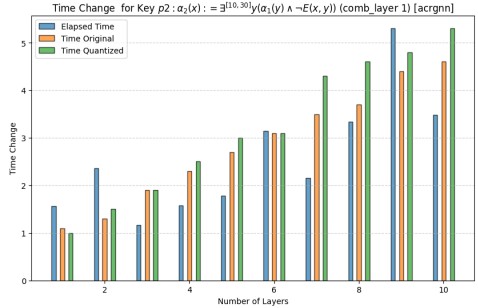

(b) Time changes in seconds for the second formula

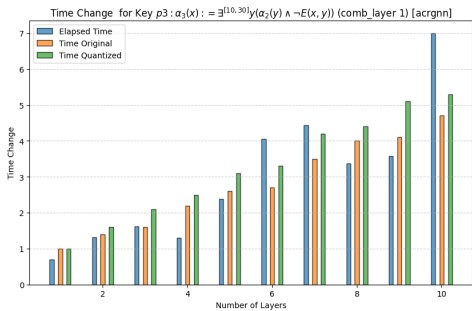

(c) Time changes in seconds for the third formula

Figure 7: Impact of dynamic Post-Training quantization on Latency (sec)

Table 14: Dataset summary.

| Dataset | Num Graphs | Node Feature Dim | Label Dim | Avg Active Labels/Node | Avg Degree |
|---------|-----------|------------------|-----------|------------------------|------------|
| Train | 20 | 50 | 121 | 37.20 | 54.62 |
| Validation | 2 | 50 | 121 | 35.64 | 61.07 |
| Test | 2 | 50 | 121 | 36.22 | 58.64 |

approximately 36 labels, indicating a densely labelled dataset. The average node degree is also high, ranging from 54.6 in the training set to 61.1 in the validation set, reflecting the dense connectivity of the protein-protein interaction graphs. The dataset presents a complex multi-label classification task with consistently rich structure across all splits.

Table 15: Dataset statistics summary.

| | Node | | | Edge | | |
|---------|------|------|---------|------|------|---------|
| Dataset | Min | Max | Avg | Min | Max | Avg |
| Train | 591 | 3480 | 2245.30 | 7708 | 106754 | 61318.40 |
| Validation | 3230 | 3284 | 3257.00 | 97446 | 101474 | 99460.00 |
| Test | 2300 | 3224 | 2762.00 | 61328 | 100648 | 80988.00 |

The statistics of the dataset presented in Table 15 contain large graphs with varying sizes between the train, the validation, and the test splits. Training graphs range from 591 to 3,480 nodes, with an average of 2,245 nodes per graph, and between 7,708 and 106,754 edges (average 61,318 edges). Validation graphs are more consistent in size, with 3,230 to 3,284 nodes and 97,446 to 101,474 edges, averaging 3,257 nodes and 99,460 edges. The test graphs have 2,300 to 3,224 nodes, averaging 2,762 nodes, and 61,328 to 100,648 edges, averaging 80,988. These statistics confirm that the dataset

contains large and densely connected graphs and demonstrate a distributional shift in graph size and edge count between training and test data. This information is helpful in evaluating the model's ability to generalize to unseen and variable graph structures.

One key difference between the synthetic data and the PPI dataset is that the latter involves a multi-label classification task, rather than a binary classification task, because the PPI dataset is a common benchmark where each node (representing proteins) can have multiple labels, such as protein functions or interactions. Also, it is important to mention the key differences between the synthetic data and the real one. Here, the authors used the code function `EarlyStopping`: Utility for stopping training early if no further improvement is observed. The second difference is that the code is structured to run multiple experiments to collect statistics (mean and standard deviation) of the model performance, ensuring that the results are robust across different random initializations. In this case, we performed the experiments 10 times for each model, with a combination layer equal to 1 and a number of layers ranging from 1 to 10. The number of hidden dimensions is equal to 256.

For these experiments, we used two activation functions to compare the results with synthetic data. The presentation of the results follows the same approach as for synthetic data. Moreover, in the case of real data [4] used the F1 Score as an evaluation metric. This metric is commonly used to evaluate classification tasks.

According to the Scikit-learn library [25], the F1 score is defined in the following way. The F1 score can be interpreted as a harmonic mean of precision and recall, where an F1 score reaches its best value at 1 and its worst score at 0. The relative contribution of precision and recall to the F1 score is equal. The formula for the F1 score is as follows:

$$\text{F1} = \frac{2\text{TP}}{2\text{TP} + \text{FP} + \text{FN}}$$

where, TP – is the number of true positives, FN – is the number of false negatives, FP – is the number of false positives. F1 is calculated by default as 0.0 when there are no true positives, false negatives, or false positives.

The reference code's results [5] are structured as follows: a table showing the loss and accuracy for each dataset (train, validation, and test). Here, we present only the accuracy of the model according to the number of layers, as we do for the synthetic data. For better representation, we formed the model's output in a tabular representation.

Table 16: Accuracy for the original and quantized (dynamic PTQ) models. PPI Benchmark.

(a) Accuracy of the ACR-GNN with ReLU according to the number of layers.

| Layer | Train | Validation | Test |
|-------|-------|------------|------|
| 1 | 54.7% | 43.1% | 39.5% |
| 2 | 52.5% | 44.6% | 45.7% |
| 3 | 52.3% | 42.6% | 44.0% |
| 4 | 52.3% | 39.2% | 40.6% |
| 5 | 49.6% | 39.7% | 39.1% |
| 6 | 49.3% | 43.5% | 43.3% |
| 7 | 51.7% | 39.9% | 38.5% |
| 8 | 50.8% | 36.3% | 35.8% |
| 9 | 48.0% | 43.8% | 33.2% |
| 10 | 47.1% | 36.9% | 36.8% |

(b) Accuracy of the ACR-GNN with ReLU after dynamic PTQ according to the number of layers.

| Layer | Train | Validation | Test |
|-------|-------|------------|------|
| 1 | 55.0% | 50.8% | 50.2% |
| 2 | 52.3% | 47.8% | 47.2% |
| 3 | 51.9% | 45.7% | 42.8% |
| 4 | 51.9% | 37.4% | 34.1% |
| 5 | 48.9% | 39.1% | 40.8% |
| 6 | 48.9% | 42.9% | 43.8% |
| 7 | 51.4% | 43.0% | 40.6% |
| 8 | 50.5% | 35.9% | 36.8% |
| 9 | 47.7% | 40.8% | 40.9% |
| 10 | 46.5% | 36.2% | 38.7% |

Table 16 reports the precision of the ACR-GNN model with ReLU activation in varying numbers of layers, both in its original form and after applying dynamic post-training quantization (dPTQ). The results are presented for the training, validation, and test sets of the PPI benchmark. For both versions of the model, the performance does not increase consistently with the number of layers. Instead, accuracy typically peaks within the first few layers and tends to degrade or fluctuate as the network's depth increases. In particular, the highest accuracies for the training, validation, and test

sets are achieved with 1 or 2 layers, indicating that shallower architectures are better suited for this task. Specifically, the original model achieves its best test accuracy (45.7%) at 2 layers, while the quantized model achieves an even higher test accuracy (50.2%) at just 1 layer. Dynamic quantization slightly improves generalization performance in the early layers. At layer 1, the quantized model surpasses the original in both validation (50.8% vs. 43.1%) and test accuracy (50.2% vs. 39.5%), suggesting that quantization can have a regularizing effect in low-depth configurations. However, as the number of layers increases beyond 4, the performance of both models tends to decline, likely due to over-smoothing or optimization difficulties common in deep GNNs.

Table 17: Difference in accuracy of ACR-GNN with ReLU before and after dynamic PTQ. PPI Benchmark.

| Layer | Train | Validation | Test |
|---|---|---|---|
| 1 | 0.3% | 7.7% | 10.7% |
| 2 | -0.2% | 3.2% | 1.5% |
| 3 | -0.4% | 3.1% | -1.2% |
| 4 | -0.4% | -1.8% | -6.5% |
| 5 | -0.7% | -0.6% | 1.7% |
| 6 | -0.4% | -0.6% | 0.5% |
| 7 | -0.3% | 3.1% | 2.1% |
| 8 | -0.3% | -0.4% | 1.0% |
| 9 | -0.3% | -3.0% | 7.7% |
| 10 | -0.6% | -0.7% | 1.9% |

Table 17 reports the absolute difference in precision between the quantized and original ACR-GNN model with ReLU on the PPI benchmark, between training, validation and test sets for varying numbers of layers. Positive values indicate better performance after quantization, while negative values reflect performance degradation. At layer 1, the quantized model shows the largest gains, with improvements of 7.7% on validation and 10.7% on the test set, suggesting a clear generalization advantage in shallow architectures. Smaller, but consistent improvements are also observed at layers 2 and 7, particularly in the validation and test sets. In contrast, certain layers exhibit minor drops in accuracy. For example, layer 4 shows the largest decrease in the test set (6.5%). Overall, the results indicate that dynamic quantization can lead to modest accuracy improvements, particularly in shallow to mid-depth GNNs, with negligible or slightly negative effects in deeper configurations. This highlights the potential of quantization for lightweight deployment with minimal accuracy trade-offs.

Table 18: Detailed information about the model size before and after quantization. PPI Benchmark. Sizes are in megabytes.

| Layer | Original Model (MB) | Quantized Model (MB) | Difference (MB) | Reduction (%) |
|---|---|---|---|---|
| 1 | 0.922 | 0.242 | 0.680 | -73.749% |
| 2 | 1.718 | 0.451 | 1.267 | -73.765% |
| 3 | 2.515 | 0.660 | 1.855 | -73.772% |
| 4 | 3.311 | 0.868 | 2.443 | -73.776% |
| 5 | 4.108 | 1.077 | 3.031 | -73.778% |
| 6 | 4.904 | 1.286 | 3.618 | -73.779% |
| 7 | 5.701 | 1.495 | 4.206 | -73.780% |
| 8 | 6.497 | 1.704 | 4.794 | -73.780% |
| 9 | 7.294 | 1.912 | 5.382 | -73.781% |
| 10 | 8.090 | 2.121 | 5.969 | -73.781% |

Table 18 presents the memory footprint of the ACR-GNN model at different layer depths, comparing the original model (complete precision) with its dynamically quantized counterpart. The table also includes both absolute and percentage differences in size, highlighting the compression effect introduced by dynamic post-training quantization. Across all layers, the quantized model consistently exhibits a size reduction of approximately 73.78% compared to the original model. For example, at 10

layers, the model size decreases from 8.09MB to 2.12MB, yielding an absolute reduction of 5.97MB. This trend is consistent and proportional across all depths, indicating that the memory savings scale linearly with the model's complexity (i.e., the number of layers). These results demonstrate the effectiveness of dynamic quantization in significantly reducing model size without the need for retraining.

Table 19: Elapsed times (in seconds) for the original and quantized (dynamic PTQ) models. PPI Benchmark.

<table>
<tr><td colspan="4">(a) Elapsed times for the original model.</td><td colspan="4">(b) Elapsed times for the quantized model.</td></tr>
<tr><td>Layer</td><td>Train</td><td>Validation</td><td>Test</td><td>Layer</td><td>Train</td><td>Validation</td><td>Test</td></tr>
<tr><td>1</td><td>0.913</td><td>0.115</td><td>0.113</td><td>1</td><td>0.921</td><td>0.134</td><td>0.112</td></tr>
<tr><td>2</td><td>1.400</td><td>0.158</td><td>0.182</td><td>2</td><td>1.469</td><td>0.178</td><td>0.129</td></tr>
<tr><td>3</td><td>1.447</td><td>0.188</td><td>0.172</td><td>3</td><td>1.410</td><td>0.211</td><td>0.173</td></tr>
<tr><td>4</td><td>1.982</td><td>0.257</td><td>0.224</td><td>4</td><td>1.694</td><td>0.252</td><td>0.181</td></tr>
<tr><td>5</td><td>2.225</td><td>0.295</td><td>0.247</td><td>5</td><td>2.538</td><td>0.322</td><td>0.304</td></tr>
<tr><td>6</td><td>2.846</td><td>0.318</td><td>0.236</td><td>6</td><td>2.878</td><td>0.307</td><td>0.313</td></tr>
<tr><td>7</td><td>3.420</td><td>0.442</td><td>0.328</td><td>7</td><td>3.538</td><td>0.328</td><td>0.299</td></tr>
<tr><td>8</td><td>3.120</td><td>0.437</td><td>0.343</td><td>8</td><td>3.236</td><td>0.360</td><td>0.342</td></tr>
<tr><td>9</td><td>3.626</td><td>0.433</td><td>0.390</td><td>9</td><td>3.936</td><td>0.605</td><td>0.481</td></tr>
<tr><td>10</td><td>4.011</td><td>0.410</td><td>0.376</td><td>10</td><td>3.783</td><td>0.464</td><td>0.375</td></tr>
</table>

Table 21 reports the inference times of the original and dynamically post-training quantized ACR-GNN models across training, validation, and test datasets, measured at various layer depths. The results reveal that quantization does not significantly reduce inference time in most configurations and, in some cases, results in slightly higher latency. For the training set, the execution time of the quantized model closely follows that of the original, with negligible differences across all layers. In the validation and test sets, while some improvements are observed at shallow depths (e.g., the layer 2 test time reduces from 0.182 to 0.129 s), the overall pattern indicates no consistent speedup from quantization. In fact, certain configurations, such as layers 9 and 10 in the validation set, exhibit increased latency in the quantized version compared to the original.

Table 20: Difference in elapsed time (in seconds) and corresponding percentage difference of ACR-GNN with ReLU before and after dynamic PTQ on the PPI Benchmark.

| Layer | Train | | Validation | | Test | |
|---|---|---|---|---|---|---|
| | Diff (s) | % Diff | Diff (s) | % Diff | Diff (s) | % Diff |
| 1 | -0.008 | 0.915% | -0.019 | 16.307% | 0.001 | -1.085% |
| 2 | -0.069 | 4.931% | -0.020 | 12.308% | 0.053 | -29.114% |
| 3 | 0.037 | -2.525% | -0.023 | 12.238% | -0.001 | 0.309% |
| 4 | 0.288 | -14.531% | 0.005 | -1.990% | 0.043 | -19.096% |
| 5 | -0.313 | 14.091% | -0.027 | 9.291% | -0.057 | 23.218% |
| 6 | -0.032 | 1.131% | 0.011 | -3.463% | -0.077 | 32.455% |
| 7 | -0.118 | 3.465% | 0.114 | -25.741% | 0.029 | -8.918% |
| 8 | -0.116 | 3.709% | 0.077 | -17.556% | 0.001 | -0.276% |
| 9 | -0.310 | 8.555% | -0.172 | 39.611% | -0.091 | 23.218% |
| 10 | 0.228 | -5.678% | -0.054 | 13.105% | 0.001 | -0.192% |

Table 20 presents the difference in inference time between the original and dynamically quantized (dPTQ) ACR-GNN models, reported in absolute (seconds) and relative (%) terms, across various layer depths. The results show that quantization has an inconsistent effect on inference time, with no clear trend of improvement. In some configurations, dynamic quantization slightly reduces inference time; for example, layer 2 shows a 0.053s reduction on the test set, corresponding to a

29.11% improvement. Similarly, layer 5 achieves an improvement in test time of 23.22%, and layer 6 shows the largest test time speedup of 32.46%. However, in other cases, such as layer 4 in the training set (+0.288s, -14.53%) and layer 10 (+0.228s, -5.68%), quantization increases execution time. The relative differences on the validation set also vary widely, with notable slowdowns at layers 7 (–25.74%) and 9 (–39.61%). These inconsistencies highlight that run-time performance does not always benefit from dynamic quantization, and the effectiveness likely depends on the specific computation pattern and how well the underlying hardware supports quantized operations.

Table 21: Elapsed time (in seconds) for ACR-GNN with and without dynamic post-training quantization (dPTQ). PPI Benchmark

| Layer | Train | | Validation | | Test | |
|---|---|---|---|---|---|---|
| | Original | dPTQ | Original | dPTQ | Original | dPTQ |
| 1 | 0.780 | 0.858 | 0.102 | 0.112 | 0.077 | 0.094 |
| 2 | 0.986 | 0.966 | 0.130 | 0.131 | 0.109 | 0.107 |
| 3 | 1.138 | 1.161 | 0.157 | 0.159 | 0.149 | 0.140 |
| 4 | 1.371 | 1.366 | 0.159 | 0.204 | 0.156 | 0.160 |
| 5 | 1.645 | 1.682 | 0.201 | 0.211 | 0.173 | 0.199 |
| 6 | 1.833 | 1.766 | 0.242 | 0.256 | 0.188 | 0.205 |
| 7 | 2.166 | 2.156 | 0.282 | 0.261 | 0.239 | 0.242 |
| 8 | 2.355 | 2.534 | 0.317 | 0.300 | 0.241 | 0.283 |
| 9 | 2.539 | 2.652 | 0.337 | 0.349 | 0.302 | 0.292 |
| 10 | 2.842 | 3.122 | 0.386 | 0.461 | 0.326 | 0.348 |

Table 21 reports the elapsed time (in seconds) required to perform inference on the training, validation, and test sets using the ACR-GNN model with ReLU activation, both in its original form and after applying dynamic post-training quantization (dPTQ). The measurements reflect the running time of the trained models only; the time required for model training is not included in these results. The values indicate that inference time generally increases with the number of layers, as expected, and the impact of quantization on runtime varies across depths. In some cases, dPTQ slightly reduces inference time (e.g., Layer 6, Train), while in others it introduces moderate overhead, particularly for deeper models.

The experiments were run on a Samsung Galaxy Book4 laptop with an Intel Core i7-150U processor, 16 GB RAM, and 1 TB SSD storage. Additional experiments were conducted using Kaggle's cloud platform with an NVIDIA Tesla P100 GPU (16 GB RAM).

# H  Description logics with global and local cardinality constraints

The Description Logic $\mathcal{ALCSCC}^{++}$ [2] extends the basic Description Logic $\mathcal{ALC}$ [3] with concepts that capture cardinality and set constraints expressed in the quantifier-free fragment of Boolean Algebra with Presburger Arithmetic (QFBAPA) [20].

We assume that we have a set of *set variables* and a set of *integer constants*.

A QFBAPA *formula* is a Boolean combination ($\wedge$, $\vee$, $\neg$) of *set constraints* and *cardinality constraints*.

A *set term* is a Boolean combination ($\cup$, $\cap$, $\overline{\cdot}$) of *set variables*, and *set constants* $\mathcal{U}$, and $\emptyset$. If $S$ is a set term, then its cardinality $|S|$ is an *arithmetic expressions*. Integer constants are also arithmetic expressions. If $T_1$ and $T_2$ are arithmetic expressions, so is $T_1 + T_2$. If $T$ is an arithmetic expression and $c$ is an integer constant, then $c \cdot T$ is an arithmetic expression.

Given two set terms $B_1$ and $B_2$, the expressions $B_1 \subseteq B_2$ and $B_1 = B_2$ are *set constraints*. Given two arithmetic expressions $T_1$ and $T_2$, the expressions $T_1 < T_2$ and $T_1 = T_2$ are *cardinality constraints*. Given an integer constant $c$ and an arithmetic expression $T$, the expression $c \ dvd \ T$ is a *cardinality constraint*.

A *substitution* $\sigma$ assigns $\emptyset$ to the set constant $\emptyset$, a finite set $\sigma(\mathcal{U})$ to the set constant $\mathcal{U}$, and a subset of $\sigma(\mathcal{U})$ to every set variable. A substitution is first extended to set terms by applying the standard

1285  set-theoretic semantics of the Boolean operations. It is further extended to map arithmetic expressions
1286  to integers, in such that way that every integer constant $c$ is mapped to $c$, for every set term $B$, the
1287  arithmetic expression $|B|$ is mapped to the cardinality of the set $\sigma(B)$, and the standard semantics for
1288  addition and multiplication is applied.

1289  The substitution $\sigma$ *(QFBAPA) satisfies* the set constraint $B_1 \subseteq B_2$ if $\sigma(B_1) \subseteq \sigma(B_2)$, the set
1290  constraint $B_1 = B_2$ if $\sigma(B_1) = \sigma(B_2)$, the cardinality constraint $T_1 < T_2$ if $\sigma(T_1) < \sigma(T_2)$, the
1291  cardinality constraint $T_1 = T_2$ if $\sigma(T_1) = \sigma(T_2)$, and the cardinality constraint $c \; dvd \; T$ if $c$ divides
1292  $\sigma(T)$.

1293  We can now define the syntax of $\mathcal{ALCSCC}^{++}$ concept descriptions and their semantics. Let $N_C$ be
1294  a set of concept names, and $N_R$ be a set of role names, such that $N_C \cap N_R = \emptyset$. Every $A \in N_C$
1295  is a *concept description* of $\mathcal{ALCSCC}^{++}$. Moreover, if $C, C_1, C_2, \ldots$ are *concept descriptions* of
1296  $\mathcal{ALCSCC}^{++}$, then so are: $C_1 \sqcap C_2, C_1 \sqcup C_2, \neg C$, and $\mathsf{sat}(\chi)$, where $\chi$ is a set or cardinality QFBAPA
1297  constraint, with elements of $N_R$ and concept descriptions $C_1, C_2, \ldots$ used in place of set variables.

1298  A *finite interpretation* is a pair $I = (\Delta^I, \cdot^I)$, where $\Delta^I$ is a finite non-empty set of individuals, and
1299  $\cdot^I$ is a function such that: every $A \in N_C$ is mapped to $A^I \subseteq \Delta^I$, and every $R \in N_R$ is mapped to
1300  $R^I \subseteq \Delta^I \times \Delta^I$. Given an element of $d \in \Delta^I$, we define $R^I(d) = \{d' \mid (d, d') \in R^I\}$.

1301  The semantics of the language of $\mathcal{ALCSCC}^{++}$ makes use QFBAPA substitutions to interpret QFBAPA
1302  constraints in terms of $\mathcal{ALCSCC}^{++}$ finite interpretations. Given an element $d \in \Delta^I$, we can define
1303  the substitution $\sigma_d^I$ in such a way that: $\sigma_d^I(\mathcal{U}) = \Delta^I$, $\sigma_d^I(\emptyset) = \emptyset$, and $A \in N_C$ and $R \in N_R$ are
1304  considered QFBAPA set variables and substituted as $\sigma_d^I(A) = A^I$, and $\sigma_d^I(R) = R^I(d)$.

1305  The finite interpretation $I$ and the QFBAPA substitutions $\sigma_d^I$ are mutually extended to complex ex-
1306  pressions such that: $\sigma_d^I(C_1 \sqcap C_2) = (C_1 \sqcap C_2)^I = C_1^I \cap C_2^I$; $\sigma_d^I(C_1 \sqcup C_2) = (C_1 \sqcup C_2)^I =$
1307  $C_1^I \cup C_2^I$; $\sigma_d^I(\neg C) = (\neg C)^I = \Delta^I \setminus C^I$; and $\sigma_d^I(\mathsf{sat}(\chi)) = (\mathsf{sat}(\chi))^I = \{d' \in \Delta^I \mid$
1308  $\sigma_{d'}^I$ (QFBAPA) satisfies $\chi\}$.

1309  **Definition 24.** *The $\mathcal{ALCSCC}^{++}$ concept description $C$ is* satisfiable *if there is a finite interpretation*
1310  *$I$ such that $C^I \neq \emptyset$.*

1311  **Theorem 25** ([2]). *The problem of deciding whether an $\mathcal{ALCSCC}^{++}$ concept description is satisfiable*
1312  *is NEXPTIME-complete.*

# I   $\mathcal{ALCQ}$ and $T_C$Boxes consistency

1314  $\mathcal{ALCQ}$ is the Description Logic adding qualified number restrictions to the standard Description
1315  Logic $\mathcal{ALC}$, analogously to how Graded Modal Logic extends standard Modal Logic with graded
1316  modalities.

1317  Let $N_C$ and $N_R$ be two non-intersecting sets of concept names, and role names respecivey. A
1318  concept name $A \in N_C$ is an $\mathcal{ALCQ}$ concept expressions of $\mathcal{ALCQ}$. If $C$ is an $\mathcal{ALCQ}$ concept
1319  expression, so is $\neg C$. If $C_1$ and $C_2$ are $\mathcal{ALCQ}$ concept expressions, then so is $C_1 \sqcap C_2$. If $C$ is an
1320  $\mathcal{ALCQ}$ concept expression, $R \in N_R$, and $n \in \mathbb{N}$, then $\geq n \; R.C$ is an $\mathcal{ALCQ}$ concept expression.

1321  A *cardinality restriction* of $\mathcal{ALCQ}$ is is an expression of the form $(\geq n \; C)$ or $(\leq n \; C)$, where $C$ an
1322  $\mathcal{ALCQ}$ concept expression and $n \in \mathbb{N}$.

1323  An $\mathcal{ALCQ}$-$T_C$Box is a finite set of cardinality restrictions.

1324  An *interpretation* is a pair $I = (\Delta^I, \cdot^I)$, where $\Delta^I$ is a non-empty set of individuals, and $\cdot^I$ is
1325  a function such that: every $A \in N_C$ is mapped to $A^I \subseteq \Delta^I$, and every $R \in N_R$ is mapped
1326  to $R^I \subseteq \Delta^I \times \Delta^I$. Given an element of $d \in \Delta^I$, we define $R^I(d) = \{d' \mid (d, d') \in R^I\}$.
1327  An interpretation $I$ is extended to complex concept descriptions as follows: $(\neg C)^I = \Delta^I \setminus C^I$;
1328  $(C_1 \sqcap C_2)^I = C_1^I \cap C_2^I$; and $(\geq n \; R.C)^I = \{d \mid |R^I(d) \cap C^I| \geq n\}$.

1329  An interpretation $I$ satisfies the cardinality restriction $(\geq n \; C)$ iff $|C^I| \geq n$ and it satisfies
1330  the cardinality restriction $(\leq n \; C)$ iff $|C^I| \leq n$. A $T_C$Box $TC$ is *consistent* if there exists an
1331  interpretation that satisfies all the cardinality restrictions in $TC$.

1332  **Theorem 26** ([36]). *Deciding the consistency of $\mathcal{ALCQ}$-$T_C$Boxes is NEXPTIME-hard.*

1333  The proof can be slightly adapted to show that the result holds even when there is only one role.

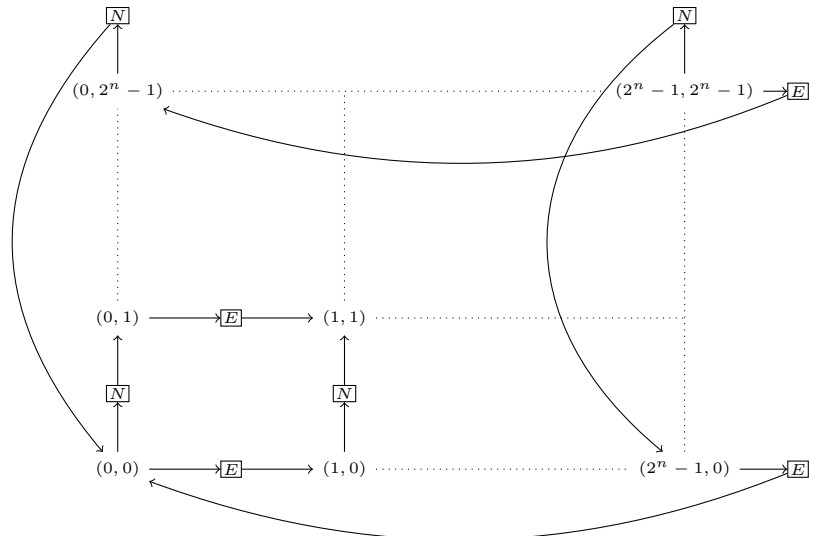

Figure 8: Encoding a torus of exponential size with an $\mathcal{ALCQ}\text{-}T_C$Box with one role.

Some abbreviations are useful. For every pair of concepts $C$ and $D$, $C \to D$ stands for $\neg C \sqcup D$. For every concept $C$, role $R$, and non-negative integer $n$, we define: $(\leq n\ R.C) := \neg(\geq (n+1)\ R.C)$, $(\forall\ R.C) := (\leq 0\ R.\neg C)$, $(\forall\ C) := (\leq 0\ \neg C)$, $(= n\ R.C) := (\geq n\ R.C) \sqcap (\leq n\ R.C)$, and $(= n\ C) := (\geq n\ C) \sqcap (\leq n\ C)$.

**Theorem 27.** *Deciding the consistency of $\mathcal{ALCQ}\text{-}T_C$Boxes is NEXPTIME-hard even if $|N_R| = 1$.*

*Proof.* Let $next$ be the unique role in $N_R$. We use the atomic concepts $N$ to denote an individual 'on the way north' and $E$ to denote an individual 'on the way east'. See Figure 8.

For every $n \in \mathbb{N}$, we define the following $\mathcal{ALCQ}\text{-}T_C$Box.

$$
\begin{aligned}
T_n = \{\ & (\forall\ \neg(N \sqcup E) \to (= 1\ next.N))\ , & & (\forall\ \neg(N \sqcup E) \to (= 1\ next.E)) \\
& (\forall\ N \to (= 1\ next.\top))\ , & & (\forall\ E \to (= 1\ next.\top)) \\
& (= 1\ C_{(0,0)})\ , & & (= 1\ C_{(2^n-1,2^n-1)}) \\
& (\forall\ \neg(N \sqcup E) \to D_{east})\ , & & (\forall\ \neg(N \sqcup E) \to D_{north}) \\
& (\leq\ (2^n \times 2^n)\ \neg(N \sqcup E)), & (\leq\ (2^n \times 2^n)\ N), & (\leq\ (2^n \times 2^n)\ E)\ \}
\end{aligned}
$$

such that the concepts $C_{(0,0)}$, $C_{(2^n-1,2^n-1)}$ are defined like in [36, Figure 3], and so are the concepts $D_{north}$ and $D_{east}$, except that for every concept $C$, $\forall east.C$ now stands for $\forall next.(E \to \forall next.C)$ and $\forall north.C$ now stands for $\forall next.(N \to \forall next.C)$.

The problem of deciding whether a domino system $\mathcal{D} = (D, V, H)$, given an initial condition $w_0 \dots w_{n-1}$, can tile a torus of exponential size can be reduced to the problem of consistency of $\mathcal{ALCQ}\text{-}T_C$Boxes, checking the consistency of $T(n, \mathcal{D}, w) = T_n \cup T_\mathcal{D} \cup T_w$, where $T_n$ is as above, $T_\mathcal{D}$ encodes the domino system, and $T_w$ encodes the initial condition as follows.

$$
\begin{aligned}
T_\mathcal{D} = \{\ & (\forall\ \neg(N \sqcup E) \to (\textstyle\bigsqcup_{d \in D} C_d)), \\
& (\forall\ \neg(N \sqcup E) \to (\textstyle\bigsqcap_{d \in D} \bigsqcap_{d' \in D \setminus \{d\}} \neg(C_d \sqcap C_{d'}))), \\
& (\forall\ \textstyle\bigsqcap_{d \in D}(C_d \to (\forall east.\bigsqcup_{(d,d') \in H} C_{d'}))), \\
& (\forall\ \textstyle\bigsqcap_{d \in D}(C_d \to (\forall north.\bigsqcup_{(d,d') \in V} C_{d'})))\ \}
\end{aligned}
$$

$$
T_w = \{\ (\forall\ C_{(0,0)} \to C_{w_0}), \dots, (\forall\ C_{(n-1,0)} \to C_{w_{n-1}})\ \}
$$

The rest of the proof remains unchanged. $\qquad\square$

