# OpenReview forum: "On the Complexity of Verifying Quantized GNNs with Readout"
_NeurIPS.cc/2025/Conference — Submitted to NeurIPS 2025_

### Official Review · Reviewer_153V · 2025-06-01

**Clarity:** 2
**Significance:** 2
**Originality:** 3
**Rating:** 4
**Confidence:** 2

**Summary:**

This work studies the "verification" of GNNs with global readouts. It proposes a logic to determine the satisfiability of a formula involving such GNNs, and prove that the proposed logic is NEXPTIME, i.e., it can be determined by a non-deterministic exponential-time algorithm. Given this impractical result, this work impose a further constraint on the size of the problem via limiting the number of vertices.

**Questions:**

See weakness.

**Ethical Concerns:**

["NO or VERY MINOR ethics concerns only"]

**Final Justification:**

Verification problem definition and significance have been resolved.

**Limitations:**

There are no ethical concerns.

Regarding the limitation of this work, I do not see why the studied problem is relevant.

**Paper Formatting Concerns:**

No.

**Quality:**

2

**Strengths And Weaknesses:**

Strength:
- Many examples to illustrate the idea.
- There is numerical experiment supporting the reasoning.

Weakness:
- The studied problem: From the definition in the paper (correct me if I am wrong), "verification" refers to deciding the satisfiability, i.e., determining whether there exists a node in the GNN that satisfies the given formula. I never saw such "verification" problem before, and do not think this problem is meaningful. Further, the only prior work [32] that this work built upon is UNPUBLISHED and submitted to arxiv THREE MONTHS ago.
- Contribution: The authors propose a new logic to verify GNNs, but it turns out that the complexity of this logic is prohibitive. This neither proves that the new logic is useful nor proves that the problem is intrinsically hard. The authors should explain clearly what their result implies.
- Experiment: After checking the code "src_verificationtool", it seems that the verification aims to verify properties of certain attributes computed by the GNN. Shouldn't this be directly accessible by doing inference on the GNN? The authors should define the specification more clearly.

---

> ### Author Rebuttal · Authors · 2025-07-30
>
> Thank you for reviewing the paper!
>
> ---
>
> ### Strengths
>
> We thank you for your appreciation of our examples and experiments.
>
> ## Weaknesses
>
> > I never saw such "verification" problem before, and do not think this problem is meaningful. Further, the only prior work [32] that this work built upon is UNPUBLISHED and submitted to arxiv THREE MONTHS ago.
>
> In this paper "verification" is about solving the verification tasks VT1-3 (line 132-134). We may expand the description of these problems more. These problems actually are not new and are also presented (not for quantized GNNs) in [24]. As far as we know, it seems that [32] is going to appear at IJCAI 2025 (see the list of accepted papers on the official website). Our work is not built upon [32]. Note that we do not use any result from [32]. We only use the idea of the language qL in [32].
>
> > The authors propose a new logic to verify GNNs, but it turns out that the complexity of this logic is prohibitive. This neither proves that the new logic is useful nor proves that the problem is intrinsically hard. The authors should explain clearly what their result implies.
>
> Theorem 12 says that it is not only the satisfiability problem of qL that is NEXPTIME-hard, but also the verification tasks! The logic is not simply a formal tool with which reasoning is highly inefficient. It is a tool to establish that the GNN verification problems (VT1, VT2, VT3) in presence of readout are intrinsically hard.
>
> > After checking the code "src_verificationtool", it seems that the verification aims to verify properties of certain attributes computed by the GNN. Shouldn't this be directly accessible by doing inference on the GNN?
>
> The verification tasks are exactly doing that. We only make inference on the GNN (provided the bound N). We do not take a given graph as an input. The specification of the program is given in the paper: the ACR-GNN satisfiability problem with a bound on the number of vertices (l. 279-281). It is also stated in the README.md.
>
> ---
>
> Does the reviewer find our answer useful? We would be very happy to provide additional clarifications at the reviewer's request.

---

> ### Comment · Reviewer_153V · 2025-08-01
>
> Dear authors,
>
> Thanks for the reply. I found the content very helpful.
>
> 1. After reexamining the corresponding lines and definitions, currently my understanding is: the verification aims at verifying (VT1) whether satisfying $\phi$ implies a positive class (cls=1), (VT2) whether a positive class implies $\phi$ is satisfied and (VT3) whether there are some directed graphs that has a positive class and satisfies $\phi$. Is this correct? May you point out the formal definition of $[[\phi]]$? Could you interpret your verification problem in plain language in the revision?
>
> 2. Could you clarify how getting complexity results based on $qL$ implies the complexity of the problem itself?
>
> 3. Thanks for the clarification. This is resolved.
>
> 4. (Followup) given the complexity results, could the authors comment on the practical implications of their results? It seems that the results extends the traditional complexity results in NN verification, where the general problem is usually coNP-hard or harder.
>
> My current evaluation for this work is 3.

---

> > ### Author Response · Authors · 2025-08-02
> > **Response to Official Comment by Reviewer 153V**
> >
> > Thank you for the prompt reply!
> >
> > 1. Your understanding is perfect. Yes, the formal definition of [[phi]] is given l. 128 if the logic is qL itself. If formula phi is a modal logic formula, then we may first translate in qL and use Lemma 3.
> >
> > 2. For the lower bound, the full argument of why the GNN verification problems are inherently hard, is given in Appendix, p. 17, l. 630-642.
> > - For VT1, if the formula phi is \top (true), it is already coNEXPTIME-hard.
> > - VT2 and VT3 it is sufficient for the formula \phi to be a formula in graded modal logic + graded universal modality.
> >
> > We acknowledge that the proof of Theorem 4 (upper bound) does not explicitly explain why VT1, VT2 are in coNEXPTIME and VT3 in NEXPTIME. The reductions from the GNN verification problems to the problem of (non-)satifisfiability in qL are simple. We will add the following explanations in Appendix:
> > - For VT1:
> >   * compute mod2expr(phi) via Lemma 3
> >   * construct the qL formula: mod2expr(phi) --> expression(A)
> >   * check that mod2expr(phi) --> expression(A) is valid (true in all pointed directed graphs), i.e. checking that not(mod2expr(phi) --> expression(A)) is not satisfiable (the "not" in "not satisfiable makes it in *co-*NEXPTIME).
> > - For VT2: similar but with qL formula expression(A) --> mod2expr(phi)
> > - For VT3:
> >   * compute "mod2expr(phi) and expression(A)"
> >   * check that mod2expr(phi) and expression(A) is satisfiable
> >
> > 3. OK
> >
> > 4. Yes, the traditional verification of NN is NP-complete or coNP-complete (depending on the exact verification task). It is because the size of the input is known in advance. In the quantized case it is essentially "guessing values of the input" + "guessing the behaviors of activation functions at each neuron, e.g. negative => 0 or positive => keeping the value for ReLU). In the idealistic case (let say real numbers), it is essentially guessing the behaviors of activation functions and reducing to Linear Programming.
> >
> > With GNN, the size of the input (the directed graph) is arbitrarily large. So we enter the world of "modal logic". The satisfiability problem of standard modal logic (e.g., logic K) is typically PSPACE-complete. Here, we are close to graded modal logic + graded universal modality (corresponding to global readout), and its satisfiability problem is NEXPTIME-complete.

---

> > > ### Comment · Reviewer_153V · 2025-08-04
> > >
> > > Dear authors,
> > >
> > > Thanks for the reply. This clears my current concerns. I decided to raise my score, but remain unconfident about my evaluation given the involved topic and my limited expertise in this topic.

---

### Official Review · Reviewer_TCR3 · 2025-07-03

**Clarity:** 3
**Significance:** 2
**Originality:** 2
**Rating:** 4
**Confidence:** 3

**Summary:**

This paper introduce a modal-style logic capable of expressing both the computation of quantized aggregate-combine-readout GNNs (ACR-GNNs). They prove that verifying such networks is NEXPTIME-complete while implementing a prototype verifier that translates the bounded problem to SMT-based model checking. Experimental results show quantization incurs only negligible accuracy loss while sharply reducing model size and inference cost, confirming the practicality of quantized ACR-GNNs in resource-constrained settings.

**Questions:**

1. How does the prototype verifier scale as N grows beyond a few hundred vertices or when hidden dimensions increase?

2. How do you select a safe yet tractable N for bounded verification in real applications?

3. What would be the complexity impact of supporting other activations (e.g., GELU and sigmoid) ?

**Ethical Concerns:**

["NO or VERY MINOR ethics concerns only"]

**Final Justification:**

The authors propose a new logic for verifying GNNs. As I have limited expertise in this area and am not familiar with much prior work in this domain, my current rating is primarily based on the theoretical contribution. However, since the problem remains NEXPTIME-complete, I believe that this work should be more appropriately positioned within the bounded setting.

**Limitations:**

Yes

**Quality:**

3

**Strengths And Weaknesses:**

**Strengths**

1. This paper introduces a novel logic language which is proven to capture quantized ACR-GNNs with arbitrary activation functions.
2. The authors have provided tight complexity results, closing an open question for truncated-ReLU GNNs.
3. Quantization experiments demonstrate sizable compression with negligible accuracy loss.

 **Weaknesses**

1. Full verification remains NEXPTIME, and even the bounded prototype may struggle on realistic graph sizes.
2. Limited experimental results, namely, only one synthetic benchmark and one real dataset.

---

> ### Author Rebuttal · Authors · 2025-07-30
>
> Thanks for your thoughtful review!
>
> ---
>
> ### Strengths
> Thank you. We agree that our work is a crucial step in better understanding of global readout.
>
> ### Weaknesses
> 1. Yes, our paper is mainly theoretical. "The full verification remains NEXPTIME": actually we and all other researchers have to face the reality: the problem is NEXPTIME-complete! Yes section 6 is a hope for having more practical work on bounded verification.
> 2. Yes, this is because of some limitations and restrictions to the real-world dataset, not a lot of them contracted as a set of graphs with a limited number of nodes.
>
> ## Questions
>
> > How does the prototype verifier scale as N grows beyond a few hundred vertices or when hidden dimensions increase?
>
> Currently, not well. The implementation is naïve. To get an idea, on a very small GNN (three features, 3 layers of three internal features each), we get the following time to perform the verification for different values of N:
> - N = 1: 0.08997130393981934s
> - N = 2: 0.10367822647094727s
> - N = 3: 0.845531702041626s
> - N = 4: 2.5763635635375977s
> - N = 5: 10.406831502914429s
> - N = 6: 32.66759514808655s
>
> Said differently, our work provides non-trivial benchmarks for the verification communauty (people working on model checkers, SMT solvers, etc.).
>
>
> > How do you select a safe yet tractable N for bounded verification in real applications?
>
> In real applications, e.g. chemistry, we may have a natural bound N. We could then certify that the GNN complies with some rule (e.g. having a Uranium atom in the molecule implies that the molecule is classified as toxic) for all molecules containing less than N atoms. So N depends on the application.
>
>
>
> > What would be the complexity impact of supporting other activations (e.g., GELU and sigmoid)?
>
> That is a good question! As long as the functions can be computed in exponential time (wrt to bitwidth), the verification task is still in NEXPTIME.
> NEXPTIME-hardness is however not guaranteed. In this article, we made experiments and compared the influence of the activation function only for the cases ReLu and trReLU. The impact of other activation functions can be explored in future work.
>
> ---
>
> We will gladly answer further questions if the reviewer needs more clarifications.

---

> > ### Comment · Reviewer_TCR3 · 2025-08-05
> >
> > Thanks for your response.
> >
> > My last question is:  According to your N=1..6 results, could we say that the time to perform the verification is linear to the values of N?

---

> > > ### Author Response · Authors · 2025-08-06
> > >
> > > Thank you for engaging in a discussion and that follow-up question!
> > >
> > > ---
> > >
> > > No, the complexity is not linear in N, but exponential. Here’s why:
> > >
> > > The model checker ESBMC verifies the C program that encodes the execution of the given GNN on an arbitrary graph of size N. Internally, ESBMC translates this C program into an SMT formula whose variables include:
> > >
> > > - Boolean variables $E_{ji}$ for each $i, j \in {0, …, N−1}$, indicating whether vertex $i$ is connected to vertex $j$.
> > > - The $n$-bit values of all (input and intermediate) features for each vertex $i \in {0, …, N−1}$.
> > >
> > > Conceptually, this SMT formula can be viewed as a Boolean formula with $N^2 + N · |\varphi|$ Boolean variables.
> > > In the worst case, the solver may need to consider all $2^{(N^2 + N · |\varphi|)}$
> > > possible assignments, which grows exponentially with N. Therefore, the worst-case runtime is exponential in N, not linear.

---

### Official Review · Reviewer_jCNx · 2025-07-04

**Clarity:** 3
**Significance:** 3
**Originality:** 3
**Rating:** 4
**Confidence:** 3

**Summary:**

This paper investigates the formal verification of quantized graph neural networks (GNNs) that use a global readout operation—commonly used for graph-level tasks. The authors introduce a logic named qL, capable of expressing the computation of Aggregate-Combine-Readout GNNs (ACR-GNNs) as well as rich specifications. They prove that satisfiability and verification in this logic is NEXPTIME-complete, even for truncated ReLU activations and bounded-bit quantization. For bounded graph sizes, the problems reduce to NP-complete, and the authors implement an SMT-based prototype using ESBMC. A series of experiments show that post-training quantization yields good model compression with minimal accuracy degradation.

**Questions:**

* How large a graph and GNN can your verifier handle in practice? Have you benchmarked runtime or memory consumption?
* Your quantization study focuses on accuracy and size. How does quantization affect robustness or verifiability?
* Are there structural restrictions (e.g., treewidth, layer count) under which verification becomes feasible despite worst-case hardness?
* Can qL express temporal or fairness-related properties? Would extensions (e.g., temporal qL) preserve decidability?

**Ethical Concerns:**

["NO or VERY MINOR ethics concerns only"]

**Final Justification:**

No concrete technical flaws were identified; the results appear potentially novel and valuable if correct. Given the absence of strong objections and the possible significance of the contribution, I recommend borderline accept.

**Limitations:**

Yes.

**Paper Formatting Concerns:**

N.A.

**Quality:**

3

**Strengths And Weaknesses:**

**Strengths:**

* The paper rigorously proves tight complexity bounds for satisfiability and verification of ACR-GNNs under quantization and global readout, using reductions from established problems.
* The proposed logic elegantly extends previous logics to support global aggregation and quantized semantics.
* The ESBMC-based bounded verifier shows the method is implementable, and the quantization experiments validate the practicality of ACR-GNNs.

**Weaknesses:**

* Although the logic and complexity are well analyzed, the prototype verifier is evaluated only on synthetic cases. No real-world verification tasks or practical properties (e.g., fairness, safety) are tested.
* Some of the key hardness reductions (e.g., from Wang tiling) are technically sound but underexplained. Visuals or concrete examples would help readers unfamiliar with such reductions.
* While qL is expressive, it would be valuable to discuss what kinds of properties cannot be captured or how the logic compares in succinctness to FOC$_2$ or modal logics.

---

> ### Author Rebuttal · Authors · 2025-07-30
>
> Thank you for the review and intriguing questions!
>
> ---
>
> ### Strengths
>
> We concur that verifying ACR-GNNs is a challenging task, and now that it is established as NEXPTIME-complete, this is even clearer.
>
>
> ### Weaknesses
>
> - No, we plan to address that in future work.
> - Agreed. We aim to include figures in the Appendix to help readers better understand the tiling problem.
> - Very interesting! We plan to look at this in the future, thanks for the ideas!
>
> ## Questions
>
> > How large a graph and GNN can your verifier handle in practice? Have you benchmarked runtime or memory consumption?
>
> Our work is mainly theoretical. That is why we did not concentrate our work on it. The tool we propose is still a naïve prototype that directly transforms our instance as a C program to verify by the model checker ESBMC. Efficient encoding into Satisfiability Modulo Theory (SMT) is a research area of its own, and an efficient encoding of GNN verification tasks would be a contribution of its own. To get an idea, on a very small GNN (three features, 3 layers of three internal features each), we get the following time to perform the verification for different values of N:
> - N = 1: 0.08997130393981934s
> - N = 2: 0.10367822647094727s
> - N = 3: 0.845531702041626s
> - N = 4: 2.5763635635375977s
> - N = 5: 10.406831502914429s
> - N = 6: 32.66759514808655s
>
> Said differently, our work provides non-trivial benchmarks for the verification community (people working on model checkers, SMT solvers, etc.).
>
> > Your quantization study focuses on accuracy and size. How does quantization affect robustness or verifiability?
>
> Yes, in this paper we focused more on the impact of quantization on the accuracy, size, and latency of the model. We do not know how quantization affects robustness. It will be an interesting line of future work.
>
> In terms of verifiability, observe that the verification of GNNs is exponential in the bit-width of the parameters. It is a consequence of our complexity results. (See for instance l. 266 p. 7.) We will make this more explicit. So quantization has in theory a very positive effect on the complexity of verification.
>
> > Are there structural restrictions (e.g., treewidth, layer count) under which verification becomes feasible despite worst-case hardness?
>
> Very good question. The answer is that we do not know. However, your question is highly relevant since for the encoding we use a 2n x 2n-grid which is of treewidth 2^n (which is a lot).
>
> > Can qL express temporal or fairness-related properties? Would extensions (e.g., temporal qL) preserve decidability?
>
> Currently no. However, qL could be extended with temporal operators, and hopefully capture some aspects of GNNs applied to temporal graphs. This remains to be investigated.

---

> ### Comment · Reviewer_jCNx · 2025-08-09
>
> Thank the authors for the detailed response and clarifications for my questions. I remain positive on the paper's ideas and approach, and so will maintain my positive rating.

---

### Official Review · Reviewer_YCJJ · 2025-07-04

**Clarity:** 2
**Significance:** 2
**Originality:** 4
**Rating:** 4
**Confidence:** 2

**Summary:**

In this article, the authors present new results about the logical expressivity of a variant of $\textit{quantized}$ ReLU GNNs, meaning that the numbers used in computations all have a finite number of bits. The main contribution of the authors is to design a certain logic called $q\mathcal{L}$, to capture global readout.

The main contributions are:

- The logic qL is NEXPTIME, meaning that given, the language associated to qL, then recognizing if a formula based on this language can be satisfied, can be performed in  $O(2^{p(n)})$ with a non-deterministic Turing machine, for some polynomial $p$ (n is a parameter measuring the size of the formula).

- If one restricts to the version of GNNs without global readout, adapting qL to that case yields a PSPACE-complete version, meaning that (i) it is in PSPACE: one can solve the same problem as above in $\textit{space}$ that is polynomial with respect to the input (formula) size, and (ii) any problem  in PSPACE can be reduced polynomially to a problem of satisfiability of a formula of the qL version without global readout.

- Preliminary experiments on the impact of quantization on the performance and model size in practice. The quantized version seems to gain performance for generalization.

**Questions:**

- In the , characterizing the modal flavor of qL for other activation functions than ReLU. New extensions of qL could be proposed to tackle other classes GNNs.

- The authors in the second page refer to trRELU, I suppose this is a typo and they meant ReLU (?)

- The authors line 41 that ``qL is expressive enough to capture quantized ACR-GNNs with arbitrary activation functions.'' However in their discussion page 9, they mention that characterizing the modal flavor of qL for other activation functions than ReLU is a path for future work. Do the author refer to existing results, for example of [4], or they imply that they provided additional results for ReLU GNNs, lines 144-155? If this is the case, this would deserve to be stated in a formal statement.

- Section 5: Complexity Lower Bound: is the Wang tiles problem (domino problem mentioned in the appendix) undecidable? Is it what is meant by ``lower bound'', i.e. it is not in NP?

**Ethical Concerns:**

["NO or VERY MINOR ethics concerns only"]

**Final Justification:**

REBUTTAL -

I am satisfied with the authors' depth and clarity in their answers about the core of their contributions. In particular, I find the NEXPTIME-hardness interesting and valuable. Therefore, I have updated my score to a (borderline) accept. However, I have not checked all the details of the proofs and I am not entirely sure about soundness, so I prefer to keep my confidence score as low.

**Limitations:**

Yes, the authors included a paragraph after the discussion section about some limitations of their theoretical contributions. I suggest they also mention the limitations of their experiments as mentioned above.

**Quality:**

3

**Strengths And Weaknesses:**

Strengths

The contributions of the article are clearly stated, and the results are contribution to the Theory of GNN expressivity.
These are steps towards a more complete understanding of the role of readout, and the impact of quantization on GNNs.


Weaknesses

(i) This article can be very interesting for logicians, and has the potential to be relevant for a more general audience, however I feel that the effort put in the paper to present the necessary notions (Background section) should have been developed more, so that it becomes readable for researchers interested in the expressivity of GNNs (this could have been done at the cost of moving other sections to the appendix, for example the reduction Section 4.3 could be summarized, and the full reduction left to the appendix).

(ii) For the experimental part, despite the appealing result, it is difficult to draw meaningful and statistically significant conclusions using only a few thousand of graphs. It would have been interesting to test whether this gain in generalization performance remains for different distributions and say, hundreds of thousands or millions of graphs for each distribution (of reasonable size).

I would be willing to increase my score based on the authors' answers to my questions below as well as comments on the point (i) above.

---

> ### Author Rebuttal · Authors · 2025-07-30
>
> Thank you for your thoughtful review and the in-depth questions you asked.
>
> ---
>
> ### Contribution & Strengths
>
> - Importantly, we also establish NEXPTIME-hardness, indicating that the upper bound NEXPTIME is not merely a preliminary result. In other words, we illustrate that verifying ACR-GNNs can be executed in NEXPTIME, and it is intrinsically very challenging: no exact solution exists that can perform better than nondeterministic exponential time. Practitioners must be aware of this.
> - Given this intense complexity, we also highlight that some hope for verifying ACR-GNNs in practice might arise from the verification of ACR-GNNs by increasing the number of vertices in the counterexamples (Section 6).
>
> Thanks for pointing out one of the key ares of our work touches on: understanding what GNNs, if using global readout but limited by an actually implemented arithmetic, can express!
>
> ### Weaknesses
>
> (i) Thank you for the suggestion. If the reviewers/ACs think that Section 4.3 should go in Appendix, we can expand more the background section, and section 3. For instance, we could explain more the verification tasks in words (line 132-134).
>
> (ii) We thank the reviewer for highlighting the need for larger‐scale and distributionally diverse evaluations. Observe that we are interested in node classification. Therefore, the number of nodes, and not the number of graphs, is the relevant measure for the number of data-points.
>
> In terms of the real-world datasets, almost every graph dataset for node classification is made of only one large graph. The real-world PPI benchmark is chosen in the paper specifically because it is an exception. In fact, we are not aware of any other such dataset. We would be very glad to hear about other suitable real-world data.
>
> At the moment, we rely on synthetic data.
> To generate the synthetic data, we employed the Erdős-Rényi model. As we mentioned in our paper, we bounded the number of nodes. If we use this model, we can control both the number of nodes and the number of edges. In this case, we can generate many graph instances with predictable average size, density, and connectivity. Another key aspect is that if we fix the seed, we obtain reproducible data, because independent edge sampling makes it trivial to share or reconstruct the exact same graph corpus.
>
>
>
> ## Questions
>
> > In the , characterizing the modal flavor of qL for other activation functions than ReLU. New extensions of qL could be proposed to tackle other classes GNNs.
>
> Sure, qL can easily be adapted for other activation functions.
>
> > The authors in the second page refer to trRELU, I suppose this is a typo and they meant ReLU (?)
>
> trReLU stands for "truncated ReLU," defined as trReLU(x) = max(0, min(1, x)). It is sometimes referred to as "heaviside ReLU." You are correct, we first define it on page 4 (l. 137). We will define it before mentioning it in a revised version of the paper.
>
> > The authors line 41 that ``qL is expressive enough to capture quantized ACR-GNNs with arbitrary activation functions.'' However in their discussion page 9, they mention that characterizing the modal flavor of qL for other activation functions than ReLU is a path for future work. Do the author refer to existing results, for example of [4], or they imply that they provided additional results for ReLU GNNs, lines 144-155? If this is the case, this would deserve to be stated in a formal statement.
>
> Line 41 discusses how qL captures ACR-GNNs, which is acceptable. The note on Page 9 about "First, characterizing the modal flavour of qL for other activation functions than ReLU" refers to exploring the connections between existing modal logics and qL. We currently do not possess the characterisations for other activation functions like sigmoid or others. This represents a direction for future work.
>
> > Section 5: Complexity Lower Bound: is the Wang tiles problem (domino problem mentioned in the appendix) undecidable? Is it what is meant by ``lower bound'', i.e. it is not in NP?
>
> There are different Wang tile problems. While determining if the entire plane can be tiled is undecidable, here we focus on tiling a square torus of size 2^n x 2^n, where n is provided in unary. This problem is NEXPTIME-hard. The lower bound remains NEXPTIME-hard. Yes, NP is included in NEXPTIME, and NEXPTIME is known to be distinct from NP, making our problem not just outside NP but significantly more challenging than problems in NP.
>
> ### Limitations
>
> The experiments are presented to justify the theoretical models. We establish that the quantized ACR-GNNs models can be verified in theory. Although it comes at a prohibitive computational cost, a pressing question remains: did we make the models too simple?
>
> We believe that the extent of our experiments provide solid evidence that the models are in fact of practical interest. The reviewer is right, however, that the limitations of the experiments should be stated.
>
> 1. We use Barcelò et al. ICLR 2020 as a baseline. We are interested in comparing the performances before and after quantization.
> 2. Lack of relevant real-world benchmarks.
> Lack of multi-graph benchmarks. Most publicly available node-classification datasets consist of a single large graph (e.g., Cora, PubMed), rather than collections of smaller graphs. This limits our ability, but also the ability of practicioners using GNNs for node classification in general, to evaluate quantization effects across diverse graph instances.
> 4. Synthetic vs. real-world data. Our reliance on Erdős–Rényi random graphs provides controlled conditions for quantization analysis, but does not capture the community structure or feature correlations present in real networks. Future work using quantized ACR-GNNs should validate on domain-specific benchmarks (social, citation, and molecular graphs).
> 5. Quantization techniques.
> We applied only post-training dynamic quantization from Float32 to Int8. In future work, we are planning to add more techniques and compare their influence on the accuracy, size, latency, and robustness of the model.
>
> ---
>
> We will gladly answer other questions if the reviewer needs further clarifications!

---

### Official Review · Reviewer_j9GH · 2025-07-13

**Clarity:** 2
**Significance:** 3
**Originality:** 3
**Rating:** 4
**Confidence:** 2

**Summary:**

The paper introduces a language qL for specifying and verifying properties of ACR-GNNs. The authors show several interesting and key properties for this language, namely that adding modal operators does not add expressivity in terms of what can be specified, the satisfiability problem in modal qL is NEXPTIME, and when you restrict the number of vertices, the problem is NP-Complete, which is more tractable. The paper also empirically demonstrates that ACR-GNNs can be quantized post-training while preserving most of their performance, pointing towards quantized ACR-GNNs being potentially useful in applications where computational costs are a bottleneck.

**Questions:**

Addressing the weaknesses mentioned above would significantly strengthen the paper. I have a few more smaller comments/questions:
1. In Section 4.3, could you formally define the different sets and variables? X_H, H-vertices, S_H and so on?
2. It would be nice to add informal proof sketches for the key results where possible. I haven't checked the proofs in the appendix in detail, and an informal proof describing the key ideas/approach would help verify quickly.
3. There are a few example graphs and specifications in the paper, but I think it may help to extend those to Section 4 in the form of a running example.

**Ethical Concerns:**

["NO or VERY MINOR ethics concerns only"]

**Final Justification:**

In light of the author's rebuttal and the initial preliminary empirical results, the paper is above the bar for acceptance but given the terse nature of the manuscript and the lack of sufficient empirical evidence regarding the practicality of the proposed approach, I recommend a weak accept.

**Limitations:**

yes

**Quality:**

3

**Strengths And Weaknesses:**

Strong:
1. The paper looks at verification for quantized ACR-GNNs. One of the requirements for unlocking the use of quantizated ACR-GNNs (besides their performance) in various applications is the ability to guarantee useful properties over them. This paper looks at this problem, and proves interesting properties about the problem -- the results are interesting and useful for understanding the tractability of various specifications. They also provide a subclass of problems where verification is more tractable.

The theoretical framework and the analysis provided in this paper should provide a useful tutorial value to those working on verifying quantized GNNs, and also valuable first steps for others to build on.

Weaknesses:
1. The paper is notationally dense, and I think the readers would benefit from natural language descriptions that provide intuitions beyond the formal descriptions of the variables (which I think is valuable as well given the goals of the paper). An example of where there is done well is the informal description of Hinitikka sets (lines 196-203).
2. Some of the text is very confusion, and consists of extremely long sentences (e.g., 233-236) -- I would suggest refraining from very long sentences, especially when discussing a specific technical detail (e.g., explaining a convoluted equation). It may help to first begin by providing an informal explanation, and then breaking down what each term in the formula is doing, versus reading out the formula in text.
3. In my opinion, the value added from the empirical experiments is marginal. The experiments, to the best of my understanding, estimate the impact of quantizing ACR-GNNs but don't actually look at verifying interesting properties over these GNNs. There is an implementation proposal (282-286) for problems with bounded number of vertices, and I think evaluating and releasing an implementation based on this proposal would be very valuable, and significantly strengthen the paper.
4. The paper has many typos, I list a few below:
- Missing citation (line 46)
- "have" to "have been" (line 77)
- Should this be K^3m->K^m (as opposed to n?) (line 97)
- 3agg(x1) should be replaced by 3agg(x2)? (line 140)
- the expressions say psi as opposed to the symbol (141-142)
- Is point 3 in Definition 5 accurate? There is an equality for  v >=k but then it becomes an equality?

---

> ### Author Rebuttal · Authors · 2025-07-30
>
> Thank you for your useful feedback and the numerous things you pointed out to improve our paper.
>
> ---
>
> ### Strengths
>
> Yes, while quantized ACR-GNNs have proved to be beneficial, evaluating the possiblity of verifying them formally, meaning the related complexity, is crucial for safe ML. Understanding that this verification is NEXPTIME-complete highlights to the community the algorithmic challenges of verifying them in their entirety. Section 6 offers a reason for optimism and a research direction for the community.
>
>
> ### Weaknesses
>
> 1. Yes, we aim to add a notation table in Appendix in a revised version of the paper.
> 2. Thank you for the suggestion, we will try to clear our wording in a potential final version.
> 3. Indeed, the experiments aim to evaluate the impact of quantizing ACR-GNNs. Although the work is theoretical, we wanted to demonstrate that the models are also of practical interest. We believe that this is import.
> We also wanted to provide at least a proof-of-concept for the verification of GNNs. The prototype implementation of the verification of GNN properties is part of the Supplementary Material; apologies for not making this clear in the text. Some performance results are reported in our answer to Reviewer jCNx.
> 4. Thanks for pointing these out! We appreciate your attention to detail and intend to fix these in a revised version.
>
> ## Questions
>
> > In Section 4.3, could you formally define the different sets and variables? X_H, H-vertices, S_H and so on?
>
> X_H is not a set of variables but a *set variable*. The variable X_H denotes a set [[X_H]]_G of vertices in a graph G. An H-vertex u is a vertex in a graph G such that all formulas in H are true for G and u. Again, S_H is a set variable (and not a set of variables!). The variable S_H denotes the set of all successors of an H-vertex.
>
> > It would be nice to add informal proof sketches for the key results where possible. I haven't checked the proofs in the appendix in detail, and an informal proof describing the key ideas/approach would help verify quickly.
>
> Here are some proposed ideas (just as short bullet points) which we try to add in a revision where space allows it:
> - Lemma 3: By induction on phi.
> - Proposition 7: We could add the short proof to the main body.
> - Proposition 8: We perform a reduction to the QFBAPA satisfiability by encoding the constraints of $\mathbb K$
> - Proposition 9: Via loops over $2^{n|\phi|}$ Hintikka sets, and $2^n$ values in $\mathbb K$.
> - Proposition 10: tr(\phi) encodes all the constraints for phi to be satisfiable. (=>) Given G, u satisfying phi, we assign the variables X_{expr' = k}, X_H and S_H to their intuitive meaning and check that tr(phi) is true wrt to that assignment. (<=) We construct a graph where the set of vertices is the interpretation of $\mathcal U$, while the set of edges is defined wrt to the interpretation of X_H and S_H, and the labeling is defined wrt to X_{x_i=k}.
> - Theorem 14: We could add the short proof to the main body.
>
>
> > There are a few example graphs and specifications in the paper, but I think it may help to extend those to Section 4 in the form of a running example
>
> The main issue is that the number of Hintikka sets is exponential in phi and n. So writing down formulas (1-4) would take a lot of space. We try to come up with a small example in a revised version if space allows it.

---

> ### Comment · Reviewer_j9GH · 2025-08-07
> **Thank you for the response!**
>
> I have read the response, and thank you for addressing some of the concerns raised from my end and the other reviewers. The empirical results shared with Reviewer jCNx are interesting indeed, and should also be included in the paper. I think pending a larger scale empirical investigation into the practicality of the proposed implementation/prototype, I am going keep my current assessment of the paper. Adding an example in the supplementary section would also be useful to readers!

---

### Note · Authors · 2025-08-12

We sincerely thank the reviewers and the ACs for their time and effort. We will implement the changes as indicated in the comments.

We hope the PC will appreciate the main contribution that demonstrates that the verification of GNNs is provably hard. No research has shown this before. We believe that it is significant and useful for the community of neural information processing systems. Our theoretical paper mainly helps understand the inherent complexity in the field of GNN verification.

A practical solution to verifying GNNs needs further research that is prompted, but extends far beyond the theoretical work of this paper.

Our results are substantial and largely exceed those of comparable research in the field. In this paper, we:
- address GNNs used in practice (as they are quantized)
- make a correspondence between quantized GNNs and logic
- provide a tight complexity result (both upper bound and lower bound)
- provide a future pathway to verify GNNs
- provide experiments to validate the use of quantized GNNs

---

### Decision · Program_Chairs · 2025-09-17

**Decision:**

Reject

**Comment:**

The reviewers all like the paper and unanimously recommend a borderline accept. The main case against the paper is that the clarity of the write-up needs to be improved. If there is sufficient space for the paper, it could be accepted but in light of the quality of other submissions, for now I will not recommend to accept the paper for publication at this year's NeurIPS. I wish the authors the best of success with their next version.